# Optimal Single-Policy Sample Complexity and Transient Coverage for Average-Reward Offline RL

**Matthew Zurek**
Department of Computer Sciences
University of Wisconsin–Madison
matthew.zurek@wisc.edu

**Guy Zamir**
Department of Computer Sciences
University of Wisconsin–Madison
gzamir@wisc.edu

**Yudong Chen**
Department of Computer Sciences
University of Wisconsin–Madison
yudongchen@cs.wisc.edu

## Abstract

We study offline reinforcement learning in average-reward MDPs, which presents increased challenges from the perspectives of distribution shift and non-uniform coverage, and has been relatively underexamined from a theoretical perspective. While previous work obtains performance guarantees under single-policy data coverage assumptions, such guarantees utilize additional complexity measures which are uniform over all policies, such as the uniform mixing time. We develop sharp guarantees depending only on the target policy, specifically the bias span and a novel policy hitting radius, yielding the first fully single-policy sample complexity bound for average-reward offline RL. We are also the first to handle general weakly communicating MDPs, contrasting restrictive structural assumptions made in prior work. To achieve this, we introduce an algorithm based on pessimistic discounted value iteration enhanced by a novel quantile clipping technique, which enables the use of a sharper empirical-span-based penalty function. Our algorithm also does not require any prior parameter knowledge for its implementation. Remarkably, we show via hard examples that learning under our conditions requires coverage assumptions beyond the stationary distribution of the target policy, distinguishing single-policy complexity measures from previously examined cases. We also develop lower bounds nearly matching our main result.

## 1 Introduction

Reinforcement learning (RL) has achieved impressive results for many control problems where it is possible to collect large amounts of experience through online interaction with the environment. However, many real-world application areas where we would like to apply RL methods, such as robotics, education, or healthcare, there may not exist simulators and data collection can be expensive or dangerous. Offline RL is a subfield of RL which seeks to address these issues by learning from historical data without online interaction, and hence achieving the maximum possible statistical efficiency is the paramount concern. The lack of online experience collection poses many related challenges to offline RL methods. One issue, often termed *distribution shift*, is that improving a policy's performance will inherently change the distribution of states and actions it experiences, potentially moving it away from the distribution of the historical dataset. Another closely related issue, sometimes referred to as *non-uniform coverage*, is that our dataset may generally be unevenly

39th Conference on Neural Information Processing Systems (NeurIPS 2025).

concentrated so that it is impossible to estimate the performance of all policies to uniform accuracy, and instead we must balance exploitation with varying degrees of confidence.

Recent research has made significant progress on the theoretical limits of offline RL by addressing these issues. However, many of these advances have been confined to the finite horizon setting, or the discounted infinite horizon setting, which can also behave like a finite horizon due to the irrelevance of distant future rewards. In this paper we focus on the challenging average-reward setting where the goal is to maximize the long-term average of rewards, which has been underexplored from a theoretical perspective. We briefly argue that the two aforementioned difficulties are amplified in the average-reward setting, and have not been satisfactorily addressed by previous work. First, since the average-reward objective captures performance in the long-horizon limit, we must contend with distribution shifts that occur after arbitrarily long time scales. Secondly, the issue of non-uniform coverage is magnified because while the (effective) horizon can serve as an extrinsic upper bound on the complexity of a particular policy, in the average-reward setting different policies can have arbitrarily different intrinsic complexities (as measured by parameters such as the span of the policy's relative value function). Existing work has developed algorithms which succeed under single-policy data coverage assumptions/concentrability coefficients, but has only done so when also using parameters that upper bound the complexity of all policies. Such large uniform-policy complexity measures can lead to vacuous bounds and overall fail to fully address both of the above issues. Additionally, algorithms from prior work fail to obtain optimal statistical efficiency and require foreknowledge of unlearnable parameters (such as coverage coefficients or environmental complexity parameters) for their implementation.

## 1.1 Our contributions

We address all of these challenges, developing an algorithm for (single-policy coverage) offline average-reward RL which is the first to handle the weakly communicating setting where not all policies have constant gains, as well as the first to obtain a convergence rate dependent on the bias span of only the target policy (as opposed to uniform complexity measures). Informally, our main theorem provides a high-probability guarantee on the suboptimality of the output policy $\widehat{\pi}$ of the form

$$\left\|\rho^\star - \rho^{\widehat{\pi}}\right\|_\infty \leq \widetilde{O}\left(\sqrt{\frac{S\|h^{\pi^\star}\|_{\mathrm{span}}}{m}}\,\right), \tag{1}$$

where $\|h^{\pi^\star}\|_{\mathrm{span}}$ is the bias-span of the target policy $\pi^\star$ and $S$ is the number of states. This holds whenever the sample size $n(s,a)$ per state-action pair $(s,a)$ satisfies $n(s,\pi^\star(s)) \geq m\mu^{\pi^\star}(s) + \widetilde{O}\left(T_{\mathrm{hit}}(P,\pi^\star)^2\right)$ for all states $s$. Here $\mu^{\pi^\star}$ is the stationary distribution of the target policy, $m$ is the "effective dataset size," and $T_{\mathrm{hit}}(P,\pi^\star)$ is a novel *policy hitting radius* that measures the time for $\pi^\star$ to reach a particular state in the support of its stationary distribution, and is thus also a single-policy complexity measure.

Interestingly, this condition requires data even for state-action pairs $(s,\pi^\star(s))$ for which $s$ is transient ($\mu^{\pi^\star}(s) = 0$) under the target policy, and we show via a hard example that this requirement is nearly unimprovable. In particular, this implies two surprising findings: i) with a fully "single-policy" sample complexity, learning a near-optimal policy is impossible under coverage conditions with respect to only the stationary distribution of the target policy, even with arbitrarily large amounts of data; ii) on the other hand, only a bounded amount of data from the transient state-action pairs of the target policy is sufficient to achieve vanishing suboptimality. We also show another lower bound which implies the optimality of the guarantee (1) in terms of its dependence on $m$, making our result the first among offline average-reward RL approaches to achieve an optimal rate for large $m$.

Our algorithm is based upon a pessimistic discounted value iteration procedure, involving a very large and prior-knowledge-free choice of discount factor. Most notably we develop a *quantile clipping* technique which enables the use of a sharper empirical-span-based penalty function.

## 1.2 Related work

First we discuss prior work on average-reward offline RL. To the best of our knowledge the only works with explicit results for this setting are Ozdaglar et al. [2024] and Gabbianelli et al. [2023]. Ozdaglar et al. [2024] assume that the MDP is unichain, and obtain guarantees with a constrained linear

programming (LP) algorithm in terms of the uniform mixing time $\tau_{\text{unif}}$ (defined in Section 2), for both general function approximation and tabular settings. We also discuss quantitative comparisons to the tabular results from Ozdaglar et al. [2024] after presenting our main theorem. Gabbianelli et al. [2023] assume that all policies in the MDP have constant (state-independent) gain, which is more general than unichain MDPs but does not hold in weakly communicating MDPs. Gabbianelli et al. [2023] consider the linear MDP setting, develop an algorithm based on primal-dual methods for solving LPs, and obtain guarantees in terms of a uniform bound on the span of all policies $H_{\text{unif}}$. The algorithms in both of these works require knowledge of certain concentrability coefficients.

Next we briefly discuss related work for offline RL outside of the average-reward setting. Our algorithm is essentially a careful refinement of the pessimistic value iteration approach of Li et al. [2023] for the discounted tabular setting, which in turn is a refinement of Rashidinejad et al. [2022]. Many works (e.g., Liu et al. [2020], Jin et al. [2021], Xie et al. [2021], Uehara and Sun [2021], Rashidinejad et al. [2022]) have demonstrated the ability for pessimistic approaches to address the distribution shift/non-uniform coverage challenges of offline RL and achieve near-optimal performance under single-policy concentrability assumptions.

Finally we discuss prior work on average-reward RL under uniform coverage assumptions. Many papers on average-reward RL considering the tabular generative model setting [Kearns and Singh, 1998] actually only require a dataset with an equal number of samples from all state-action pairs (e.g., Wang et al. [2022, 2023], Zurek and Chen [2024, 2025a,b]), and hence we believe such papers could be easily extended to the uniform coverage setting, obtaining a guarantee dependent on the smallest number of samples for any state-action pair. While such works might be considered offline RL, we reserve this term for guarantees involving only single-policy coverage assumptions. Achieving instance-dependent guarantees in terms of the bias span of an optimal policy (e.g., Zhang and Xie [2023], Wang et al. [2022], Zurek and Chen [2025b]) and removing the need for prior knowledge of complexity parameters (e.g., Jin et al. [2024], Neu and Okolo [2024], Tuynman et al. [2024], Zurek and Chen [2025a]) have been the objectives of extensive research in the uniform coverage setting.

## 2 Background and problem setup

### 2.1 Background

A Markov decision process (MDP) is a tuple $(\mathcal{S}, \mathcal{A}, P, r)$ where $\mathcal{S}$ and $\mathcal{A}$ respectively denote the finite state and action spaces, $P : \mathcal{S} \times \mathcal{A} \to \Delta(\mathcal{S})$ is the transition kernel (with $\Delta(\mathcal{S})$ denoting the probability simplex on $\mathcal{S}$), and $r : [0, 1]^{\mathcal{S} \times \mathcal{A}}$ is the reward function. We let $S = |\mathcal{S}|$ and $A = |\mathcal{A}|$. We generally omit the explicit reference to $\mathcal{S}$ and $\mathcal{A}$ when defining MDPs. A (Markovian/stationary) policy is a mapping $\pi : \mathcal{S} \to \Delta(\mathcal{A})$. We call a policy deterministic if for all $s \in \mathcal{S}$, $\pi(s)$ only places probability mass on one action, and in this case we also treat $\pi$ as a mapping $\mathcal{S} \to \mathcal{A}$. Let $\Pi$ denote the set of all stationary deterministic policies. An initial state $s_0 \in \mathcal{S}$ and policy $\pi$ induce a distribution over trajectories $(s_0, A_0, S_1, A_1, \dots)$ where $A_t \sim \pi(S_t)$, $S_{t+1} \sim P(\cdot \mid S_t, A_t)$, and we let $\mathbb{E}_{s_0}^\pi$ denote the expectation with respect to this distribution. We often treat $P$ as an $(\mathcal{S} \times \mathcal{A})$-by-$\mathcal{S}$ matrix where $P_{sa,s'} = P(s' \mid s, a)$, and let $P_{sa}$ denote the $sa$-th row of this matrix (treated as a "row vector", so $P_{sa}X = \sum_{s'} P_{sa}(s')X(s')$ for $X \in \mathbb{R}^{\mathcal{S}}$). For $X \in \mathbb{R}^{\mathcal{S}}$ and $s \in \mathcal{S}, a \in \mathcal{A}$, define the next-state value variance $\mathbb{V}_{P_{sa}}[X] = \sum_{s' \in \mathcal{S}} P(s' \mid s, a)X(s')^2 - (\sum_{s' \in \mathcal{S}} P(s' \mid s, a)X(s'))^2$.

A discounted MDP is a tuple $(\mathcal{S}, \mathcal{A}, P, r, \gamma)$ where $\gamma \in [0, 1)$ is the discount factor. For a policy $\pi$, the discounted value function $V_\gamma^\pi \in [0, \frac{1}{1-\gamma}]^{\mathcal{S}}$ is defined $V_\gamma^\pi(s) = \mathbb{E}_s^\pi[\sum_{t=0}^\infty \gamma^t R_t]$ where $R_t = r(S_t, A_t)$, and the gain $\rho^\pi \in [0, 1]^{\mathcal{S}}$, is $\rho^\pi(s) = \text{C-lim}_{t \to \infty} \mathbb{E}_s^\pi[R_t] = \lim_{T \to \infty} \frac{1}{T} \mathbb{E}_s^\pi[\sum_{t=0}^{T-1} R_t]$ where C-lim is the Cesaro limit. We define the optimal gain $\rho^\star = \sup_{\pi \in \Pi} \rho^\pi$, and we say a policy $\pi$ is gain-optimal if $\rho^\pi = \rho^\star$. A gain-optimal policy always exists [Puterman, 1994]. The bias function of a policy $\pi$, $h^\pi \in \mathbb{R}^{\mathcal{S}}$, is $h^\pi(s) = \text{C-lim}_{T \to \infty} \mathbb{E}_s^\pi[\sum_{t=0}^{T-1}(R_t - \rho^\pi(S_t))]$.

$M : \mathbb{R}^{\mathcal{S} \times \mathcal{A}} \to \mathbb{R}^{\mathcal{S}}$ denotes the action maximization operator where $M(Q)(s) = \max_{a \in \mathcal{A}} Q(s, a)$, and $M^\pi$ denotes the policy matrix where $M^\pi(Q)(s) = \sum_{a \in \mathcal{A}} \pi(s)(a)Q(s, a)$, for any $Q \in \mathbb{R}^{\mathcal{S} \times \mathcal{A}}$, $s \in \mathcal{S}$, and policy $\pi$. We often drop the parenthesis and write $MQ := M(Q)$. For any $Q \in \mathbb{R}^{\mathcal{S} \times \mathcal{A}}$, the discounted (action-value) Bellman operator $\mathcal{T} : \mathbb{R}^{\mathcal{S} \times \mathcal{A}} \to \mathbb{R}^{\mathcal{S} \times \mathcal{A}}$ is $\mathcal{T}(Q) := r + \gamma P M(Q)$, and the policy-evaluation Bellman operator $\mathcal{T}^\pi$ is $\mathcal{T}^\pi(Q) := r + \gamma P M^\pi Q$, for any policy $\pi$.

Let $\mathbb{N} = \{1, 2, \dots\}$ denote the set of natural numbers. Define $\mathbf{0}, \mathbf{1}$ as the all-zero and all-one vectors, respectively. For $X \in \mathbb{R}^{\mathcal{S}}$, let $\|X\|_{\mathrm{span}} = \max_{s \in \mathcal{S}} X(s) - \min_{s \in \mathcal{S}} X(s)$ denote the span semi-norm. We use $\widetilde{O}(\cdot), \widetilde{\Theta}(\cdot), \widetilde{\Omega}(\cdot)$ notation to ignore constants as well as logarithmic factors in $S, A, \frac{1}{1-\gamma}, \frac{1}{\delta}$, and $n_{\mathrm{tot}}$, where $\delta$ and $n_{\mathrm{tot}}$ are the failure probability and the total dataset size, to be defined below. Let $e_s \in \mathbb{R}^{\mathcal{S}}$ denote the vector which is all zero except for a $1$ in entry $s \in \mathcal{S}$. For two vectors $v, v' \in \mathbb{R}^d$, $v \geq v'$ denotes the elementwise inequality $v(i) \geq v'(i)$ for all $i$.

Under the transition kernel $P$, a policy $\pi$ induces a Markov chain over state $\mathcal{S}$, whose transition matrix is denoted by $P_\pi$. The policy $\pi$ is said to be unichain if it induces a unichain Markov chain, meaning that the chain consists of a single (irreducible) recurrent class plus a possibly empty set of transient states. An MDP is unichain if all deterministic policies in the MDP are unichain. An MDP is communicating (aka strongly connected) if for any pair of states $s, s' \in \mathcal{S}$, $s'$ is accessible from $s$, meaning there exists some policy $\pi$ and some $k \in \mathbb{N}$ such that $\mathbb{E}_s^\pi \mathbb{I}(S_k = s') > 0$. An MDP is weakly communicating if it consists of a set of states $\mathcal{S}_c$ such that, for any $s, s' \in \mathcal{S}_c$, $s'$ is accessible from $s$, plus a set of states $\mathcal{S}_t = \mathcal{S} \setminus \mathcal{S}_c$ which are transient under all policies. All unichain and communicating MDPs are weakly communicating.

A unichain policy $\pi$ has constant (state-independent) $\rho^\pi$, and thus in unichain MDPs, all policies have constant gains. In weakly communicating MDPs, the optimal gain $\rho^\star$ is constant, but sub-optimal policies $\pi$ may have non-constant $\rho^\pi$. For any unichain policy $\pi$, we write its (unique) stationary distribution as $\mu^\pi \in \mathbb{R}^{\mathcal{S}}$ (which we treat as a "row vector"). For any unichain policy $\pi$, we define its mixing time $\tau(\pi) = \inf\{t \geq 0 : \|e_s^\top P_\pi^t - \mu^\pi\|_1 \leq \frac{1}{2}\}$. Define the uniform mixing time as $\tau_{\mathrm{unif}} = \sup_{\pi \in \Pi} \tau(\pi)$. Also define the uniform span bound $H_{\mathrm{unif}} = \sup_{\pi \in \Pi} \|h^\pi\|_{\mathrm{span}}$. For any $s \in \mathcal{S}$, let $\eta_s := \inf\{t \geq 0 : S_t = s\}$ be the first hitting time of state $s$. Define the diameter $D = \max_{s,s' \in \mathcal{S}} \min_{\pi \in \Pi} \mathbb{E}_s^\pi[\eta_{s'}]$, and we sometimes write $D_P$ to emphasize the dependence on $P$.

## 2.2 Offline RL setting

We assume a sample size function $n : \mathcal{S} \times \mathcal{A} \to \mathbb{N}$ is fixed a priori, and for each $s \in \mathcal{S}, a \in \mathcal{A}$, we assume that we have $n(s, a)$ samples $S_{s,a}^1, \dots, S_{s,a}^{n(s,a)}$ sampled independently from the next-state transition distribution $P(\cdot \mid s, a)$. We define the dataset $\mathcal{D} = \big((s, a, S_{s,a}^i)\big)_{s \in \mathcal{S}, a \in \mathcal{A}, 1 \leq i \leq n(s,a)}$ and let $n_{\mathrm{tot}} = \sum_{s \in \mathcal{S}, a \in \mathcal{A}} n(s, a)$ denote the total dataset size. We assume the reward function $r$ is known.

We introduce a new quantity which plays a key role in both our main theorem and our lower bounds. For any transition kernel matrix $P$ and policy $\pi$, we define the *policy hitting radius*

$$T_{\mathrm{hit}}(P, \pi) := \inf_{s^\star \in \mathcal{S}} \sup_{s_0 \in \mathcal{S}} \mathbb{E}_{s_0}^\pi[\eta_{s^\star}], \tag{2}$$

where again $\eta_s$ is the first hitting time of state $s$. In words, $T_{\mathrm{hit}}(P, \pi)$ measures the largest expected amount of time required to hit the "center" state $s^\star$, for the optimal choice of $s^\star$ (which will always be a recurrent state). As shown in Lemma B.10, $T_{\mathrm{hit}}(P, \pi)$ is always finite if $P_\pi$ is unichain. We also always have that $\|h^\pi\|_{\mathrm{span}} \leq 4T_{\mathrm{hit}}(P, \pi)$ for any $\pi$ (Lemma B.13). There is generally no relationship between $T_{\mathrm{hit}}(P, \pi)$ and $\tau(\pi)$; see the discussion in Appendix B.5.1.

# 3 Main results

## 3.1 Algorithm

First we describe the algorithm used to obtain our main result. We employ a discounted reduction approach, i.e., approximating the average-reward MDP by a discounted MDP with an appropriate choice of discount factor. The main component of our approach, Algorithm 1, is a pessimistic value iteration subroutine which can be understood as solving a discounted MDP.

Now we define the pessimistic Bellman operator $\widehat{\mathcal{T}}_{\mathrm{pe}} : \mathbb{R}^{\mathcal{S} \times \mathcal{A}} \to \mathbb{R}^{\mathcal{S} \times \mathcal{A}}$ used in Algorithm 1. $\widehat{\mathcal{T}}_{\mathrm{pe}}$ is a function of $\gamma$ as well as the dataset $\mathcal{D}$, utilizing the empirical transition matrix $\widehat{P}$ where $\widehat{P}(s' \mid s, a) = \frac{1}{n(s,a)} \sum_{i=1}^{n(s,a)} \mathbb{I}(S_{sa}^i = s')$. If $n(s, a) = 0$ for some $s, a$ then for concreteness we

---

**Algorithm 1** Pessimistic Value Iteration With Quantile Clipping

---

**input:** Dataset $\mathcal{D}$, reward function $r$, discount factor $\gamma \in (0,1)$, failure probability $\delta \in (0,1)$

1: Form empirical transition matrix $\widehat{P}$ used in $\widehat{\mathcal{T}}_{\mathrm{pe}}$ from $\mathcal{D}$

2: Let $\widehat{Q}_0 = \mathbf{0}$ and $K = \left\lceil \frac{\log\left(\frac{2n_{\mathrm{tot}}}{1-\gamma}\right)}{1-\gamma} \right\rceil$          ▷ initialization and number of iterations

3: **for** $t = 1, \ldots, K$ **do**

4:      Let $\widehat{Q}_t = \widehat{\mathcal{T}}_{\mathrm{pe}}(\widehat{Q}_{t-1})$

5: **end for**

6: Let $\widehat{Q} = \widehat{Q}_K$ and for each $s \in \mathcal{S}$, let $\widehat{\pi}(s) \in \mathrm{argmax}_{a \in \mathcal{A}} \widehat{Q}(s,a)$

7: **return** $\widehat{\pi}, \widehat{Q}$

---

define $\widehat{P}(s' \mid s, a) = 1/S$, although any default probability distribution over $\mathcal{S}$ would be fine, since our construction of $\widehat{\mathcal{T}}_{\mathrm{pe}}$ does not depend on rows $\widehat{P}_{sa}$ such that $n(s,a) = 0$.[1]

For any $Q \in \mathbb{R}^{\mathcal{S} \times \mathcal{A}}$ and any $s \in \mathcal{S}, a \in \mathcal{A}$, we define

$$\widehat{\mathcal{T}}_{\mathrm{pe}}(Q)(s,a) := r(s,a) + \gamma \max\left\{ \widehat{P}_{sa} T_{\beta(s,a)}(\widehat{P}_{sa}, MQ) - b(s,a,MQ), \min_{s'}(MQ)(s') \right\}. \quad (3)$$

Here $MQ \in \mathbb{R}^{\mathcal{S}}$ takes the maximum over actions of the Q-function $Q$ (and thus should be understood as the corresponding value function). The term $b(s,a,MQ) \geq 0$ is a certain Bernstein-style penalty, which is chosen below to ensure that $\widehat{\mathcal{T}}_{\mathrm{pe}}(Q)$ lower-bounds the true (unknown) Bellman operator $\mathcal{T}(Q)$ for any $Q$. The expression $\widehat{P}_{sa} T_{\beta(s,a)}(\widehat{P}_{sa}, MQ)$ denotes the inner product of the probability distribution $\widehat{P}_{sa}$ with the vector $T_{\beta(s,a)}(\widehat{P}_{sa}, MQ) \in \mathbb{R}^{\mathcal{S}}$, which is a "quantile-clipped" version of $MQ$ to be defined momentarily. For $\beta \in [0,1]$, the quantile clipping operator $T_\beta : \mathbb{R}^{\mathcal{S}} \times \mathbb{R}^{\mathcal{S}} \to \mathbb{R}^S$ is defined as follows: for any $V \in \mathbb{R}^{\mathcal{S}}$, $s \in \mathcal{S}$, and probability distribution $\mu \in \mathbb{R}^{\mathcal{S}}$, let

$$T_\beta(\mu, V)(s) = \min\left\{ V(s), \sup\left\{ V(s') : s' \in \mathcal{S}, \sum_{s'' \in \mathcal{S}:V(s'') \geq V(s')} \mu(s') \geq \beta \right\} \right\}. \quad (4)$$

In words, all entries of $V$ larger than the (largest) $1 - \beta$ quantile with respect to $\mu$ are clipped down to this quantile. To extend the definition to $\beta > 1$, we set $T_\beta(\mu, V)(s) = \min_{s' \in \mathcal{S}} V(s')$, that is all entries will be clipped to the minimum entry of $V$. Finally we define the penalty term

$$b(s,a,V) = \max\left\{ \sqrt{\beta(s,a)\mathbb{V}_{\widehat{P}_{sa}}\left[T_{\beta(s,a)}(\widehat{P}_{sa}, V)\right]}, \beta(s,a)\left\|T_{\beta(s,a)}(\widehat{P}_{sa}, V)\right\|_{\mathrm{span}} \right\} + \frac{5}{n_{\mathrm{tot}}} \quad (5)$$

where $\alpha = 8\log\left(\frac{6S^2 An_{\mathrm{tot}}}{(1-\gamma)\delta}\right)$ and $\beta(s,a) = \frac{\alpha}{\max\{n(s,a)-1,1\}}$. Note that $\beta(s,a) = \widetilde{O}(\frac{1}{n(s,a)})$ (whenever $n(s,a) > 0$).

The pessimistic Bellman operator $\widehat{\mathcal{T}}_{\mathrm{pe}}$ has several nice properties that are crucial to our analysis.

**Lemma 3.1.** $\widehat{\mathcal{T}}_{\mathrm{pe}}$ *satisfies the following:*

    *1. Monotonicity: If $Q \geq Q'$ then $\widehat{\mathcal{T}}_{\mathrm{pe}}(Q) \geq \widehat{\mathcal{T}}_{\mathrm{pe}}(Q')$.*

    *2. Constant shift: For any $c \in \mathbb{R}$, $\widehat{\mathcal{T}}_{\mathrm{pe}}(Q + c\mathbf{1}) = \widehat{\mathcal{T}}_{\mathrm{pe}}(Q) + \gamma c\mathbf{1}$.*

    *3. $\gamma$-contractivity: $\widehat{\mathcal{T}}_{\mathrm{pe}}$ is a $\gamma$-contraction and has a unique fixed point $\widehat{Q}^\star_{\mathrm{pe}} \in [0, \frac{1}{1-\gamma}]^{\mathcal{S}}$.*

See Lemma B.1 for a more complete statement. In summary, like previous pessimistic value iteration approaches [Li et al., 2023, Rashidinejad et al., 2022], our pessimistic Bellman operator shares key properties with usual Bellman operators enabling us to find an approximate fixed point in $\widetilde{O}(\frac{1}{1-\gamma})$ value iteration steps, and then we will choose policy $\widehat{\pi}$ to be greedy with respect to this fixed point.

Now we discuss the motivation for quantile clipping, and the differences from prior work. In particular we highlight the constant shift property enjoyed by $\widehat{\mathcal{T}}_{\mathrm{pe}}$. This is highly desirable for the average-reward

---

[1] If $n(s,a) = 0$ then $\beta(s,a) = \alpha > 1$ and $T_{\beta(s,a)}(\widehat{P}_{sa}, MQ) = (\min_{s'}(MQ)(s'))\mathbf{1}$, causing the max in (3) to equal $\min_{s'}(MQ)(s')$.

setting, and more generally any weakly communicating MDPs, since in such MDPs the optimal value function behaves as $V_\gamma^\star \approx \frac{1}{1-\gamma}\rho^\star + h^\star$ and $\rho^\star$ is a multiple of $\mathbf{1}$. The constant shift property essentially guarantees that we only penalize the variability in the relative value differences between states, not the overall horizon-dependent scale $\frac{1}{1-\gamma}$ of the cumulative rewards. The $\|\cdot\|_{\text{span}}$-based second term in our penalty function definition (5) of $b$ is essential for this constant-shift property, since the span semi-norm is invariant to translation by multiples of $\mathbf{1}$. Previous "Bernstein-style" penalty functions [Li et al., 2023] use a larger term like $\beta(s,a)\frac{1}{1-\gamma} \approx \frac{1}{n(s,a)}\frac{1}{1-\gamma}$, which breaks the constant shift property and can dominate the first (variance-based) term in (5) when used with large horizons. Naively using $\beta(s,a)\|V\|_{\text{span}}$ in the second term of (5) actually fails to ensure the monotonicity and contractivity properties of $\widehat{\mathcal{T}}_{\text{pe}}$, for reasons that we elaborate upon in Section 4. Fortunately, the introduction of quantile clipping remedies these issues, and only introduces small additional bias: since only entries representing at most $\beta(s,a) = \widetilde{O}(\frac{1}{n(s,a)})$ of the probability mass with respect to $\widehat{P}_{sa}$ have their values clipped, we have $\widehat{P}_{sa}T_{\beta(s,a)}(\widehat{P}_{sa},V) \leq \widehat{P}_{sa}V \leq \widehat{P}_{sa}T_{\beta(s,a)}(\widehat{P}_{sa},V) + \beta(s,a)\|V\|_{\text{span}}$, and introducing quantile clipping within the two terms of the penalty function $b$ in (5) only reduces the penalty value, relative to instead using $\mathbb{V}_{\widehat{P}_{sa}}[V]$ and $\|V\|_{\text{span}}$. (See Lemma B.14.)

## 3.2 Main theorem

Now we present our main theorem on the performance of Algorithm 1. We will apply Algorithm 1 with a very large discount factor $\gamma$ such that the effective horizon is $\frac{1}{1-\gamma} = n_{\text{tot}}$.

**Theorem 3.2.** *There exist absolute constants $C_1, C_2$ such that the following holds: Fix $\delta > 0$. Let $\gamma = 1 - \frac{1}{n_{\text{tot}}}$ and $\alpha = 8\log\left(\frac{6S^2 A n_{\text{tot}}}{(1-\gamma)\delta}\right)$. Let $\pi^\star$ be a deterministic gain-optimal policy which is unichain with stationary distribution $\mu^{\pi^\star}$. Suppose there exists some $m \in \mathbb{N}$ such that*

$$n(s, \pi^\star(s)) \geq m\mu^{\pi^\star}(s) + \alpha\left(C_2 T_{\text{hit}}(P, \pi^\star)\right)^2 + 4.$$

*Then letting $\widehat{\pi}$ be the policy returned by Algorithm 1 with inputs $\mathcal{D}$, $r$, $\gamma = 1 - \frac{1}{n_{\text{tot}}}$, and $\delta$, we have with probability at least $1 - 5\delta$ that*

$$\rho^{\widehat{\pi}} \geq \rho^\star - \sqrt{\frac{C_1 S(\|h^{\pi^\star}\|_{\text{span}} + 1)\alpha}{m}}.$$

We prove Theorem 3.2 in Appendix B. Theorem 3.2 demonstrates that as the "effective dataset size" $m$ increases, the suboptimality of $\widehat{\pi}$ decreases at a rate of $\widetilde{O}(\sqrt{S\|h^{\pi^\star}\|_{\text{span}}/m})$, which matches our lower bound Theorem 3.4. Our coverage assumption is qualitatively different than previous works on average-reward RL, since even for states $s$ which are transient under $\pi^\star$ (and thus have $\mu^{\pi^\star}(s) = 0$), we still require $\widetilde{O}(T_{\text{hit}}(P, \pi^\star)^2)$ samples from the state-action pair $(s, \pi^\star(s))$. Note that up to a log factor this transient state coverage assumption is independent of $m$, meaning that vanishing suboptimality is possible with only an essentially bounded amount of data from transient states. (In the absence of this additional term we could treat $n_{\text{tot}}/m$ as a "concentrability coefficient" similar to prior work, but we believe our results are stated more clearly in terms of the effective dataset size $m$.) As shown in Theorem 3.3, this transient data requirement is necessary to obtain a $\|h^{\pi^\star}\|_{\text{span}}$-based guarantee, and our dependence on $T_{\text{hit}}(P, \pi^\star)$ is nearly optimal. Theorem 3.2 requires $\pi^\star$ to be unichain, which is a mild assumption, since even in weakly communicating MDPs where not all policies are unichain, there always exists a unichain gain-optimal policy [Bertsekas, 2018].

No prior parameter knowledge, such as of $\|h^{\pi^\star}\|_{\text{span}}$ or the value of $m$ (or equivalently a coverage coefficient) is needed for Algorithm 1 to be implemented and enjoy the above guarantee. In particular $\gamma$ is set so that the effective horizon is $n_{\text{tot}}$. Actually our theorem would hold for arbitrarily larger choices of the effective horizon, and the guarantee would not degrade except for a logarithmic dependence on the effective horizon, but this would be suboptimal from a computational perspective, since $\widetilde{O}(1/(1-\gamma))$ iterations are required for convergence in Algorithm 1. Also see Theorem B.20 for a version of Theorem 3.2 allowing $\pi^\star$ to be gain-suboptimal.

In the unichain tabular setting, Ozdaglar et al. [2024] obtain a suboptimality bound like $\widetilde{O}(\sqrt{C^2\tau_{\text{unif}}^2 S/n_{\text{tot}}})$ where $C \geq 1$ is a certain coverage coefficient roughly equivalent to $n_{\text{tot}}/m$. With this substitution their bound becomes $\widetilde{O}(\sqrt{C\tau_{\text{unif}}^2 S/m})$, which interestingly degrades with the

coverage coefficient $C$ even as the effective dataset size $m$ is held constant, while our bound has no such issue. We also have $\|h^{\pi^\star}\|_{\mathrm{span}} \leq O(\tau_{\mathrm{unif}})$, and qualitatively $\|h^{\pi^\star}\|_{\mathrm{span}}$ is much sharper since it depends only on $\pi^\star$ rather than all policies.

## 3.3 Lower bounds

In this subsection we present two lower bounds implying the near-optimality of our Theorem 3.2. Below, for an MDP $(P_\theta, r)$, $\rho_\theta^\pi$, $h_\theta^\pi$ and $\mu_\theta^\pi$ denote the gain, bias and stationary distribution of a policy $\pi$, respectively; $\rho_\theta^*$ and $D_\theta$ denote the optimal gain and the diameter of the MDP, respectively; and $\mathbb{P}_{\theta,n}$ denotes the distribution of the dataset $\mathcal{D}$ under this MDP when the sample size function is $n$.

First, we present the surprising fact that, to obtain convergence rates dependent on certain single-policy complexity measures including $\|h^{\pi^\star}\|_{\mathrm{span}}$ and $T_{\mathrm{hit}}(P, \pi^\star)$, coverage assumptions with respect to only the stationary distribution of the target policy are insufficient to learn a near-optimal policy, even with an arbitrarily large amount of data.

**Theorem 3.3.** *For any $T \geq 4$ and any $m \in \mathbb{N}$, there exist a finite index set $\Theta$, transition matrices $P_\theta$ for each $\theta \in \Theta$, and a reward function $r$, such that for all $\delta \in \left(0, \frac{1}{e^9}\right]$, there exists a function $n : \mathcal{S} \times \mathcal{A} \to \mathbb{N}$ satisfying the following:*

1. *For each $\theta \in \Theta$, the MDP $(P_\theta, r)$ is unichain and communicating, with $A \leq O\left(\left\lceil \frac{m}{T} \right\rceil\right)$ actions and diameter $T$.*

2. *For each $\theta \in \Theta$, the MDP $(P_\theta, r)$ has a unique deterministic gain-optimal policy $\pi_\theta^\star$ such that $T_{\mathrm{hit}}(P_\theta, \pi_\theta^\star) \leq T$ and $n(s, \pi_\theta^\star(s)) \geq m\mu_\theta^{\pi_\theta^\star}(s) + \frac{T}{6}\log\left(\frac{1}{\delta}\right)$ for all $s \in \mathcal{S}$.*

3. *For any algorithm $\mathscr{A}$ that maps the dataset $\mathcal{D}$ to a stationary policy, we have*

$$\max_{\theta \in \Theta} \mathbb{P}_{\theta,n}\left(\rho_\theta^* - \rho_\theta^{\mathscr{A}(\mathcal{D})} > 1/2\right) \geq \delta.$$

Note that the "effective dataset size" parameter $m$ can be taken arbitrarily large, meaning that learning better than a $\frac{1}{2}$-suboptimal policy is impossible even with arbitrarily large amounts of data from the stationary distribution of the target policy. This does not contradict the error bounds from prior work which make stationary-distribution-based coverage assumptions and involve uniform complexity measures $\tau_{\mathrm{unif}}, H_{\mathrm{unif}}$ [Ozdaglar et al., 2024, Gabbianelli et al., 2023], since the parameters $\tau_{\mathrm{unif}}, H_{\mathrm{unif}}$ scale with $m$ in our hard instances in such a way as to render such bounds vacuous. In contrast, the parameters $\|h_\theta^{\pi_\theta^\star}\|_{\mathrm{span}}, T_{\mathrm{hit}}(P_\theta, \pi_\theta^\star)$, and $D_\theta$ remain bounded, implying that a convergence rate involving any of these parameters is impossible without data coverage beyond the stationary distribution, revealing a qualitatively different behavior of such parameters. While oftentimes results for average-reward setups can be predicted/derived by taking appropriate large-$\gamma$ limits of results for discounted settings, taking the limit as $\gamma \to 1$ of usual discounted occupancy coverage assumptions (e.g., $C^\star$ in Rashidinejad et al. [2022, Theorem 6]) only leads to requirements on covering the stationary distribution.

The setup in Theorem 3.3 even provides the learner with $\widetilde{\Omega}(T_{\mathrm{hit}}(P_\theta, \pi_\theta^\star))$ samples from state-action pairs which are transient under the target policy ($\mu_\theta^{\pi_\theta^\star}(s) = 0$), and this is still insufficient for learning near-optimal policies. This implies that the transient state dataset coverage requirement of Theorem 3.2 is nearly unimprovable, up to an additional factor of $\widetilde{O}(T_{\mathrm{hit}}(P, \pi^\star))$. A complete proof of Theorem 3.3 is provided in Appendix C and a sketch is provided in Section 4, but we briefly summarize the key idea: even with an arbitrarily large (but finite) amount of data from the recurrent class of the target policy, we may inevitably learn a policy with a small probability of leaving these well-covered states. Without any data we cannot learn how to recover from such a transition and navigate back to highly-rewarding regions quickly enough. This unfavorable but rare transition has negligible impact for finite horizon/discounted RL objectives (if the starting state is within the highly-rewarding region). In unichain MDPs all policies are guaranteed to eventually return to the recurrent class of the optimal policy eventually (because all recurrent classes must overlap, otherwise it would be possible to construct a multichain policy), but the fact that some policies take a long time to do so means that the uniform mixing time $\tau_{\mathrm{unif}}$ is very large, even if the optimal policy can recover quickly. Despite being unichain, such MDPs are qualitatively close to being non-unichain (but weakly communicating).

Next, we present a lower bound which demonstrates that dependence on $m$ in Theorem 3.2 is tight.

**Theorem 3.4.** *There exist absolute constants $c_1, c_2, c_3 > 0$ such that for any $T \geq c_1$, $S \geq c_2$, $k \geq 0$, and $m \geq \max\{TS, kS\}$, one can construct a finite index set $\Theta$, transition matrices $P_\theta$ for each $\theta \in \Theta$, a reward function $r$, and a function $n : \mathcal{S} \times \mathcal{A} \to \mathbb{N}$ such that the following hold:*

1. *For each $\theta \in \Theta$, the MDP $(P_\theta, r)$ is unichain and communicating, with $S$ states and diameter $T$.*

2. *For each $\theta \in \Theta$, the MDP $(P_\theta, r)$ has a unique stationary gain-optimal policy $\pi_\theta^\star$ such that $T_{\mathrm{hit}}(P_\theta, \pi_\theta^\star) \leq T$ and $n(s, \pi_\theta^\star(s)) \geq m\mu_\theta^{\pi_\theta^\star}(s) + k$ for all $s \in \mathcal{S}$.*

3. *For any algorithm $\mathscr{A}$ that maps the dataset $\mathcal{D}$ to a stationary policy, we have*

$$\max_{\theta \in \Theta} \mathbb{P}_{\theta, n}\left( \rho_\theta^* - \rho_\theta^{\mathscr{A}(\mathcal{D})} > c_3\sqrt{\frac{TS}{m}} \right) \geq \frac{1}{64}. \tag{6}$$

Since generally $T_{\mathrm{hit}}(P, \pi) \geq \|h^\pi\|_{\mathrm{span}}/4$ (see Lemma B.13), Theorem 3.2 implies a lower bound in terms of $\|h^{\pi^*}\|_{\mathrm{span}}$ ($\|h^{\pi_\theta^\star}_\theta\|_{\mathrm{span}}$ and $T_{\mathrm{hit}}(P_\theta, \pi_\theta^\star)$ are on the same order in the instances of Theorem 3.4). We add the parameter $k$ to demonstrate that a coverage requirement in the form of Theorem 3.2 does not affect the dependence on $m$ in (6) for sufficiently large $m$. In particular after setting $k = \widetilde{\Theta}(T^2)$ to match Theorem 3.2, its dependence on $\|h^{\pi^*}\|_{\mathrm{span}}, S$, and $m$ matches (6) and thus is unimprovable up to $\widetilde{O}(\cdot)$ factors as long as $m \geq \widetilde{\Theta}(T^2 S)$. Theorem 3.4 is proven in Appendix D.

## 4 Proof sketches

### 4.1 Main theorem

First we discuss the proof of Theorem 3.2, including the motivation for quantile clipping. The key idea of pessimistic value iteration is to choose $\widehat{\mathcal{T}}_{\mathrm{pe}}$ so that $\widehat{\mathcal{T}}_{\mathrm{pe}}(\widehat{Q}_{\mathrm{pe}}^\star) \leq \mathcal{T}(\widehat{Q}_{\mathrm{pe}}^\star)$, and then letting $\widehat{\pi}$ be greedy with respect to $\widehat{Q}_{\mathrm{pe}}^\star$ (meaning $\mathcal{T}(\widehat{Q}_{\mathrm{pe}}^\star) = \mathcal{T}^{\widehat{\pi}}(\widehat{Q}_{\mathrm{pe}}^\star)$), we have

$$\widehat{Q}_{\mathrm{pe}}^\star = \widehat{\mathcal{T}}_{\mathrm{pe}}(\widehat{Q}_{\mathrm{pe}}^\star) \leq \mathcal{T}(\widehat{Q}_{\mathrm{pe}}^\star) = \mathcal{T}^{\widehat{\pi}}(\widehat{Q}_{\mathrm{pe}}^\star)$$

so by standard monotonicity arguments we have $\widehat{Q}_{\mathrm{pe}}^\star \leq Q^{\widehat{\pi}}$. The challenge is then to choose $\widehat{\mathcal{T}}_{\mathrm{pe}}$ as "close" to $\mathcal{T}$ as possible, so that $\widehat{Q}_{\mathrm{pe}}^\star$ is as close as possible to $Q^\star$ (while ensuring $\widehat{\mathcal{T}}_{\mathrm{pe}}(\widehat{Q}_{\mathrm{pe}}^\star) \leq \mathcal{T}(\widehat{Q}_{\mathrm{pe}}^\star)$), in order to maximize $Q^{\widehat{\pi}}$. Using $\alpha$ to hide $\widetilde{O}(\cdot)$ terms, an empirical Bernstein-like bound [Maurer and Pontil, 2009] for the quantity $\widehat{V}_{\mathrm{pe}}^\star = M\widehat{Q}_{\mathrm{pe}}^\star$, and upper-bounding a sum by $\max$, yields

$$P_{sa}\widehat{V}_{\mathrm{pe}}^\star \geq \widehat{P}_{sa}\widehat{V}_{\mathrm{pe}}^\star - \max\left\{ \sqrt{\alpha\frac{\mathbb{V}_{\widehat{P}_{sa}}\left[\widehat{V}_{\mathrm{pe}}^\star\right]}{n(s, a)}}, \alpha\frac{\left\|\widehat{V}_{\mathrm{pe}}^\star\right\|_{\mathrm{span}}}{n(s, a)} \right\} =: \widehat{P}_{sa}\widehat{V}_{\mathrm{pe}}^\star - \tilde{b}(s, a, \widehat{V}_{\mathrm{pe}}^\star) \ \forall s, a. \tag{7}$$

This sharp span-based form of penalty function $\tilde{b}$ is crucial for the constant shift property described in Lemma 3.1, since both $\mathbb{V}_{\widehat{P}_{sa}}[\cdot]$ and $\|\cdot\|_{\mathrm{span}}$ are invariant to shifts by multiples of $\mathbf{1}$. As discussed there this property is essential for the average-reward setting, and the Bernstein-style penalty used in Li et al. [2023] replaces the second term from the max in (7) with $\frac{1}{1-\gamma} \geq \|\widehat{V}_{\mathrm{pe}}^\star\|_{\mathrm{span}}$ and hence does not enjoy this property. However, we cannot simply use an operator like $\widetilde{\mathcal{T}}(Q)(s, a) := r(s, a) + \gamma\widehat{P}_{sa}\widehat{V}_{\mathrm{pe}}^\star - \gamma\tilde{b}(s, a, \widehat{V}_{\mathrm{pe}}^\star)$, because the span term within $\tilde{b}$ would lead to non-monotonicity of $\widetilde{\mathcal{T}}$ and disrupt many other essential properties (like $\gamma$-contractivity). To see the non-monotonicity, suppose some $s'$ has $\widehat{P}(s' \mid s, a) < \frac{\alpha}{n(s,a)}$. Then, for $V \in \mathbb{R}^\mathcal{S}$ where $V(s')$ is the largest entry, ignoring non-differentiability edge cases, we have

$$\frac{d}{dV(s')}\left( \widehat{P}_{sa}V - \alpha\frac{\|V\|_{\mathrm{span}}}{n(s, a)} \right) = \frac{d}{dV(s')}\left( \widehat{P}_{sa}V - \alpha\frac{V(s') - \min_{s'' \in \mathcal{S}} V(s'')}{n(s, a)} \right)$$

$$= \widehat{P}(s' \mid s, a) - \frac{\alpha}{n(s, a)} < 0.$$

However, if we replace $V$ with the quantile-clipped quantity $T_{\alpha/n(s,a)}(\widehat{P}_{s,a}, V)$, then increasing $V(s')$ (when it is the largest entry of $V$) will only increase $T_{\alpha/n(s,a)}(\widehat{P}_{s,a}, V)$ if $\widehat{P}(s' \mid s, a)$ has at least $\alpha/n(s, a)$ probability mass. Hence, by fixing the overpenalization caused by $\|\cdot\|_{\mathrm{span}}$, quantile clipping is essential to define our empirical-span-based pessimistic Bellman operator.

Now we discuss a few other aspects of the proof of Theorem 3.2. Obtaining the Bernstein-style inequality (7) is nontrivial due to statistical dependence between $\widehat{P}_{sa}$ and $\widehat{V}_{\mathrm{pe}}^{\star}$. We remedy this with an argument based on leave-one-out/absorbing MDP techniques [Agarwal et al., 2020], which requires additional covering steps due to the presence of quantile clipping. (See Lemmas B.6 and B.5.)

It is somewhat surprising that Theorem 3.2 is able to obtain a bias-span-based guarantee without requiring any prior bias-span knowledge, since prior work in related uniform coverage settings has shown this is impossible when the effective horizon is large/on the same order as the size of the dataset [Zurek and Chen, 2024]. This is closely related to the issue that the bias span $\|h^{\pi}\|_{\mathrm{span}}$ of a policy $\pi$ is not estimable to multiplicative error with a sample complexity polynomial in only $S$, $A$, and $\|h^{\pi}\|_{\mathrm{span}}$ [Zurek and Chen, 2025b, Tuynman et al., 2024]. However, our proof suggests that $\|h^{\pi}\|_{\mathrm{span}}$ *is* estimable if we allow a dependence on the policy hitting radius $T_{\mathrm{hit}}(P, \pi)$, which we believe is an independently interesting finding. (See Lemma B.18.) This fact plays a key role in bounding the suboptimality in terms of $\|h^{\pi}\|_{\mathrm{span}}$.

### 4.2 Transient lower bound

Next we briefly describe the idea behind the hard instances within Theorem 3.3, which implies that transient coverage is required for offline RL with single-policy complexity parameters. Consider the MDP $P$ in Figure 1, which is parameterized by $m$, which we imagine as arbitrarily large, and $T$, which we imagine as measuring the complexity of $P$. There are two states with two actions each, an absorbing stay action and a leave action which has a small chance of leading to the other state. State 1 has reward 1 for both actions and state 2 has reward 0 for both actions, so clearly the optimal policy $\pi^{\star}$ is to take leave in state 2 and take stay in state 1, and the associated stationary distribution has all its mass on state 1. Also, assuming $m \geq T$, $T_{\mathrm{hit}}(P, \pi^{\star}) = T$, since this is the expected amount of time to hit state 1 starting from state 2. Therefore to satisfy the coverage assumption $n(s, \pi^{\star}(s)) \geq m\mu^{\pi^{\star}}(s) + T_{\mathrm{hit}}(P, \pi^{\star})$, it would suffice to provide $m$ samples for *both* state 1 actions, and $T$ samples for *both* state 2 actions.

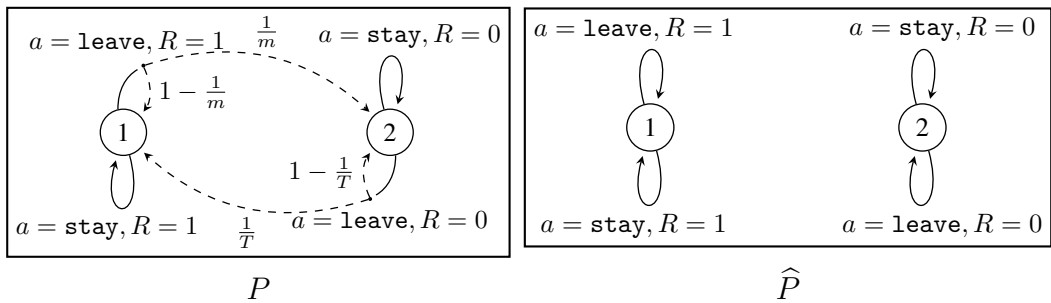

Figure 1: An MDP $P$ parameterized by $m, T$, and an empirical MDP $\widehat{P}$ which has constant probability of being sampled from $P$. Each solid arrow indicates an action and is annotated with its reward. Arrows which split into multiple dashed arrows indicate possible stochastic transitions, and each dashed arrow is annotated with the associated probabilities.

For this sample size function $n$, with constant probability we will not observe any transitions to the other state from either of the leave actions (that is, the samples from each of these state-action pairs would all be of the form $(s, \texttt{leave}, s)$). Under such an event, illustrated by the empirical MDP $\widehat{P}$, no algorithm could distinguish between the leave and stay actions in either state better than random guessing. If an algorithm is forced to return a deterministic policy, then there would be a constant probability of choosing the policy $\pi$ where $(\pi(1), \pi(2)) = (\texttt{leave}, \texttt{stay})$, which will remain in state 2 (and hence have gain $\mathbf{0}$). To generalize to algorithms which may choose randomized policies,

we add more copies of the `stay` action to state 2, so that a "guessed" randomized policy has a low chance of returning to state 1 quickly enough for good performance. Also $P$ is not unichain, but we can add an arbitrarily small ($O(m^{-2})$) probability for the `stay` actions in state 2 to return to state 1, which ensures unichainedness without meaningfully changing the story. We emphasize that the hardness is not due to the inability to identify the `stay` action in state 1, since in general we cannot expect to perfectly match the stationary distribution of the target policy (and in this example, the policy (`leave`, `leave`) still has suboptimality only $O(T/m)$). Rather, the hardness is due to the fact that it is nontrivial to navigate (quickly) back to the target policy's stationary distribution after leaving it, and learning to do so requires data coverage beyond said stationary distribution.

## 5  Conclusion

We developed the first average-reward offline RL algorithms for MDPs where not all policies have constant gain, and also the first convergence rates depending only on the bias span of a single policy. A main limitation of our work is its focus on the tabular setting, hence an important direction is to extend these improvements to function approximation setups to avoid dependence on $S$ in the results. While Theorem 3.3 demonstrates the necessity of data from the target policy from all states, this may be limiting in practice, so an interesting future direction is to explore additional assumptions or information that could be provided to the algorithm to circumvent this requirement.

## Acknowledgments and Disclosure of Funding

Y. Chen and M. Zurek acknowledge support by National Science Foundation grants CCF-2233152 and DMS-2023239.

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

## A  Additional notation and guide to appendices

Let $\pi$ be some stationary policy. Note that $P_\pi$ (defined above as the Markov chain over states induced by policy $\pi$ on the transition kernel $P$) is equal to $M^\pi P$. We also define $r_\pi = M^\pi r$. Then we have $V_\gamma^\pi = (I - \gamma P_\pi)^{-1} r_\pi$. $\|\cdot\|_\infty$ and $\|\cdot\|_1$ denote the usual $\ell_\infty/\ell_1$-norms, respectively. $\|W\|_{\infty\to\infty}$ denotes the $\|\cdot\|_\infty \to \|\cdot\|_\infty$ operator norm of a matrix $W$. In particular $\left\|(I - \gamma P_\pi)^{-1}\right\|_{\infty\to\infty} = \frac{1}{1-\gamma}$. We note that the action maximization operator $M$ and the policy matrix $M^\pi$ both satisfy monotonicity: $V \geq V'$ (elementwise, for $Q, Q' \in \mathbb{R}^{\mathcal{S}\times\mathcal{A}}$) implies $M(Q) \geq M(Q')$, and likewise that $M^\pi Q \geq M^\pi Q'$. These two operators also both satisfy the "constant-shift" property, that for any $c \in \mathbb{R}$ and any $Q \in \mathbb{R}^{\mathcal{S}\times\mathcal{A}}$, we have $M(Q + c\mathbf{1}) = c\mathbf{1} + M(Q)$ and $M^\pi(Q + c\mathbf{1}) = c\mathbf{1} + M^\pi(Q)$. Also we note that $M$ and $M^\pi$ are both 1-Lipschitz with respect to $\|\cdot\|_\infty$, that is $\|MQ - MQ'\|_\infty \leq \|Q - Q'\|_\infty$ and $\|M^\pi Q - M^\pi Q'\|_\infty \leq \|Q - Q'\|_\infty$. For any vector $x$ we let $x^{\circ k}$ denote its elementwise $k$th power. We let $\mathbb{I}$ denote the usual indicator function used in probability where $\mathbb{I}(E)$ is a random variable with value 1 if the event $E$ holds and 0 otherwise.

In Appendix B we prove the main theorem, Theorem 3.2. In Appendix C we prove Theorem 3.3 and in Appendix D we prove Theorem 3.4. Appendix E contains additional supporting results.

## B  Proof of main theorem

### B.1  Well-definedness

We also define a fixed-policy/policy evaluation version of $\widehat{\mathcal{T}}_{\mathrm{pe}}$ which will be useful within the analysis. For any fixed stationary policy $\pi$, we let

$$\widehat{\mathcal{T}}_{\mathrm{pe}}^\pi(Q)(s,a) := r(s,a) + \gamma \max\left\{ \widehat{P}_{sa} T_{\beta(s,a)}(\widehat{P}_{sa}, M^\pi Q) - b(s,a,M^\pi Q), \min_{s'}(M^\pi Q)(s') \right\}. \tag{8}$$

We also define $\widehat{V}_{\mathrm{pe}}^\pi := M^\pi \widehat{Q}_{\mathrm{pe}}^\pi$, where $\widehat{Q}_{\mathrm{pe}}^\pi$ is the unique fixed point of $\widehat{\mathcal{T}}_{\mathrm{pe}}^\pi$ (justified in the below lemma).

The following is a more comprehensive variant of Lemma 3.1.

**Lemma B.1.**      *1.* $\widehat{\mathcal{T}}_{\mathrm{pe}}$ *satisfies the following properties:*

(a) *Monotonicity: If $Q \geq Q'$ then $\widehat{\mathcal{T}}_{\mathrm{pe}}(Q) \geq \widehat{\mathcal{T}}_{\mathrm{pe}}(Q')$.*

(b) *Constant shift: For any $c \in \mathbb{R}$, $\widehat{\mathcal{T}}_{\mathrm{pe}}(Q + c\mathbf{1}) = \widehat{\mathcal{T}}_{\mathrm{pe}}(Q) + \gamma c\mathbf{1}$.*

(c) *$\gamma$-contractivity: $\widehat{\mathcal{T}}_{\mathrm{pe}}$ is a $\gamma$-contraction and has a unique fixed point $\widehat{Q}_{\mathrm{pe}}^\star$.*

(d) *Boundedness: $\mathbf{0} \leq \widehat{Q}_{\mathrm{pe}}^\star \leq \frac{1}{1-\gamma}\mathbf{1}$.*

*2. For any fixed stationary deterministic policy $\pi$, the analogous statements hold for $\widehat{\mathcal{T}}_{\mathrm{pe}}^\pi$:*

(a) *Monotonicity: If $Q \geq Q'$ then $\widehat{\mathcal{T}}_{\mathrm{pe}}^\pi(Q) \geq \widehat{\mathcal{T}}_{\mathrm{pe}}^\pi(Q')$.*

(b) *Constant shift: For any $c \in \mathbb{R}$, $\widehat{\mathcal{T}}_{\mathrm{pe}}^\pi(Q + c\mathbf{1}) = \widehat{\mathcal{T}}_{\mathrm{pe}}^\pi(Q) + \gamma c\mathbf{1}$.*

(c) *$\gamma$-contractivity: $\widehat{\mathcal{T}}_{\mathrm{pe}}^\pi$ is a $\gamma$-contraction and has a unique fixed point $\widehat{Q}_{\mathrm{pe}}^\pi$.*

(d) *Boundedness: $\mathbf{0} \leq \widehat{Q}_{\mathrm{pe}}^\pi \leq \frac{1}{1-\gamma}\mathbf{1}$.*

*3. For any fixed stationary deterministic policy $\pi$, we have $\widehat{Q}_{\mathrm{pe}}^\star \geq \widehat{Q}_{\mathrm{pe}}^\pi$.*

*Proof.* We note that a few steps are similar to Li et al. [2023, Lemma 1], but our new choice of penalty requires much more involved analysis.

We define an auxiliary operator $\overline{\mathcal{T}}_{\mathrm{pe}} : \mathbb{R}^{\mathcal{S}} \to \mathbb{R}^{\mathcal{S}\mathcal{A}}$ by, for any $V \in \mathbb{R}^{\mathcal{S}}$,

$$\overline{\mathcal{T}}_{\mathrm{pe}}(V)(s,a) := r(s,a) + \gamma \max\left\{ \widehat{P}_{sa} T_{\beta(s,a)}(\widehat{P}_{sa}, V) - b(s,a,V), \min_{s'}(V)(s') \right\}.$$

We defer the verification of the following fact, which involves somewhat lengthy calculations, to Appendix E.1.

**Lemma B.2.** *Let $V, V' \in \mathbb{R}^{\mathcal{S}}$ be arbitrary and suppose that $V \geq V'$. Then (elementwise)*

$$\overline{\mathcal{T}}_{\mathrm{pe}}(V) \geq \overline{\mathcal{T}}_{\mathrm{pe}}(V').$$

Given Lemma B.2, we can relatively easily verify Lemma B.1. We note that Lemma B.2 makes use of the quantile clipping in an essential way.

Now we will show item 1 except for the boundedness property. Notice that $\widehat{\mathcal{T}}_{\mathrm{pe}}(Q) = \overline{\mathcal{T}}_{\mathrm{pe}}(MQ)$ (for any $Q \in \mathbb{R}^{\mathcal{SA}}$). Therefore letting $Q, Q' \in \mathbb{R}^{\mathcal{SA}}$ with $Q \geq Q'$, we have by monotonicity of $M$ that $MQ \geq MQ'$, and thus by monotonicity of $\overline{\mathcal{T}}_{\mathrm{pe}}$ we conclude that

$$\widehat{\mathcal{T}}_{\mathrm{pe}}(Q) = \overline{\mathcal{T}}_{\mathrm{pe}}(MQ) \geq \overline{\mathcal{T}}_{\mathrm{pe}}(MQ') = \widehat{\mathcal{T}}_{\mathrm{pe}}(Q')$$

as desired. Next we check the constant shift property of $\overline{\mathcal{T}}_{\mathrm{pe}}$. Fix $c \in \mathbb{R}, V \in \mathbb{R}^{\mathcal{S}}$, and $s \in \mathcal{S}, a \in \mathcal{A}$. Then we have that $T_{\beta(s,a)}(\widehat{P}_{sa}, V + c\mathbf{1}) = T_{\beta(s,a)}(\widehat{P}_{sa}, V) + c\mathbf{1}$, regardless of whether $\beta(s,a) \in [0,1]$ or $\beta(s,a) > 1$, since when $\beta(s,a) > 1$ we have $T_{\beta(s,a)}(\widehat{P}_{sa}, V + c\mathbf{1}) = \min_{s \in \mathcal{S}}(V + c\mathbf{1})\mathbf{1} = (\min_{s \in \mathcal{S}}(V) + c)\mathbf{1}$, and when $\beta(s,a) \leq 1$, by (4) we have

$$T_{\beta(s,a)}(\widehat{P}_{sa}, V + c\mathbf{1})(s) = \min\left\{V(s) + c, \sup\left\{V(s') + c : s' \in \mathcal{S}, \sum_{s'' \in \mathcal{S}: V(s'') + c \geq V(s') + c} \widehat{P}_{sa}(s') \geq \beta\right\}\right\}$$

$$= c + \min\left\{V(s), \sup\left\{V(s') : s' \in \mathcal{S}, \sum_{s'' \in \mathcal{S}: V(s'') \geq V(s')} \widehat{P}_{sa}(s') \geq \beta\right\}\right\}$$

$$= c + T_{\beta(s,a)}(\widehat{P}_{sa}, V)(s).$$

Therefore

$$\mathbb{V}_{\widehat{P}_{sa}}\left[T_{\beta(s,a)}(\widehat{P}_{sa}, V + c\mathbf{1})\right] = \mathbb{V}_{\widehat{P}_{sa}}\left[T_{\beta(s,a)}(\widehat{P}_{sa}, V) + c\mathbf{1}\right] = \mathbb{V}_{\widehat{P}_{sa}}\left[T_{\beta(s,a)}(\widehat{P}_{sa}, V)\right]$$

and $\left\|T_{\beta(s,a)}(\widehat{P}_{sa}, V + c\mathbf{1})\right\|_{\mathrm{span}} = \left\|T_{\beta(s,a)}(\widehat{P}_{sa}, V) + c\mathbf{1}\right\|_{\mathrm{span}} = \left\|T_{\beta(s,a)}(\widehat{P}_{sa}, V)\right\|_{\mathrm{span}}$

and therefore we have that $b(s, a, V) = b(s, a, V + c\mathbf{1})$. Additionally we have that

$$\min_{s'}(V + c\mathbf{1})(s') = \min_{s'} V(s') + c.$$

Hence

$$\overline{\mathcal{T}}_{\mathrm{pe}}(V + c\mathbf{1})(s, a) = r(s, a) + \gamma \max\left\{\widehat{P}_{sa} T_{\beta(s,a)}(\widehat{P}_{sa}, V + c\mathbf{1}) - b(s, a, V + c\mathbf{1}), \min_{s'}(V + c\mathbf{1})(s')\right\}$$

$$= r(s, a) + \gamma \max\left\{\widehat{P}_{sa} T_{\beta(s,a)}(\widehat{P}_{sa}, V) + c\widehat{P}_{sa}\mathbf{1} - b(s, a, V), \min_{s'}(V)(s') + c\right\}$$

$$= r(s, a) + \gamma c + \gamma \max\left\{\widehat{P}_{sa} T_{\beta(s,a)}(\widehat{P}_{sa}, V) - b(s, a, V), \min_{s'}(V)(s')\right\}$$

$$= \gamma c + \overline{\mathcal{T}}_{\mathrm{pe}}(V)(s, a) \qquad (9)$$

(since $\widehat{P}_{sa}\mathbf{1} = 1$). Using (9) and the fact that $M(Q + c\mathbf{1}) = MQ + c\mathbf{1}$ we can show that $\widehat{\mathcal{T}}_{\mathrm{pe}}$ satisfies the constant shift property as well:

$$\widehat{\mathcal{T}}_{\mathrm{pe}}(Q + c\mathbf{1}) = \overline{\mathcal{T}}_{\mathrm{pe}}(M(Q + c\mathbf{1})) = \overline{\mathcal{T}}_{\mathrm{pe}}(MQ + c\mathbf{1}) = \overline{\mathcal{T}}_{\mathrm{pe}}(MQ) + \gamma c\mathbf{1} = \widehat{\mathcal{T}}_{\mathrm{pe}}(Q) + \gamma c\mathbf{1}$$

as desired. Finally we can check contractivity of $\widehat{\mathcal{T}}_{\mathrm{pe}}$. We note that it suffices to show that $\overline{\mathcal{T}}_{\mathrm{pe}}$ is $\gamma$-Lipschitz, since then we would have for any $Q_1, Q_2 \in \mathbb{R}^{\mathcal{SA}}$ that

$$\left\|\widehat{\mathcal{T}}_{\mathrm{pe}}(Q_1) - \widehat{\mathcal{T}}_{\mathrm{pe}}(Q_2)\right\|_{\infty} = \left\|\overline{\mathcal{T}}_{\mathrm{pe}}(MQ_1) - \overline{\mathcal{T}}_{\mathrm{pe}}(MQ_2)\right\|_{\infty} \leq \gamma \left\|MQ_1 - MQ_2\right\|_{\infty} \leq \gamma \left\|Q_1 - Q_2\right\|_{\infty}$$

as desired, where the first inequality is due to the (assumed) Lipschitzness of $\overline{\mathcal{T}}_{\mathrm{pe}}$ and the second inequality is due to the 1-Lipschitzness of $M$. Now we verify that $\overline{\mathcal{T}}_{\mathrm{pe}}$ is indeed $\gamma$-Lipschitz. For any $V_1, V_2 \in \mathbb{R}^{\mathcal{S}}$ we have $V_1 \leq V_2 + \left\|V_1 - V_2\right\|_{\infty} \mathbf{1}$ (elementwise), so by monotonicity of $\overline{\mathcal{T}}_{\mathrm{pe}}$ (Lemma

B.2), and then using the fact that $\overline{\mathcal{T}}_{\mathrm{pe}}$ satisfies the constant shift property (shown in (9)) in the next inequality, we have

$$\overline{\mathcal{T}}_{\mathrm{pe}}(V_1) \leq \overline{\mathcal{T}}_{\mathrm{pe}}(V_2 + \|V_1 - V_2\|_\infty \mathbf{1})$$
$$= \overline{\mathcal{T}}_{\mathrm{pe}}(V_2) + \gamma \|V_1 - V_2\|_\infty \mathbf{1}$$

so by rearranging

$$\overline{\mathcal{T}}_{\mathrm{pe}}(V_1) - \overline{\mathcal{T}}_{\mathrm{pe}}(V_2) \leq \gamma \|V_1 - V_2\|_\infty \mathbf{1}.$$

By reversing the roles of $V_1$ and $V_2$ we also have

$$\overline{\mathcal{T}}_{\mathrm{pe}}(V_2) - \overline{\mathcal{T}}_{\mathrm{pe}}(V_1) \leq \gamma \|V_1 - V_2\|_\infty \mathbf{1}$$

or equivalently

$$-\gamma \|V_1 - V_2\|_\infty \mathbf{1} \leq \overline{\mathcal{T}}_{\mathrm{pe}}(V_1) - \overline{\mathcal{T}}_{\mathrm{pe}}(V_2).$$

Combining these two inequalities involving $\overline{\mathcal{T}}_{\mathrm{pe}}(V_2) - \overline{\mathcal{T}}_{\mathrm{pe}}(V_1)$ we conclude that $\left\|\overline{\mathcal{T}}_{\mathrm{pe}}(V_1) - \overline{\mathcal{T}}_{\mathrm{pe}}(V_2)\right\|_\infty \leq \gamma \|V_1 - V_2\|_\infty$ as desired and thus $\widehat{\mathcal{T}}_{\mathrm{pe}}$ is a $\gamma$-contraction. By the Banach fixed-point theorem (e.g. [Pugh, 2015, Chapter 4.5]) this implies the existence of a unique fixed point of $\widehat{\mathcal{T}}_{\mathrm{pe}}$, which we call $\widehat{Q}^\star_{\mathrm{pe}}$. (We check that $\mathbf{0} \leq \widehat{Q}^\star_{\mathrm{pe}} \leq \frac{1}{1-\gamma}\mathbf{1}$ later.)

Now we will show item 2 except for the boundedness property. Notice that similarly to the previous case, $\widehat{\mathcal{T}}^\pi_{\mathrm{pe}}(Q) = \overline{\mathcal{T}}_{\mathrm{pe}}(M^\pi Q)$ (for any $Q \in \mathbb{R}^{\mathcal{S}\mathcal{A}}$). The only properties of $M$ used in the proofs for the previous case were monotonicity (that $Q \geq Q' \implies MQ \geq MQ'$), that $M(Q + c\mathbf{1}) = MQ + c\mathbf{1}$, and that $M$ is 1-Lipschitz. All of these properties are also true with $M^\pi$ in place of $M$, so in fact all proofs used to verify item 1 can immediately be applied (with this minor modification) to also verify item 2.

Next, item 3 would follow by showing that for any fixed $Q \in \mathbb{R}^{SA}$ we have

$$\widehat{\mathcal{T}}_{\mathrm{pe}}(Q) \geq \widehat{\mathcal{T}}^\pi_{\mathrm{pe}}(Q) \tag{10}$$

since then by a standard argument we can show for any integer $k \geq 0$ that

$$\left(\widehat{\mathcal{T}}_{\mathrm{pe}}\right)^{(k)}(\mathbf{0}) \geq \left(\widehat{\mathcal{T}}^\pi_{\mathrm{pe}}\right)^{(k)}(\mathbf{0})$$

(where $(k)$ denotes $k$ compositions of an operator) and therefore that

$$\widehat{Q}^\star_{\mathrm{pe}} = \lim_{k \to \infty} \left(\widehat{\mathcal{T}}_{\mathrm{pe}}\right)^{(k)}(\mathbf{0}) \geq \lim_{k \to \infty} \left(\widehat{\mathcal{T}}^\pi_{\mathrm{pe}}\right)^{(k)}(\mathbf{0}) = \widehat{Q}^\pi_{\mathrm{pe}}.$$

So now we focus on showing (10), but this follows immediately from the fact that $MQ \geq M^\pi Q$ and that $\overline{\mathcal{T}}_{\mathrm{pe}}$ is monotone (Lemma B.2), since we have

$$\widehat{\mathcal{T}}_{\mathrm{pe}}(Q) = \overline{\mathcal{T}}_{\mathrm{pe}}(MQ) \geq \overline{\mathcal{T}}_{\mathrm{pe}}(M^\pi Q) = \widehat{\mathcal{T}}^\pi_{\mathrm{pe}}(Q).$$

[Matthew: should be $\beta(s,a)$ not $\beta$ in the below paragraph?] [Guy: yes] Finally, we check both boundedness properties. Since we already have that $\widehat{Q}^\pi_{\mathrm{pe}} \leq \widehat{Q}^\star_{\mathrm{pe}}$, it suffices to show that $\mathbf{0} \leq \widehat{Q}^\pi_{\mathrm{pe}}$ and that $\widehat{Q}^\star_{\mathrm{pe}} \leq \frac{1}{1-\gamma}\mathbf{1}$. First, note that we have $\widehat{\mathcal{T}}^\pi_{\mathrm{pe}}(\mathbf{0}) \geq \mathbf{0}$, since for any $s \in \mathcal{S}, a \in \mathcal{A}$,

$$\widehat{\mathcal{T}}^\pi_{\mathrm{pe}}(\mathbf{0})(s,a) = r(s,a) + \gamma \max\left\{\widehat{P}_{sa}T_{\beta(s,a)}(\widehat{P}_{sa}, M^\pi\mathbf{0}) - b(s,a,M^\pi\mathbf{0}), \min_{s'}(M^\pi\mathbf{0})(s')\right\}$$
$$\geq r(s,a) + \gamma \min_{s'}(M^\pi\mathbf{0})(s') = r(s,a) \geq 0.$$

Then by monotonicity of $\widehat{\mathcal{T}}^\pi_{\mathrm{pe}}$ we have for any integer $k \geq 0$ that

$$\left(\widehat{\mathcal{T}}^\pi_{\mathrm{pe}}\right)^{(k)}(\mathbf{0}) \geq \left(\widehat{\mathcal{T}}^\pi_{\mathrm{pe}}\right)^{(k-1)}(\mathbf{0}) \geq \cdots \geq \mathbf{0}$$

and so

$$\widehat{Q}^\pi_{\mathrm{pe}} = \lim_{k \to \infty} \left(\widehat{\mathcal{T}}^\pi_{\mathrm{pe}}\right)^{(k)}(\mathbf{0}) \geq \mathbf{0}$$

as desired. Similarly, we have that $\widehat{\mathcal{T}}_{\mathrm{pe}}(\mathbf{1}/(1-\gamma)) \le \mathbf{1}/(1-\gamma)$, since for any $s \in \mathcal{S}, a \in \mathcal{A}$,

$$\widehat{\mathcal{T}}_{\mathrm{pe}}(\mathbf{1}/(1-\gamma))(s,a)$$

$$= r(s,a) + \gamma \max \left\{ \widehat{P}_{sa} T_{\beta(s,a)}(\widehat{P}_{sa}, M\mathbf{1}/(1-\gamma)) - b(s,a, M\mathbf{1}/(1-\gamma)), \min_{s'}(M\mathbf{1}/(1-\gamma))(s') \right\}$$

$$\le 1 + \gamma \frac{1}{1-\gamma} = \frac{1}{1-\gamma}.$$

By an analogous argument to the previous bound, we have from monotonicity of $\widehat{\mathcal{T}}_{\mathrm{pe}}$ that $\left(\widehat{\mathcal{T}}_{\mathrm{pe}}\right)^{(k)}(\mathbf{1}/(1-\gamma)) \le \mathbf{1}/(1-\gamma)$ for all positive integers $k$ and thus that $\widehat{Q}_{\mathrm{pe}}^{\star} \le \mathbf{1}/(1-\gamma)$. $\quad\square$

In the above proof we defined the operator $\overline{\mathcal{T}}_{\mathrm{pe}}$ and verified its Lipshitzness, which we state in the following lemma as $\overline{\mathcal{T}}_{\mathrm{pe}}$ will appear again later.

**Lemma B.3.** $\overline{\mathcal{T}}_{\mathrm{pe}}$ *is $\gamma$-Lipschitz.*

### B.2 Optimization

In this subsection we establish the basic properties of the outputs of Algorithm 1.

**Lemma B.4.** *Algorithm 1 returns $\widehat{Q}$ such that*

$$\widehat{Q} \le \widehat{Q}_{\mathrm{pe}}^{\star} \le \widehat{Q} + \frac{1}{2n_{\mathrm{tot}}}\mathbf{1} \quad and \quad \widehat{\mathcal{T}}_{\mathrm{pe}}(\widehat{Q}) \ge \widehat{Q}.$$

*Proof.* First we note that $\widehat{\mathcal{T}}_{\mathrm{pe}}(\mathbf{0}) \ge \mathbf{0}$, which follows easily from the definition (3) since (for arbitrary $s \in \mathcal{S}, a \in \mathcal{A}$)

$$\widehat{\mathcal{T}}_{\mathrm{pe}}(\mathbf{0})(s,a) = r(s,a) + \gamma \max \left\{ \widehat{P}_{sa} T_{\beta(s,a)}(\widehat{P}_{sa}, M\mathbf{0}) - b(s,a, M\mathbf{0}), \min_{s'}(M\mathbf{0})(s') \right\}$$

$$\ge r(s,a) + \gamma \min_{s'}(M\mathbf{0})(s') = r(s,a) \ge 0.$$

$\widehat{\mathcal{T}}_{\mathrm{pe}}(\widehat{Q}) \ge \widehat{Q}$ follows from this fact and monotonicity of $\widehat{\mathcal{T}}_{\mathrm{pe}}$ by standard arguments, since if for any $t \in \mathbb{N}$ we have that $\widehat{\mathcal{T}}_{\mathrm{pe}}(\widehat{Q}_t) \ge \widehat{Q}_t$ then

$$\widehat{\mathcal{T}}_{\mathrm{pe}}(\widehat{Q}_{t+1}) = \widehat{\mathcal{T}}_{\mathrm{pe}}\left(\widehat{\mathcal{T}}_{\mathrm{pe}}(\widehat{Q}_t)\right) \ge \widehat{\mathcal{T}}_{\mathrm{pe}}\left(\widehat{Q}_t\right)$$

so by induction (since $\widehat{Q}_0 = \mathbf{0}$) $\widehat{\mathcal{T}}_{\mathrm{pe}}(\widehat{Q}_t) \ge \widehat{Q}_t$ holds for $t = K$, and we have $\widehat{Q}_K = \widehat{Q}$ by definition.

Now we argue that $\widehat{Q} \le \widehat{Q}_{\mathrm{pe}}^{\star}$, which follows from $\widehat{\mathcal{T}}_{\mathrm{pe}}(\widehat{Q}) \ge \widehat{Q}$ and monotonicity of $\widehat{\mathcal{T}}_{\mathrm{pe}}$ by standard arguments, since assuming for some $t \ge 1$ that $\widehat{\mathcal{T}}_{\mathrm{pe}}^{(t)}(\widehat{Q}) \ge \widehat{Q}$, then we have by monotonicity that

$$\left(\widehat{\mathcal{T}}_{\mathrm{pe}}\right)^{(t+1)}(\widehat{Q}) = \left(\widehat{\mathcal{T}}_{\mathrm{pe}}\right)\left(\widehat{\mathcal{T}}_{\mathrm{pe}}^{(t)}(\widehat{Q})\right) \ge \widehat{\mathcal{T}}_{\mathrm{pe}}(\widehat{Q}) \ge \widehat{Q}$$

and so by induction $\left(\widehat{\mathcal{T}}_{\mathrm{pe}}\right)^{(t)}(\widehat{Q}) \ge \widehat{Q}$ for all $t \ge 1$, and thus

$$\widehat{Q}_{\mathrm{pe}}^{\star} = \lim_{t \to \infty} \left(\widehat{\mathcal{T}}_{\mathrm{pe}}\right)^{(t)}(\widehat{Q}) \ge \lim_{t \to \infty} \widehat{Q} = \widehat{Q}$$

as desired.

Finally we check that $\widehat{Q}_{\mathrm{pe}}^{\star} \le \widehat{Q} + \frac{1}{2n_{\mathrm{tot}}}\mathbf{1}$. Again note that $\widehat{Q} = \widehat{Q}_K$. By the definition of $K = \left\lceil \frac{\log\left(\frac{2n_{\mathrm{tot}}}{1-\gamma}\right)}{1-\gamma} \right\rceil$, as well as the fact that $\log(1/\gamma) \ge 1 - \gamma$ for any $\gamma$, we have

$$\gamma^K = e^{K\log(\gamma)} \le e^{\frac{\log\left(\frac{2n_{\mathrm{tot}}}{1-\gamma}\right)}{1-\gamma}\log(\gamma)} = e^{\frac{\log\left(\frac{1-\gamma}{2n_{\mathrm{tot}}}\right)}{1-\gamma}\log(1/\gamma)} \le e^{\log\left(\frac{1-\gamma}{2n_{\mathrm{tot}}}\right)} = \frac{1-\gamma}{2n_{\mathrm{tot}}}.$$

Using this bound, $\gamma$-contractivity, and the fact that $\mathbf{0} \le \widehat{Q}_{\mathrm{pe}}^{\star} \le \frac{1}{1-\gamma}\mathbf{1}$ from Lemma B.1, we have

$$\left\|\widehat{Q}_K - \widehat{Q}_{\mathrm{pe}}^{\star}\right\|_{\infty} \le \gamma^K \left\|\widehat{Q}_0 - \widehat{Q}_{\mathrm{pe}}^{\star}\right\|_{\infty} = \gamma^K \left\|\mathbf{0} - \widehat{Q}_{\mathrm{pe}}^{\star}\right\|_{\infty} \le \gamma^K \frac{1}{1-\gamma} \le \frac{1}{2n_{\mathrm{tot}}}$$

which implies $\widehat{Q}_{\mathrm{pe}}^{\star} \le \widehat{Q} + \frac{1}{2n_{\mathrm{tot}}}\mathbf{1}$. $\quad\square$

### B.3 Concentration

In this subsection we establish the key concentration inequalities, given in Lemmas B.7 and B.8, using leave-one-out techniques. We start with two helper lemmas which abstractly handle the leave-one-out-based covering steps before proving Lemmas B.7 and B.8.

**Lemma B.5.** *Fix some $\delta' > 0$ and some $s \in \mathcal{S}, a \in \mathcal{A}$. Suppose that for some random vector $X \in \mathbb{R}^{\mathcal{S}}$, there exists a (deterministic) set $U$ and some random variables $X_u \in \mathbb{R}^{\mathcal{S}}$ for each $u$ (that is, for each $u \in U$, $X_u$ is a random vector in $\mathbb{R}^{\mathcal{S}}$) such that*

1. *For all $u \in U$, $X_u$ is independent of all samples $S_{sa}^1, \ldots, S_{sa}^{n(s,a)}$ drawn from $P(\cdot \mid s, a)$.*

2. *Almost surely there exists some $u^\star \in U$ such that $\|X - X_{u^\star}\|_\infty \le \frac{1}{n_{\text{tot}}}$.*

*Also assume $n(s, a) \ge 2$. Then with probability at least $1 - 6\delta'$, we have that*

$$\left| (\widehat{P}_{sa} - P_{sa})X \right| \le \|X\|_{\text{span}} \sqrt{\frac{\log |U|/\delta'}{2n(s,a)}} + \frac{2}{n_{\text{tot}}} \left( 1 + \sqrt{\frac{\log |U|/\delta'}{2n(s,a)}} \right) \tag{11}$$

$$\left| (\widehat{P}_{sa} - P_{sa})X \right| \le \sqrt{\frac{2\mathbb{V}_{P_{sa}}[X] \log(|U|/\delta')}{n(s,a)}} + \|X\|_{\text{span}} \frac{\log(|U|/\delta')}{3n(s,a)}$$
$$+ \frac{1}{n_{\text{tot}}} \left( 2 + \sqrt{\frac{2\log(|U|/\delta')}{n(s,a)}} + 2\frac{\log(|U|/\delta')}{3n(s,a)} \right) \tag{12}$$

$$\left| \sqrt{\frac{n(s,a)}{n(s,a)-1}}\mathbb{V}_{\widehat{P}_{sa}}[X] - \sqrt{\mathbb{V}_{P_{sa}}[X]} \right| \le \|X\|_{\text{span}} \sqrt{\frac{2\log |U|/\delta'}{n(s,a)-1}} + \frac{1}{n_{\text{tot}}} \left( 2\sqrt{\frac{2\log |U|/\delta'}{n(s,a)-1}} + 3 \right) \tag{13}$$

$$\left| (\widehat{P}_{sa} - P_{sa})X \right| \le \sqrt{\frac{2\mathbb{V}_{\widehat{P}_{sa}}[X] \log(|U|/\delta')}{n(s,a)-1}} + \|X\|_{\text{span}} \frac{7}{3}\frac{\log(|U|/\delta')}{n(s,a)-1}$$
$$+ \frac{1}{n_{\text{tot}}} \left( 2 + 3\sqrt{\frac{2\log(|U|/\delta')}{n(s,a)-1}} + \frac{14}{3}\frac{\log(|U|/\delta')}{n(s,a)-1} \right) \tag{14}$$

*Proof.* We start by showing that

$$\left| (\widehat{P}_{sa} - P_{sa})X \right| \le \left| (\widehat{P}_{sa} - P_{sa})X_{u^\star} \right| + \left| (\widehat{P}_{sa} - P_{sa})(X - X_{u^\star}) \right|$$
$$\le \left| (\widehat{P}_{sa} - P_{sa})X_{u^\star} \right| + \left\| \widehat{P}_{sa} - P_{sa} \right\|_1 \|X - X_{u^\star}\|_\infty$$
$$\le \left| (\widehat{P}_{sa} - P_{sa})X_{u^\star} \right| + \frac{2}{n_{\text{tot}}} \tag{15}$$

where the final inequality is because $\left\| \widehat{P}_{sa} - P_{sa} \right\|_1 \le 2$ and $\|X - X_{u^\star}\|_\infty \le \frac{1}{n_{\text{tot}}}$.

Then since for any fixed $u \in U$ we have $(\widehat{P}_{sa} - P_{sa})X_u = \sum_{i=1}^{n(s,a)}(X_u(S_{sa}^i) - P_{sa}X_u)$, by Hoeffding's inequality conditioned on $X_u$ (since by assumption $X_u$ is independent from the $S_{sa}^i$ and each term in the above sum is contained within the interval $[\min X_u, \max X_u]$ which has length $\|X_u\|_{\text{span}}$) we have that

$$\mathbb{P}\left( \left| \sum_{i=1}^{n(s,a)}(X_u(S_{sa}^i) - P_{sa}X_u) \right| \ge \|X_u\|_{\text{span}} \sqrt{\frac{\log |U|/\delta'}{2n(s,a)}} \;\middle|\; X_u \right) \le \frac{2\delta'}{|U|}$$

and so

$$\mathbb{P}\left(\left|\sum_{i=1}^{n(s,a)}\left(X_u(S_{sa}^i)-P_{sa}X_u\right)\right|\geq\|X_u\|_{\mathrm{span}}\sqrt{\frac{\log|U|/\delta'}{2n(s,a)}}\right)\leq\mathbb{E}\frac{2\delta'}{|U|}=\frac{2\delta'}{|U|}.$$

Taking a union bound, the above inequality holds for all $u\in U$ with probability at least $1-2\delta'$. Finally, since

$$\|X_{u^\star}\|_{\mathrm{span}}\leq\|X\|_{\mathrm{span}}+\|X_{u^\star}-X\|_{\mathrm{span}}\leq\|X\|_{\mathrm{span}}+2\|X_{u^\star}-X\|_\infty\leq\|X\|_{\mathrm{span}}+\frac{2}{n_{\mathrm{tot}}},\quad(16)$$

combining with (15) we have that

$$\left|(\widehat{P}_{sa}-P_{sa})X\right|\leq\frac{2}{n_{\mathrm{tot}}}+\|X_{u^\star}\|_{\mathrm{span}}\sqrt{\frac{\log|U|/\delta'}{2n(s,a)}}\leq\frac{2}{n_{\mathrm{tot}}}+\|X\|_{\mathrm{span}}\sqrt{\frac{\log|U|/\delta'}{2n(s,a)}}+\frac{2}{n_{\mathrm{tot}}}\sqrt{\frac{\log|U|/\delta'}{2n(s,a)}}.$$

Next we would like to apply the concentration inequalities of Maurer and Pontil [2009]. To apply their theorems as stated, we must shift and normalize to define (for each $u\in U$)

$$X'_u:=\frac{X_u-\min_{x\in\mathcal{S}}X_u(x)}{\|X_u\|_{\mathrm{span}}}$$

so that $X'_u\in[0,1]$ almost surely. Fixing some $u\in U$ and applying Maurer and Pontil [2009, Theorem 10], assuming $n(s,a)\geq2$, we have with probability at least $1-2\delta'/|U|$ that

$$\left|\sqrt{\frac{n(s,a)}{n(s,a)-1}\mathbb{V}_{\widehat{P}_{sa}}\left[X'_u\right]}-\sqrt{\mathbb{V}_{P_{sa}}\left[X'_u\right]}\right|\leq\sqrt{\frac{2\ln|U|/\delta'}{n(s,a)-1}}$$

using the facts that by standard calculations, abbreviating $\tilde{n}=n(s,a)$ for convenience,

$$\mathbb{E}\left[\frac{1}{2\tilde{n}(\tilde{n}-1)}\sum_{i=1}^{\tilde{n}}\sum_{j=1}^{\tilde{n}}\left(X'_u(S_{sa}^i)-X'_u(S_{sa}^j)\right)^2\right]=\frac{\tilde{n}}{2\tilde{n}(\tilde{n}-1)}0+\frac{\tilde{n}(\tilde{n}-1)}{2\tilde{n}(\tilde{n}-1)}\mathbb{E}\left[\left(X'_u(S_{sa}^1)-X'_u(S_{sa}^2)\right)^2\right]$$

$$=\frac{1}{2}\left(2\mathbb{E}\left[\left(X'_u(S_{sa}^1)\right)^2\right]-2\mathbb{E}\left[X'_u(S_{sa}^1)\right]^2\right)$$

$$=\mathbb{V}\left[X'_u(S_{sa}^1)\right]=\mathbb{V}_{P_{sa}}\left[X'_u\right]$$

and

$$\frac{1}{2\tilde{n}(\tilde{n}-1)}\sum_{i=1}^{\tilde{n}}\sum_{j=1}^{\tilde{n}}\left(\widehat{V}_{\mathrm{pe}}^{s,u,s'}(S_{sa}^i)-\widehat{V}_{\mathrm{pe}}^{s,u,s'}(S_{sa}^j)\right)^2=\frac{\tilde{n}}{\tilde{n}-1}\widehat{P}_{sa}\left(X'_u-\widehat{P}_{sa}X'_u\mathbf{1}\right)^{\circ 2}$$

$$=\frac{\tilde{n}}{\tilde{n}-1}\mathbb{V}_{\widehat{P}_{sa}}\left[X'_u\right]$$

(since Maurer and Pontil [2009, Theorem 10] as stated involves the quantity $\frac{1}{2\tilde{n}(\tilde{n}-1)}\sum_{i=1}^{\tilde{n}}\sum_{j=1}^{\tilde{n}}\left(X'_u(S_{sa}^i)-X'_u(S_{sa}^j)\right)^2$ and its expectation). Taking a union bound and undoing the normalization and shifting, we have for all $u\in U$ that

$$\left|\sqrt{\frac{n(s,a)}{n(s,a)-1}\mathbb{V}_{\widehat{P}_{sa}}\left[X_u\right]}-\sqrt{\mathbb{V}_{P_{sa}}\left[X_u\right]}\right|\leq\|X_u\|_{\mathrm{span}}\sqrt{\frac{2\log|U|/\delta'}{n(s,a)-1}}\qquad(17)$$

with probability at least $1-2\delta'$. For any arbitrary probability distribution $\mu\in\mathbb{R}^{\mathcal{S}}$ we have that

$$\left|\sqrt{\mathbb{V}_\mu\left[X\right]}-\sqrt{\mathbb{V}_\mu\left[X_{u^\star}\right]}\right|\leq\frac{1}{n_{\mathrm{tot}}}\qquad(18)$$

since

$$\sqrt{\mathbb{V}_\mu\left[X\right]}=\sqrt{\mathbb{V}_\mu\left[X_{u^\star}+(X-X_{u^\star})\right]}\leq\sqrt{\mathbb{V}_\mu\left[X_{u^\star}\right]}+\sqrt{\mathbb{V}_\mu\left[X-X_{u^\star}\right]}$$

(where the inequality step follows from triangle inequality since $Y \mapsto \sqrt{\mathbb{E}Y^2}$ is a norm on random variables $Y$) and then we have

$$\sqrt{\mathbb{V}_\mu \left[X - X_{u^\star}\right]} \leq \|X - X_{u^\star}\|_\infty \leq \frac{1}{n_{\text{tot}}}.$$

Thus combining (17) with (18) we conclude that

$$\left| \sqrt{\frac{n(s,a)}{n(s,a)-1} \mathbb{V}_{\widehat{P}_{sa}}[X]} - \sqrt{\mathbb{V}_{P_{sa}}[X]} \right| \tag{19}$$

$$\leq \left| \sqrt{\frac{n(s,a)}{n(s,a)-1} \mathbb{V}_{\widehat{P}_{sa}}[X_{u^\star}]} - \sqrt{\mathbb{V}_{P_{sa}}[X_{u^\star}]} \right| + \sqrt{\frac{n(s,a)}{n(s,a)-1} \frac{1}{n_{\text{tot}}}} + \frac{1}{n_{\text{tot}}}$$

$$\leq \|X_{u^\star}\|_{\text{span}} \sqrt{\frac{2\log|U|/\delta'}{n(s,a)-1}} + \sqrt{\frac{n(s,a)}{n(s,a)-1} \frac{1}{n_{\text{tot}}}} + \frac{1}{n_{\text{tot}}}$$

$$\leq \|X\|_{\text{span}} \sqrt{\frac{2\log|U|/\delta'}{n(s,a)-1}} + \frac{2}{n_{\text{tot}}} \sqrt{\frac{2\log|U|/\delta'}{n(s,a)-1}} + \sqrt{\frac{n(s,a)}{n(s,a)-1} \frac{1}{n_{\text{tot}}}} + \frac{1}{n_{\text{tot}}} \tag{20}$$

using (16) again in the final inequality. To obtain the slightly simplified bound (13) we use that by assumption $n(s,a) \geq 2$, so $\sqrt{\frac{n(s,a)}{n(s,a)-1}} \leq \sqrt{2} \leq 2$.

Now, similarly to our use of Hoeffding's inequality, using Bernstein's inequality (e.g., Maurer and Pontil [2009, Theorem 3]), as well as a union bound over all $u \in U$, we have that with probability at least $1 - 2\delta'$, for all $u \in U$,

$$\left| (\widehat{P}_{sa} - P_{sa})X_u \right| \leq \sqrt{\frac{2\mathbb{V}_{P_{sa}}[X_u]\log(|U|/\delta')}{n(s,a)}} + \|X_u\|_{\text{span}} \frac{\log(|U|/\delta')}{3n(s,a)}.$$

Combining this inequality (for $u = u^\star$) along with (15), (16), and (18), we obtain that

$$\left| (\widehat{P}_{sa} - P_{sa})X \right|$$

$$\leq \left| (\widehat{P}_{sa} - P_{sa})X_{u^\star} \right| + \frac{2}{n_{\text{tot}}}$$

$$\leq \sqrt{\frac{2\mathbb{V}_{P_{sa}}[X_{u^\star}]\log(|U|/\delta')}{n(s,a)}} + \|X_u\|_{\text{span}} \frac{\log(|U|/\delta')}{3n(s,a)} + \frac{2}{n_{\text{tot}}}$$

$$\leq \sqrt{\frac{2\mathbb{V}_{P_{sa}}[X]\log(|U|/\delta')}{n(s,a)}} + \frac{1}{n_{\text{tot}}}\sqrt{\frac{2\log(|U|/\delta')}{n(s,a)}} + \|X\|_{\text{span}} \frac{\log(|U|/\delta')}{3n(s,a)} + \frac{2}{n_{\text{tot}}} \frac{\log(|U|/\delta')}{3n(s,a)} + \frac{2}{n_{\text{tot}}}.$$

Combining this with (20) we furthermore obtain that

$$\left|(\widehat{P}_{sa} - P_{sa})X\right|$$

$$\leq \sqrt{\frac{2\mathbb{V}_{P_{sa}}[X]\log(|U|/\delta')}{n(s,a)}} + \|X\|_{\mathrm{span}}\frac{\log(|U|/\delta')}{3n(s,a)} + \frac{1}{n_{\mathrm{tot}}}\left(\sqrt{\frac{2\log(|U|/\delta')}{n(s,a)}} + 2\frac{\log(|U|/\delta')}{3n(s,a)} + 2\right)$$

$$\leq \sqrt{\frac{2\mathbb{V}_{\widehat{P}_{sa}}[X]\log(|U|/\delta')}{n(s,a)-1}} + \|X\|_{\mathrm{span}}\frac{\log(|U|/\delta')}{3n(s,a)} + \frac{1}{n_{\mathrm{tot}}}\left(\sqrt{\frac{2\log(|U|/\delta')}{n(s,a)}} + 2\frac{\log(|U|/\delta')}{3n(s,a)} + 2\right)$$

$$+ \sqrt{\frac{2\log(|U|/\delta')}{n(s,a)}}\left(\|X\|_{\mathrm{span}}\sqrt{\frac{2\log|U|/\delta'}{n(s,a)-1}} + \frac{2}{n_{\mathrm{tot}}}\sqrt{\frac{2\log|U|/\delta'}{n(s,a)-1}} + \sqrt{\frac{n(s,a)}{n(s,a)-1}}\frac{1}{n_{\mathrm{tot}}} + \frac{1}{n_{\mathrm{tot}}}\right)$$

$$\leq \sqrt{\frac{2\mathbb{V}_{\widehat{P}_{sa}}[X]\log(|U|/\delta')}{n(s,a)-1}} + \|X\|_{\mathrm{span}}\frac{7}{3}\frac{\log(|U|/\delta')}{n(s,a)-1}$$

$$+ \frac{1}{n_{\mathrm{tot}}}\left(\sqrt{\frac{2\log(|U|/\delta')}{n(s,a)}} + 2\frac{\log(|U|/\delta')}{3n(s,a)} + 2 + 2\frac{2\log(|U|/\delta')}{n(s,a)-1} + 2\sqrt{\frac{2\log(|U|/\delta')}{n(s,a)-1}}\right)$$

$$\leq \sqrt{\frac{2\mathbb{V}_{\widehat{P}_{sa}}[X]\log(|U|/\delta')}{n(s,a)-1}} + \|X\|_{\mathrm{span}}\frac{7}{3}\frac{\log(|U|/\delta')}{n(s,a)-1}$$

$$+ \frac{1}{n_{\mathrm{tot}}}\left(2 + 3\sqrt{\frac{2\log(|U|/\delta')}{n(s,a)-1}} + \frac{14}{3}\frac{\log(|U|/\delta')}{n(s,a)-1}\right).$$

$$\square$$

Now we develop several leave-one-out constructions which satisfy the conditions of Lemma B.5.

**Lemma B.6.**   1. *(LOO construction for $\widehat{V}_{\mathrm{pe}}^{\star}$) For each $s,a$, there exists a set $U_{sa}^1 \subseteq \mathbb{R}$ with $|U_{sa}^1| \leq \frac{n_{\mathrm{tot}}}{1-\gamma}$ and random vectors $(X_u^1)_{u\in U_{sa}^1}$ such that 1) for all $u \in U_{sa}^1$, $X_u^1$ is independent from $S_{sa}^1, \ldots, S_{sa}^{n(s,a)}$, and 2) almost surely there exists some $u \in U_{sa}^1$ such that $\left\|\widehat{V}_{\mathrm{pe}}^{\star} - X_u^1\right\|_{\infty} \leq \frac{1}{n_{\mathrm{tot}}}$.*

2. *(LOO constructions for $T_{\beta(s,a)}(\widehat{P}_{sa}, \widehat{V}_{\mathrm{pe}}^{\star})$) For each $s,a$, there exists a set $U_{sa}^2 \subseteq \mathbb{R}$ with $|U_{sa}^2| \leq S\frac{n_{\mathrm{tot}}}{1-\gamma}$ and random vectors $(X_u^2)_{u\in U_{sa}^2}$ such that 1) for all $u \in U_{sa}^2$, $X_u^2$ is independent from $S_{sa}^1, \ldots, S_{sa}^{n(s,a)}$, and 2) almost surely there exists some $u \in U_{sa}^2$ such that $\left\|T_{\beta(s,a)}(\widehat{P}_{sa}, \widehat{V}_{\mathrm{pe}}^{\star}) - X_u^2\right\|_{\infty} \leq \frac{1}{n_{\mathrm{tot}}}$.*

3. *(LOO construction for $\widehat{V}_{\mathrm{pe}}^{\pi}$) Fix any policy $\pi$. For each $s,a$, there exists a set $U_{sa}^3 \subseteq \mathbb{R}$ with $|U_{sa}^3| \leq \frac{n_{\mathrm{tot}}}{1-\gamma}$ and random vectors $(X_u^3)_{u\in U_{sa}^3}$ such that 1) for all $u \in U_{sa}^3$, $X_u^3$ is independent from $S_{sa}^1, \ldots, S_{sa}^{n(s,a)}$, and 2) almost surely there exists some $u \in U_{sa}^3$ such that $\left\|\widehat{V}_{\mathrm{pe}}^{\pi} - X_u^3\right\|_{\infty} \leq \frac{1}{n_{\mathrm{tot}}}$.*

4. *(LOO constructions for $T_{\beta(s,a)}(\widehat{P}_{sa}, \widehat{V}_{\mathrm{pe}}^{\pi})$) Fix any policy $\pi$. For each $s,a$, there exists a set $U_{sa}^4 \subseteq \mathbb{R}$ with $|U_{sa}^4| \leq S\frac{n_{\mathrm{tot}}}{1-\gamma}$ and random vectors $(X_u^4)_{u\in U_{sa}^4}$ such that 1) for all $u \in U_{sa}^4$, $X_u^4$ is independent from $S_{sa}^1, \ldots, S_{sa}^{n(s,a)}$, and 2) almost surely there exists some $u \in U_{sa}^4$ such that $\left\|T_{\beta(s,a)}(\widehat{P}_{sa}, \widehat{V}_{\mathrm{pe}}^{\pi}) - X_u^4\right\|_{\infty} \leq \frac{1}{n_{\mathrm{tot}}}$.*

*Proof.* We start by showing item 1. Fix arbitrary $s \in \mathcal{S}, a \in \mathcal{A}$. For any $u \in \mathbb{R}$ we define the reward function $r^{s,u} \in \mathbb{R}^{\mathcal{S}\mathcal{A}}$, (random) transition matrix $\widehat{P}^s \in \mathbb{R}^{\mathcal{S}\mathcal{A}\times\mathcal{S}}$, and (random) operator

$\overline{\mathcal{T}}_{\mathrm{pe}}^{s,u} : \mathbb{R}^{\mathcal{S}} \to \mathbb{R}^{\mathcal{S}\mathcal{A}}$ by (for arbitrary $s' \in \mathcal{S}, a' \in \mathcal{S}, V \in \mathbb{R}^{\mathcal{S}}$)

$$\widehat{P}_{s'a'}^{s} = \begin{cases} e_s^{\top} & s' = s \\ \widehat{P}_{s'a'} & s' \neq s \end{cases}$$

$$r^{s,u}(s', a') = \begin{cases} u & s' = s \\ r(s', a') & s' \neq s \end{cases}$$

$$\overline{\mathcal{T}}_{\mathrm{pe}}^{s,u}(V)(s', a') = r^{s,u}(s', a') + \gamma \max\left\{ \widehat{P}_{s'a'}^{s} T_{\beta(s',a')}(\widehat{P}_{s'a'}^{s}, V) - b^s(s', a', V), \min_{s''}(V)(s'') \right\}$$

where

$$b^s(s', a', V) := \max\left\{ \sqrt{\beta(s', a')\mathbb{V}_{\widehat{P}_{s'a'}^{s}}\left[T_{\beta(s',a')}(\widehat{P}_{s'a'}^{s}, V)\right]}, \beta(s', a')\left\| T_{\beta(s',a')}(\widehat{P}_{s'a'}^{s}, V)\right\|_{\mathrm{span}} \right\} + \frac{5}{n_{\mathrm{tot}}} \tag{21}$$

Note $e_s^{\top}$ is a vector which is all $0$ except for a $1$ in state $s$, meaning that state $s$ is absorbing in $\widehat{P}^{s,u}$, for all actions. Also all actions receive reward $u$ in this state. All other state-action pairs have the same rewards and transition distributions as in the MDP $(\widehat{P}, r)$. Also, we have defined $b^s$ and $\overline{\mathcal{T}}_{\mathrm{pe}}^{s,u}$ in an identical manner to $b$ and $\overline{\mathcal{T}}_{\mathrm{pe}}$, except we now use $r^{s,u}$ and $\widehat{P}^s$ in place of $r$ and $\widehat{P}$. Since all of the properties of $\overline{\mathcal{T}}_{\mathrm{pe}}$ verified above only required $\widehat{P}$ to be a valid transition matrix and for $r$ to be a vector in $[0,1]^{\mathcal{S}\mathcal{A}}$, the properties hold identically for $\overline{\mathcal{T}}_{\mathrm{pe}}^{s,u}$, and thus by Lemma B.3 we have that $\overline{\mathcal{T}}_{\mathrm{pe}}^{s,u}$ is $\gamma$-Lipschitz.

Now we define $\widehat{\mathcal{L}}^{s,u} : \mathbb{R}^{\mathcal{S}} \to \mathbb{R}^{\mathcal{S}}$ as $\widehat{\mathcal{L}}^{s,u}(V) := M\overline{\mathcal{T}}_{\mathrm{pe}}^{s,u}(V)$ (for any $V \in \mathbb{R}^{\mathcal{S}}$). By the $\gamma$-Lipschitzness of $\overline{\mathcal{T}}_{\mathrm{pe}}^{s,u}$ and the $1$-Lipschitzness of $M$, we immediately have that $\widehat{\mathcal{L}}^{s,u}$ is a $\gamma$-contraction, since

$$\left\|\widehat{\mathcal{L}}^{s,u}(V_1) - \widehat{\mathcal{L}}^{s,u}(V_2)\right\|_{\infty} = \left\|M\overline{\mathcal{T}}_{\mathrm{pe}}^{s,u}(V_1) - M\overline{\mathcal{T}}_{\mathrm{pe}}^{s,u}(V_2)\right\|_{\infty} \leq \left\|\overline{\mathcal{T}}_{\mathrm{pe}}^{s,u}(V_1) - \overline{\mathcal{T}}_{\mathrm{pe}}^{s,u}(V_2)\right\|_{\infty} \leq \gamma\left\|V_1 - V_2\right\|_{\infty}$$

for any $V_1, V_2 \in \mathbb{R}^{\mathcal{S}}$. Therefore contractivity implies that there exists a unique fixed point of $\widehat{\mathcal{L}}^{s,u}$ (e.g. [Pugh, 2015, Chapter 4.5]), which we call $X_u^1$. Note that since $\widehat{\mathcal{L}}^{s,u}$ is defined without using $\widehat{P}_{sa}$, it is independent of all samples $S_{sa}^1, \ldots, S_{sa}^{n(s,a)}$ drawn from $P(\cdot \mid s, a)$.

Now, as intermediate steps, we show the following two properties:

A. For any $u, u' \in \mathbb{R}$, we have $\left\|X_u^1 - X_{u'}^1\right\|_{\infty} \leq \frac{|u - u'|}{1 - \gamma}$.

B. Letting $U^{\star} = \widehat{V}_{\mathrm{pe}}^{\star}(s) - \gamma \max_{\widetilde{a} \in \mathcal{A}} \max\left\{ \widehat{P}_{s\widetilde{a}}^{s} T_{\beta(s,\widetilde{a})}(\widehat{P}_{s\widetilde{a}}^{s}, \widehat{V}_{\mathrm{pe}}^{\star}) - \frac{5}{n_{\mathrm{tot}}}, \min_{s''}\widehat{V}_{\mathrm{pe}}^{\star}(s'') \right\}$, we have $X_{U^{\star}}^1 = \widehat{V}_{\mathrm{pe}}^{\star}$, and $U^{\star} \in [0, 1]$.

For A, letting $u, u' \in \mathbb{R}$, we can calculate that

$$\begin{aligned}
\left\|X_u^1 - X_{u'}^1\right\|_{\infty} &= \left\|\widehat{\mathcal{L}}^{s,u}(X_u^1) - \widehat{\mathcal{L}}^{s,u'}(X_{u'}^1)\right\|_{\infty} = \left\|M\overline{\mathcal{T}}_{\mathrm{pe}}^{s,u}(X_u^1) - M\overline{\mathcal{T}}_{\mathrm{pe}}^{s,u'}(X_{u'}^1)\right\|_{\infty} \\
&\leq \left\|\overline{\mathcal{T}}_{\mathrm{pe}}^{s,u}(X_u^1) - \overline{\mathcal{T}}_{\mathrm{pe}}^{s,u'}(X_{u'}^1)\right\|_{\infty} \\
&= \left\|r^{s,u} - r^{s,u'} + \overline{\mathcal{T}}_{\mathrm{pe}}^{s,u'}(X_u^1) - \overline{\mathcal{T}}_{\mathrm{pe}}^{s,u'}(X_{u'}^1)\right\|_{\infty} \\
&\leq \left\|r^{s,u} - r^{s,u'}\right\|_{\infty} + \left\|\overline{\mathcal{T}}_{\mathrm{pe}}^{s,u'}(X_u^1) - \overline{\mathcal{T}}_{\mathrm{pe}}^{s,u'}(X_{u'}^1)\right\|_{\infty} \\
&\leq |u - u'| + \gamma\left\|X_u^1 - X_{u'}^1\right\|_{\infty}
\end{aligned}$$

where the key equality step was that $\overline{\mathcal{T}}_{\mathrm{pe}}^{s,u}(X_u^1) = r^{s,u} - r^{s,u'} + \overline{\mathcal{T}}_{\mathrm{pe}}^{s,u'}(X_u^1)$, and in the final inequality we used $\gamma$-Lipschitzness of $\overline{\mathcal{T}}_{\mathrm{pe}}^{s,u'}$. Rearranging we obtain that $\left\|X_u^1 - X_{u'}^1\right\|_{\infty} \leq \frac{|u - u'|}{1 - \gamma}$ as desired, verifying A.

For B we first check that $X_{U^\star}^1 = \widehat{V}_{\text{pe}}^\star$. It suffices to check that $M\overline{\mathcal{T}}_{\text{pe}}^{s,U^\star}(\widehat{V}_{\text{pe}}^\star) = M\overline{\mathcal{T}}_{\text{pe}}(\widehat{V}_{\text{pe}}^\star)$, because then we would have that

$$\widehat{\mathcal{L}}^{s,U^\star}(\widehat{V}_{\text{pe}}^\star) = M\overline{\mathcal{T}}_{\text{pe}}^{s,U^\star}(\widehat{V}_{\text{pe}}^\star) = M\overline{\mathcal{T}}_{\text{pe}}(\widehat{V}_{\text{pe}}^\star) = M\widehat{\mathcal{T}}_{\text{pe}}(\widehat{Q}_{\text{pe}}^\star) = M\widehat{Q}_{\text{pe}}^\star = \widehat{V}_{\text{pe}}^\star$$

thus showing that $\widehat{V}_{\text{pe}}^\star$ is a fixed point of $\widehat{\mathcal{L}}^{s,U^\star}$, and by uniqueness of this fixed point we must have $X_{U^\star}^1 = \widehat{V}_{\text{pe}}^\star$. Comparing the definitions of $\overline{\mathcal{T}}_{\text{pe}}(\widehat{V}_{\text{pe}}^\star)$ and $\overline{\mathcal{T}}_{\text{pe}}^{s,U^\star}(\widehat{V}_{\text{pe}}^\star)$, it is immediate that $M\left(\overline{\mathcal{T}}_{\text{pe}}^{s,U^\star}(\widehat{V}_{\text{pe}}^\star)\right)(s') = M\left(\overline{\mathcal{T}}_{\text{pe}}(\widehat{V}_{\text{pe}}^\star)\right)(s')$ for all $s' \neq s$, so it remains to check the equality for $s' = s$.

First we argue that for all $a' \in \mathcal{A}$, we have $b^s(s, a', \widehat{V}_{\text{pe}}^\star) = \frac{5}{n_{\text{tot}}}$. If $\beta(s, a') > 1$ then we have $T_{\beta(s,a')}(\widehat{P}_{sa'}^s, \widehat{V}_{\text{pe}}^\star) = \left(\min_{s'} \widehat{V}_{\text{pe}}^\star(s')\right)\mathbf{1}$, and if $\beta(s, a') \leq 1$ then we have $T_{\beta(s,a')}(\widehat{P}_{sa'}^s, \widehat{V}_{\text{pe}}^\star) = \widehat{V}_{\text{pe}}^\star(s)\mathbf{1}$, since $\widehat{P}_{sa'}^s = e_s^\top$ ($\widehat{P}_{sa'}^s$ transitions to state $s$ with probability 1). Either way $T_{\beta(s,a')}(\widehat{P}_{sa'}^s, \widehat{V}_{\text{pe}}^\star)$ is a multiple of the all-ones vector, which implies $\mathbb{V}_{\widehat{P}_{s'a'}^s}\left[T_{\beta(s',a')}(\widehat{P}_{s'a'}^s, \widehat{V}_{\text{pe}}^\star)\right] = 0$ and $\left\|T_{\beta(s',a')}(\widehat{P}_{s'a'}^s, \widehat{V}_{\text{pe}}^\star)\right\|_{\text{span}} = 0$, and thus that $b^s(s, a', \widehat{V}_{\text{pe}}^\star) = \frac{5}{n_{\text{tot}}}$. Therefore by the construction of $U^\star$ we have that

$$M\left(\overline{\mathcal{T}}_{\text{pe}}^{s,U^\star}(\widehat{V}_{\text{pe}}^\star)\right)(s) = \max_{a'} U^\star + \gamma \max\left\{\widehat{P}_{sa'}^s T_{\beta(s,a')}(\widehat{P}_{sa'}^s, \widehat{V}_{\text{pe}}^\star) - b^s(s, a', \widehat{V}_{\text{pe}}^\star), \min_{s''}(\widehat{V}_{\text{pe}}^\star)(s'')\right\}$$

$$= \max_{a'} U^\star + \gamma \max\left\{\widehat{P}_{sa'}^s T_{\beta(s,a')}(\widehat{P}_{sa'}^s, \widehat{V}_{\text{pe}}^\star) - \frac{5}{n_{\text{tot}}}, \min_{s''} \widehat{V}_{\text{pe}}^\star(s'')\right\}$$

$$= \widehat{V}_{\text{pe}}^\star(s) - \gamma \max_{\widetilde{a}\in\mathcal{A}} \max\left\{\widehat{P}_{s\widetilde{a}}^s T_{\beta(s,\widetilde{a})}(\widehat{P}_{s\widetilde{a}}^s, \widehat{V}_{\text{pe}}^\star) - \frac{5}{n_{\text{tot}}}, \min_{s''} \widehat{V}_{\text{pe}}^\star(s'')\right\}$$

$$+ \gamma \max_{a'} \max\left\{\widehat{P}_{sa'}^s T_{\beta(s,a')}(\widehat{P}_{sa'}^s, \widehat{V}_{\text{pe}}^\star) - \frac{5}{n_{\text{tot}}}, \min_{s''} \widehat{V}_{\text{pe}}^\star(s'')\right\}$$

$$= \widehat{V}_{\text{pe}}^\star(s) = M\left(\overline{\mathcal{T}}_{\text{pe}}(\widehat{V}_{\text{pe}}^\star)\right)(s)$$

as desired, so we have checked that $X_{U^\star}^1 = \widehat{V}_{\text{pe}}^\star$.

Now it remains to verify that $U^\star \in [0, 1]$. Given our calculation of $T_{\beta(s,a')}(\widehat{P}_{sa'}^s, \widehat{V}_{\text{pe}}^\star)$ (for any $a' \in \mathcal{A}$) above, we have the alternate expression for $U^\star$

$$U^\star = \widehat{V}_{\text{pe}}^\star(s) - \gamma \begin{cases} \max\left\{\widehat{V}_{\text{pe}}^\star(s) - \frac{5}{n_{\text{tot}}}, \min_{s''} \widehat{V}_{\text{pe}}^\star(s'')\right\} & \exists a' \in \mathcal{A} : \beta(s, a') \leq 1 \\ \max\left\{\min_{s''} \widehat{V}_{\text{pe}}^\star(s'') - \frac{5}{n_{\text{tot}}}, \min_{s''} \widehat{V}_{\text{pe}}^\star(s'')\right\} & \text{o.w.} \end{cases}$$

$$= \widehat{V}_{\text{pe}}^\star(s) - \gamma \begin{cases} \max\left\{\widehat{V}_{\text{pe}}^\star(s) - \frac{5}{n_{\text{tot}}}, \min_{s''} \widehat{V}_{\text{pe}}^\star(s'')\right\} & \exists a' \in \mathcal{A} : \beta(s, a') \leq 1 \\ \min_{s''} \widehat{V}_{\text{pe}}^\star(s'') & \text{o.w.} \end{cases}.$$

We consider the two cases in the above expression. If $\exists a' \in \mathcal{A} : \beta(s, a') \leq 1$, then we can upper bound $U^\star$ as

$$U^\star \leq \widehat{V}_{\text{pe}}^\star(s) - \gamma \max\left\{\widehat{V}_{\text{pe}}^\star(s) - \frac{5}{n_{\text{tot}}}, \min_{s''} \widehat{V}_{\text{pe}}^\star(s'')\right\} \leq \widehat{V}_{\text{pe}}^\star(s) - \gamma\widehat{V}_{\text{pe}}^\star(s) \leq (1-\gamma)\frac{1}{1-\gamma} = 1$$

where the last inequality is due to the fact that $\widehat{V}_{\text{pe}}^\star = M\widehat{Q}_{\text{pe}}^\star \leq M\frac{1}{1-\gamma}\mathbf{1} = \frac{1}{1-\gamma}\mathbf{1}$ (by Lemma B.1). For the lower bound in this case, we have

$$U^\star = \min\left\{(1-\gamma)\widehat{V}_{\text{pe}}^\star(s) + \frac{5}{n_{\text{tot}}}, \widehat{V}_{\text{pe}}^\star(s) - \min_{s''} \widehat{V}_{\text{pe}}^\star(s'')\right\}$$

which is clearly $\geq 0$ (note the first term within the $\min$ is $\geq 0$ by Lemma B.1).

Now we consider the case that there does not exist $a' \in \mathcal{A}$ such that $\beta(s, a') \leq 1$, that is, the case that $\beta(s, a') > 1$ for all $a' \in \mathcal{A}$. Then as argued above we have for all $a' \in \mathcal{A}$ that $T_{\beta(s,a')}(\widehat{P}_{sa'}^s, \widehat{V}_{\text{pe}}^\star) =$

$\left(\min_{s''} \widehat{V}_{\mathrm{pe}}^\star(s'')\right)\mathbf{1}$, and so by the definition of $\widehat{\mathcal{T}}_{\mathrm{pe}}$ and the fact that $\widehat{Q}_{\mathrm{pe}}^\star$ is its fixed point and $\widehat{V}_{\mathrm{pe}}^\star = M\widehat{Q}_{\mathrm{pe}}^\star$, we have

$$\widehat{V}_{\mathrm{pe}}^\star(s) = \max_{a'\in\mathcal{A}} r(s,a') + \gamma \max\left\{\widehat{P}_{sa'} T_{\beta(s,a')}(\widehat{P}_{sa'}, \widehat{V}_{\mathrm{pe}}^\star) - b(s,a',\widehat{V}_{\mathrm{pe}}^\star), \min_{s''}\widehat{V}_{\mathrm{pe}}^\star(s'')\right\}$$

$$= \max_{a'\in\mathcal{A}} r(s,a') + \gamma \max\left\{\min_{s''}\widehat{V}_{\mathrm{pe}}^\star(s'') - b(s,a',\widehat{V}_{\mathrm{pe}}^\star), \min_{s''}\widehat{V}_{\mathrm{pe}}^\star(s'')\right\}$$

$$= \max_{a'\in\mathcal{A}} r(s,a') + \gamma \min_{s''}\widehat{V}_{\mathrm{pe}}^\star(s'')$$

(using the fact that $b(s,a',\widehat{V}_{\mathrm{pe}}^\star) \geq 0$ to compute the max). Hence in this case

$$U^\star = \widehat{V}_{\mathrm{pe}}^\star(s) - \gamma\min_{s''}\widehat{V}_{\mathrm{pe}}^\star(s'') = \max_{a'\in\mathcal{A}} r(s,a') + \gamma\min_{s''}\widehat{V}_{\mathrm{pe}}^\star(s'') - \gamma\min_{s''}\widehat{V}_{\mathrm{pe}}^\star(s'') = \max_{a'\in\mathcal{A}} r(s,a')$$

which is clearly in $[0,1]$. We have thus verified B.

Now unfix $u$ and let $U_{sa}^1$ be a set of $\frac{n_{\mathrm{tot}}}{1-\gamma}$ points chosen by dividing $[0,1]$ into $\frac{n_{\mathrm{tot}}}{1-\gamma}$ intervals and placing a point at the midpoint of each such interval. Note this guarantees that for any $x \in [0,1]$ there exists some $u \in U$ such that $|x - u| \leq \frac{1-\gamma}{2n_{\mathrm{tot}}}$. Therefore, letting $\widetilde{U}^\star \in U$ be this closest point in $U$ to the value $U^\star$, we have by A and B that

$$\left\|X_{\widetilde{U}^\star}^1 - \widehat{V}_{\mathrm{pe}}^\star\right\|_\infty = \left\|X_{\widetilde{U}^\star}^1 - X_{U^\star}^1\right\|_\infty \leq \frac{|\widetilde{U}^\star - U^\star|}{1-\gamma} \leq \frac{1}{1-\gamma}\frac{1-\gamma}{2n_{\mathrm{tot}}} = \frac{1}{2n_{\mathrm{tot}}} \leq \frac{1}{n_{\mathrm{tot}}}.$$

Therefore we have confirmed item 1.

Now we continue to item 2. Fix $s \in \mathcal{S}, a \in \mathcal{A}$, and define $U_{sa}^2 = U_{sa}^1 \times \mathcal{S}$. For each $u, s' \in U_{sa}^2$, we define

$$X_{u,s'}^2 = \mathrm{clip}(X_u^1, X_u^1(s')),$$

that is, we clip all entries of the vector $X_u^1$ constructed in the previous part so that they are $\leq X_u^1(s')$. Since $X_u^1$ was independent of all samples $S_{sa}^1, \ldots, S_{sa}^{n(s,a)}$ drawn from $P(\cdot \mid s,a)$, the same is true of $X_{u,s'}^2$. Define $S^\star(s,a)$ to be a state such that $Q_{\beta(s,a)}(\widehat{P}_{sa}, \widehat{V}_{\mathrm{pe}}^\star) = \widehat{V}_{\mathrm{pe}}^\star(S^\star(s,a))$ (if multiple states satisfy this, we can break ties in some consistent manner). Then for any $u, s' \in U_{sa}^2$ we have

$$\left\|T_{\beta(s,a)}(\widehat{P}_{sa}, \widehat{V}_{\mathrm{pe}}^\star) - X_{u,s'}^2\right\|_\infty = \left\|\mathrm{clip}\left(\widehat{V}_{\mathrm{pe}}^\star, Q_{\beta(s,a)}(\widehat{P}_{sa}, \widehat{V}_{\mathrm{pe}}^\star)\right) - \mathrm{clip}(X_u^1, X_u^1(s'))\right\|_\infty$$

$$= \left\|\mathrm{clip}\left(\widehat{V}_{\mathrm{pe}}^\star, \widehat{V}_{\mathrm{pe}}^\star(S^\star(s,a))\right) - \mathrm{clip}(X_u^1, X_u^1(s'))\right\|_\infty$$

$$\leq \left\|\mathrm{clip}\left(\widehat{V}_{\mathrm{pe}}^\star, \widehat{V}_{\mathrm{pe}}^\star(S^\star(s,a))\right) - \mathrm{clip}\left(X_u^1, \widehat{V}_{\mathrm{pe}}^\star(S^\star(s,a))\right)\right\|_\infty$$

$$+ \left\|\mathrm{clip}\left(X_u^1, \widehat{V}_{\mathrm{pe}}^\star(S^\star(s,a))\right) - \mathrm{clip}(X_u^1, X_u^1(s'))\right\|_\infty$$

$$\leq \left\|\widehat{V}_{\mathrm{pe}}^\star - X_u^1\right\|_\infty + \left|\widehat{V}_{\mathrm{pe}}^\star(S^\star(s,a)) - X_u^1(s')\right|. \tag{22}$$

From item 1 we know there exists some $u \in U_{sa}^1$ such that $\left\|\widehat{V}_{\mathrm{pe}}^\star - X_u^1\right\|_\infty \leq \frac{1}{2n_{\mathrm{tot}}}$, and furthermore if $s' = S^\star(s,a)$ then

$$\left|\widehat{V}_{\mathrm{pe}}^\star(S^\star(s,a)) - X_u^1(s')\right| = \left|\widehat{V}_{\mathrm{pe}}^\star(s') - X_u^1(s')\right| \leq \left\|\widehat{V}_{\mathrm{pe}}^\star - X_u^1\right\|_\infty \leq \frac{1}{2n_{\mathrm{tot}}}.$$

Combining these with (22) we conclude that almost surely there exists some $(u,s') \in U_{sa}^1 \times \mathcal{S} = U_{sa}^2$ such that $\left\|T_{\beta(s,a)}(\widehat{P}_{sa}, \widehat{V}_{\mathrm{pe}}^\star) - X_{u,s'}^2\right\|_\infty \leq \frac{1}{n_{\mathrm{tot}}}$ as desired. Therefore we have confirmed item 2.

For item 3 and item 4, we can use nearly identical constructions, with the only difference being that for item 3 we define $X_u^3$ to be the fixed point of the operator $\widehat{\mathcal{L}}^{\pi,s,u} : \mathbb{R}^\mathcal{S} \to \mathbb{R}^\mathcal{S}$ as $\widehat{\mathcal{L}}^{\pi,s,u}(V) := M^\pi \overline{\mathcal{T}}_{\mathrm{pe}}^{s,u}(V)$ (and otherwise use the same construction as for $X_u^1$), and then for item 4 we use $X_u^3$ in place of $X_u^1$ in the construction for $X_u^2$. Thus, the key difference is replacing $M$ with $M^\pi$ within the construction for $X_u^3$, and since the only properties of $M$ used were 1-Lipschitzness and that $M\mathbf{1} = \mathbf{1}$, which both hold with $M^\pi$ in place of $M$, and also the fact that $\widehat{V}_{\mathrm{pe}}^\star = M\widehat{Q}_{\mathrm{pe}}^\star$ which is analogous to the fact that $\widehat{V}_{\mathrm{pe}}^\pi = M^\pi \widehat{Q}_{\mathrm{pe}}^\pi$, all steps work in an analogous manner. $\qquad\square$

Now we can prove the key concentration inequalities needed for the rest of the proof.

**Lemma B.7.** *With probability at least $1 - \delta$, for all $s \in \mathcal{S}, a \in \mathcal{A}$, if $n(s,a) \geq 1 + 8\log\left(\frac{6S^2 A n_{\text{tot}}}{(1-\gamma)\delta}\right)$, then*

$$
\left|\left(\widehat{P}_{sa} - P_{sa}\right) T_{\beta(s,a)}(\widehat{P}_{sa}, \widehat{V}^\star_{\text{pe}})\right|
$$
$$
\leq \max\left\{\sqrt{\beta(s,a)\mathbb{V}_{\widehat{P}_{sa}}\left[T_{\beta(s,a)}(\widehat{P}_{sa}, \widehat{V}^\star_{\text{pe}})\right]}, \beta(s,a)\left\|T_{\beta(s,a)}(\widehat{P}_{sa}, \widehat{V}^\star_{\text{pe}})\right\|_{\text{span}}\right\} + \frac{4.5}{n_{\text{tot}}}
$$
$$
= b(s,a,\widehat{V}^\star_{\text{pe}}) - \frac{1}{2n_{\text{tot}}}
$$

*where $\alpha = 8\log\left(\frac{6S^2 A n_{\text{tot}}}{(1-\gamma)\delta}\right)$ and $\beta(s,a) = \frac{\alpha}{\max\{n(s,a)-1,1\}}$.*

*Proof.* Fix some $s \in \mathcal{S}$ and $a \in \mathcal{A}$. If $n(s,a) < 1 + 8\log\left(\frac{6S^2 A n_{\text{tot}}}{(1-\gamma)\delta}\right)$ then we have nothing to check. Otherwise, we can immediately combine item 2 of Lemma B.6 (which gives $|U| \leq S\frac{n_{\text{tot}}}{1-\gamma}$) with Lemma B.5 (since our condition on $n(s,a)$ clearly implies $n(s,a) \geq 2$) to conclude that with probability at least $1 - 6\delta'$,

$$
\left|(\widehat{P}_{sa} - P_{sa})T_{\beta(s,a)}(\widehat{P}_{sa}, \widehat{V}^\star_{\text{pe}})\right|
$$
$$
\leq \sqrt{\frac{2\mathbb{V}_{\widehat{P}_{sa}}\left[T_{\beta(s,a)}(\widehat{P}_{sa}, \widehat{V}^\star_{\text{pe}})\right]\log\left(S\frac{n_{\text{tot}}}{(1-\gamma)\delta'}\right)}{n(s,a)-1}} + \left\|T_{\beta(s,a)}(\widehat{P}_{sa}, \widehat{V}^\star_{\text{pe}})\right\|_{\text{span}} \frac{7}{3}\frac{\log\left(S\frac{n_{\text{tot}}}{(1-\gamma)\delta'}\right)}{n(s,a)-1}
$$
$$
+ \frac{1}{n_{\text{tot}}}\left(2 + 3\sqrt{\frac{2\log\left(S\frac{n_{\text{tot}}}{(1-\gamma)\delta'}\right)}{n(s,a)-1}} + \frac{14}{3}\frac{\log\left(S\frac{n_{\text{tot}}}{(1-\gamma)\delta'}\right)}{n(s,a)-1}\right).
$$

Taking a union bound over all $s \in \mathcal{S}, a \in \mathcal{A}$, and setting $\delta' = \frac{\delta}{6SA}$, we obtain that with probability at least $1 - \delta$, for all $s \in \mathcal{S}, a \in \mathcal{A}$ where $n(s,a) \geq 1 + 8\log\left(\frac{6S^2 A n_{\text{tot}}}{(1-\gamma)\delta}\right)$, we have

$$
\left|(\widehat{P}_{sa} - P_{sa})T_{\beta(s,a)}(\widehat{P}_{sa}, \widehat{V}^\star_{\text{pe}})\right|
$$
$$
\leq \sqrt{\frac{2\mathbb{V}_{\widehat{P}_{sa}}\left[T_{\beta(s,a)}(\widehat{P}_{sa}, \widehat{V}^\star_{\text{pe}})\right]\log\left(\frac{6S^2 A n_{\text{tot}}}{(1-\gamma)\delta}\right)}{n(s,a)-1}} + \left\|T_{\beta(s,a)}(\widehat{P}_{sa}, \widehat{V}^\star_{\text{pe}})\right\|_{\text{span}} \frac{7}{3}\frac{\log\left(\frac{6S^2 A n_{\text{tot}}}{(1-\gamma)\delta}\right)}{n(s,a)-1}
$$
$$
+ \frac{1}{n_{\text{tot}}}\left(2 + 3\sqrt{\frac{2\log\left(\frac{6S^2 A n_{\text{tot}}}{(1-\gamma)\delta}\right)}{n(s,a)-1}} + \frac{14}{3}\frac{\log\left(\frac{6S^2 A n_{\text{tot}}}{(1-\gamma)\delta}\right)}{n(s,a)-1}\right)
$$
$$
\leq \sqrt{\frac{2\mathbb{V}_{\widehat{P}_{sa}}\left[T_{\beta(s,a)}(\widehat{P}_{sa}, \widehat{V}^\star_{\text{pe}})\right]\log\left(\frac{6S^2 A n_{\text{tot}}}{(1-\gamma)\delta}\right)}{n(s,a)-1}} + \left\|T_{\beta(s,a)}(\widehat{P}_{sa}, \widehat{V}^\star_{\text{pe}})\right\|_{\text{span}} \frac{7}{3}\frac{\log\left(\frac{6S^2 A n_{\text{tot}}}{(1-\gamma)\delta}\right)}{n(s,a)-1} + \frac{4.5}{n_{\text{tot}}}
$$
$$
\leq 2\max\left\{\sqrt{\frac{2\mathbb{V}_{\widehat{P}_{sa}}\left[T_{\beta(s,a)}(\widehat{P}_{sa}, \widehat{V}^\star_{\text{pe}})\right]\log\left(\frac{6S^2 A n_{\text{tot}}}{(1-\gamma)\delta}\right)}{n(s,a)-1}}, \left\|T_{\beta(s,a)}(\widehat{P}_{sa}, \widehat{V}^\star_{\text{pe}})\right\|_{\text{span}} \frac{7}{3}\frac{\log\left(\frac{6S^2 A n_{\text{tot}}}{(1-\gamma)\delta}\right)}{n(s,a)-1}\right\} + \frac{4.5}{n_{\text{tot}}}
$$
$$
= \max\left\{\sqrt{\frac{8\mathbb{V}_{\widehat{P}_{sa}}\left[T_{\beta(s,a)}(\widehat{P}_{sa}, \widehat{V}^\star_{\text{pe}})\right]\log\left(\frac{6S^2 A n_{\text{tot}}}{(1-\gamma)\delta}\right)}{n(s,a)-1}}, \left\|T_{\beta(s,a)}(\widehat{P}_{sa}, \widehat{V}^\star_{\text{pe}})\right\|_{\text{span}} \frac{14}{3}\frac{\log\left(\frac{6S^2 A n_{\text{tot}}}{(1-\gamma)\delta}\right)}{n(s,a)-1}\right\} + \frac{4.5}{n_{\text{tot}}}
$$
$$
\leq b(s,a,\widehat{V}^\star_{\text{pe}}) - \frac{1}{2n_{\text{tot}}}.
$$

where the second inequality uses the assumption that $n(s,a) \geq 1 + 8\log\left(\frac{6S^2An_{\text{tot}}}{(1-\gamma)\delta}\right)$ and the fact that $2+3\sqrt{\frac{1}{4}+\frac{14}{3}\frac{1}{8}} < 4.5$, and then we bounded $a+b \leq 2\max\{a,b\}$. We also note that since we are in the case that $n(s,a) \geq 1 + 8\log\left(\frac{6S^2An_{\text{tot}}}{(1-\gamma)\delta}\right) \geq 9$, we have that $n(s,a) - 1 = \max\{n(s,a)-1, 1\}$. $\qquad\square$

**Lemma B.8.** *Fix any policy $\pi^\star$. With probability at least $1 - 2\delta$, for all $s \in \mathcal{S}, a \in \mathcal{A}$, if $n(s,a) \geq 1 + 8\ln\left(\frac{6S^2An_{\text{tot}}}{(1-\gamma)\delta}\right)$, then*

$$
\left|\left(\widehat{P}_{sa} - P_{sa}\right)T_{\beta(s,a)}(\widehat{P}_{sa}, \widehat{V}_{\text{pe}}^{\pi^\star})\right|
$$
$$
\leq \max\left\{\sqrt{\beta(s,a)\mathbb{V}_{\widehat{P}_{sa}}\left[T_{\beta(s,a)}(\widehat{P}_{sa}, \widehat{V}_{\text{pe}}^{\pi^\star})\right]}, \beta(s,a)\left\|T_{\beta(s,a)}(\widehat{P}_{sa}, \widehat{V}_{\text{pe}}^{\pi^\star})\right\|_{\text{span}}\right\} + \frac{5}{n_{\text{tot}}}
$$
$$
= b(s, a, \widehat{V}_{\text{pe}}^{\pi^\star})
$$

*and*

$$
\sqrt{\mathbb{V}_{\widehat{P}_{sa}}\left[\widehat{V}_{\text{pe}}^{\pi^\star}\right]} \leq \sqrt{\mathbb{V}_{P_{sa}}\left[\widehat{V}_{\text{pe}}^{\pi^\star}\right]} + \left\|\widehat{V}_{\text{pe}}^{\pi^\star}\right\|_{\text{span}}\sqrt{\frac{2\log\left(\frac{6S^2An_{\text{tot}}}{(1-\gamma)\delta}\right)}{n(s,a)}} + \frac{4}{n_{\text{tot}}} \tag{23}
$$

*and*

$$
\left|(\widehat{P}_{sa} - P_{sa})\widehat{V}_{\text{pe}}^{\pi^\star}\right| \leq \sqrt{\frac{2\mathbb{V}_{P_{sa}}\left[\widehat{V}_{\text{pe}}^{\pi^\star}\right]\log\left(\frac{6S^2An_{\text{tot}}}{(1-\gamma)\delta}\right)}{n(s,a)}} + \left\|\widehat{V}_{\text{pe}}^{\pi^\star}\right\|_{\text{span}}\frac{\log\left(\frac{6S^2An_{\text{tot}}}{(1-\gamma)\delta}\right)}{3n(s,a)} + \frac{3}{n_{\text{tot}}} \tag{24}
$$

*where $\alpha = 8\log\left(\frac{6S^2An_{\text{tot}}}{(1-\gamma)\delta}\right)$ and $\beta(s,a) = \frac{\alpha}{\max\{n(s,a)-1,1\}}$.*

*Proof.* The first statement is analogous to Lemma B.7 but uses the construction of item 4 of Lemma B.6 in place of item 2. Thus combining item 4 of Lemma B.6 with Lemma B.5, taking a union bound and performing the same simplifications, we obtain that with probability at least $1 - \delta$, for all $s \in \mathcal{S}, a \in \mathcal{A}$, if $n(s,a) \geq 1 + 8\ln\left(\frac{6S^2An_{\text{tot}}}{(1-\gamma)\delta}\right)$, then

$$
\left|\left(\widehat{P}_{sa} - P_{sa}\right)T_{\beta(s,a)}(\widehat{P}_{sa}, \widehat{V}_{\text{pe}}^{\pi^\star})\right| \leq b(s, a, \widehat{V}_{\text{pe}}^{\pi^\star}).
$$

Now we establish the second two properties. We will show that they both hold with probability $1 - \delta$, after which we are done since we can then use a union bound to combine with the above. Fixing some $s \in \mathcal{S}$ and $a \in \mathcal{A}$, if $n(s,a) < 1 + 8\log\left(\frac{6S^2An_{\text{tot}}}{(1-\gamma)\delta}\right)$ then we have nothing to check. Otherwise, we can immediately combine item 3 of Lemma B.6 (which gives $|U| \leq \frac{n_{\text{tot}}}{1-\gamma} \leq S\frac{n_{\text{tot}}}{1-\gamma}$) with Lemma B.5 (since our condition on $n(s,a)$ implies $n(s,a) \geq 2$) to conclude that with probability at least $1 - 6\delta'$, we have both

$$
\left|(\widehat{P}_{sa} - P_{sa})\widehat{V}_{\text{pe}}^{\pi^\star}\right| \leq \sqrt{\frac{2\mathbb{V}_{P_{sa}}\left[\widehat{V}_{\text{pe}}^{\pi^\star}\right]\log\left(S\frac{n_{\text{tot}}}{(1-\gamma)\delta'}\right)}{n(s,a)}} + \left\|\widehat{V}_{\text{pe}}^{\pi^\star}\right\|_{\text{span}}\frac{\log\left(S\frac{n_{\text{tot}}}{(1-\gamma)\delta'}\right)}{3n(s,a)}
$$
$$
+ \frac{1}{n_{\text{tot}}}\left(2 + \sqrt{\frac{2\log\left(S\frac{n_{\text{tot}}}{(1-\gamma)\delta'}\right)}{n(s,a)}} + 2\frac{\log\left(S\frac{n_{\text{tot}}}{(1-\gamma)\delta'}\right)}{3n(s,a)}\right) \tag{25}
$$

and

$$
\sqrt{\frac{n(s,a)}{n(s,a)-1}}\sqrt{\mathbb{V}_{\widehat{P}_{sa}}\left[\widehat{V}_{\text{pe}}^{\pi^\star}\right]} \leq \sqrt{\mathbb{V}_{P_{sa}}\left[\widehat{V}_{\text{pe}}^{\pi^\star}\right]} + \left\|\widehat{V}_{\text{pe}}^{\pi^\star}\right\|_{\text{span}}\sqrt{\frac{2\log\left(\frac{Sn_{\text{tot}}}{(1-\gamma)\delta'}\right)}{n(s,a)-1}} \tag{26}
$$
$$
+ \frac{1}{n_{\text{tot}}}\left(2\sqrt{\frac{2\log\left(\frac{Sn_{\text{tot}}}{(1-\gamma)\delta'}\right)}{n(s,a)-1}} + 3\right). \tag{27}
$$

Taking a union bound over all $s, a \in \mathcal{S}, \mathcal{A}$ and setting $\delta' = \frac{\delta}{6SA}$, we have that with probability at least $1 - \delta$, for all $s, a$ such that $n(s, a) \geq 1 + 8 \ln\left(\frac{6S^2 A n_{\text{tot}}}{(1-\gamma)\delta}\right)$, both

$$\left|(\widehat{P}_{sa} - P_{sa})\widehat{V}_{\text{pe}}^{\pi^\star}\right| \leq \sqrt{\frac{2\mathbb{V}_{P_{sa}}\left[\widehat{V}_{\text{pe}}^{\pi^\star}\right]\log\left(\frac{6S^2 A n_{\text{tot}}}{(1-\gamma)\delta}\right)}{n(s, a)}} + \left\|\widehat{V}_{\text{pe}}^{\pi^\star}\right\|_{\text{span}}\frac{\log\left(\frac{6S^2 A n_{\text{tot}}}{(1-\gamma)\delta}\right)}{3n(s, a)} + \frac{3}{n_{\text{tot}}}$$

and

$$\sqrt{\mathbb{V}_{\widehat{P}_{sa}}\left[\widehat{V}_{\text{pe}}^{\pi^\star}\right]} \leq \sqrt{\frac{n(s, a) - 1}{n(s, a)}}\sqrt{\mathbb{V}_{P_{sa}}\left[\widehat{V}_{\text{pe}}^{\pi^\star}\right]} + \left\|\widehat{V}_{\text{pe}}^{\pi^\star}\right\|_{\text{span}}\sqrt{\frac{2\log\left(\frac{6S^2 A n_{\text{tot}}}{(1-\gamma)\delta}\right)}{n(s, a)}}$$

$$+ \frac{1}{n_{\text{tot}}}\left(2\sqrt{\frac{2\log\left(\frac{6S^2 A n_{\text{tot}}}{(1-\gamma)\delta}\right)}{n(s, a)}} + 3\right)$$

$$\leq \sqrt{\mathbb{V}_{P_{sa}}\left[\widehat{V}_{\text{pe}}^{\pi^\star}\right]} + \left\|\widehat{V}_{\text{pe}}^{\pi^\star}\right\|_{\text{span}}\sqrt{\frac{2\log\left(\frac{6S^2 A n_{\text{tot}}}{(1-\gamma)\delta}\right)}{n(s, a)}} + \frac{4}{n_{\text{tot}}}$$

where for the first bound we simplified (25) using the condition on $n(s, a)$ and the fact that $2 + \sqrt{\frac{2}{8}} + \frac{2}{3}\frac{1}{8} < 3$, and for the second bound we simplified (27) also using the condition on $n(s, a)$ and then the fact that $2\sqrt{\frac{2}{8}} + 3 = 4$. $\qquad\square$

## B.4 Pessimism

In this subsection we establish the following essential pessimism property, making use of the previous concentration results and our construction of $\widehat{\mathcal{T}}_{\text{pe}}$.

**Lemma B.9.** *Under the event in Lemma B.7, we have that*

$$Q^{\widehat{\pi}} \geq \widehat{Q}.$$

*Proof.* We will show that $\mathcal{T}^{\widehat{\pi}}(\widehat{Q}) \geq \widehat{Q}$ (where $\mathcal{T}^{\widehat{\pi}}(Q) := r + PM^{\widehat{\pi}}Q$ is the Bellman evaluation operator for $\widehat{\pi}$), which by a standard argument implies that $Q^{\widehat{\pi}} \geq \widehat{Q}$, since we can then easily derive (by monotonicity of $\mathcal{T}^{\widehat{\pi}}$) that $(\mathcal{T}^{\widehat{\pi}})^{(k)}(\widehat{Q}) \geq \widehat{Q}$ for any integer $k \geq 0$, and thus

$$Q^{\widehat{\pi}} = \lim_{k \to \infty}(\mathcal{T}^{\widehat{\pi}})^{(k)}(\widehat{Q}) \geq \widehat{Q}.$$

Fixing arbitrary $s \in \mathcal{S}, a \in \mathcal{A}$, we will now verify that $\mathcal{T}^{\widehat{\pi}}(\widehat{Q})(s, a) \geq \widehat{Q}(s, a)$. From Lemma B.4 we have that $\widehat{\mathcal{T}}_{\text{pe}}(\widehat{Q})(s, a) \geq \widehat{Q}(s, a)$. We consider two cases based upon the value of $\widehat{\mathcal{T}}_{\text{pe}}(\widehat{Q})(s, a)$, which by (3) is either 1) equal to $r(s, a) + \gamma\widehat{P}_{sa}T_{\beta(s,a)}(\widehat{P}_{sa}, M\widehat{Q}) - \gamma b(s, a, M\widehat{Q})$ or 2) equal to $r(s, a) + \gamma\min_{s'}(M\widehat{Q})(s')$. In the simpler case 2, we thus have that

$$\widehat{\mathcal{T}}_{\text{pe}}(\widehat{Q})(s, a) = r(s, a) + \gamma\min_{s'}(M\widehat{Q})(s') \leq r(s, a) + \gamma P_{sa}M\widehat{Q} = r(s, a) + \gamma P_{sa}M^{\widehat{\pi}}\widehat{Q} = \mathcal{T}^{\widehat{\pi}}(\widehat{Q})(s, a)$$

using the facts that $\min_{s'}V(s') \leq P_{sa}V$ for any $V \in \mathbb{R}^{\mathcal{S}}$ (since $P_{sa}$ is a probability distribution) and that $M\widehat{Q} = M^{\widehat{\pi}}\widehat{Q}$ since $\widehat{\pi}$ is greedy with respect to $\widehat{Q}$. We therefore have that $\widehat{Q}(s, a) \leq \widehat{\mathcal{T}}_{\text{pe}}(\widehat{Q})(s, a) \leq \mathcal{T}^{\widehat{\pi}}(\widehat{Q})(s, a)$ in case 2, as desired. Now we consider case 1. Note that since we are in case 1, we must have that $\beta(s, a) \leq 1$, which implies that $n(s, a) \geq \alpha + 1$ (because if we had $\beta(s, a) > 1$, then we would have $T_{\beta(s,a)}(\widehat{P}_{sa}, M\widehat{Q}) = \min_{s'}(M\widehat{Q})(s')$, and $b(s, a, M\widehat{Q}) > 0$, so the term $T_{\beta(s,a)}(\widehat{P}_{sa}, M\widehat{Q}) - b(s, a, M\widehat{Q})$ could not have achieved the maximum in the definition (3)

of $\widehat{\mathcal{T}}_{\mathrm{pe}}$). Then we have that

$$
\begin{aligned}
\widehat{Q}(s,a) &\leq \widehat{\mathcal{T}}_{\mathrm{pe}}(\widehat{Q})(s,a) \\
&\leq \widehat{\mathcal{T}}_{\mathrm{pe}}(\widehat{Q}_{\mathrm{pe}}^{\star})(s,a) = r(s,a) + \gamma \widehat{P}_{sa} T_{\beta(s,a)}(\widehat{P}_{sa}, M\widehat{Q}_{\mathrm{pe}}^{\star}) - \gamma b(s,a, M\widehat{Q}_{\mathrm{pe}}^{\star}) \\
&\leq r(s,a) + \gamma P_{sa} T_{\beta(s,a)}(\widehat{P}_{sa}, M\widehat{Q}_{\mathrm{pe}}^{\star}) + \gamma \left| (\widehat{P}_{sa} - P_{sa}) T_{\beta(s,a)}(\widehat{P}_{sa}, M\widehat{Q}_{\mathrm{pe}}^{\star}) \right| - \gamma b(s,a, M\widehat{Q}_{\mathrm{pe}}^{\star}) \\
&\leq r(s,a) + \gamma P_{sa} T_{\beta(s,a)}(\widehat{P}_{sa}, M\widehat{Q}_{\mathrm{pe}}^{\star}) + \gamma b(s,a, M\widehat{Q}_{\mathrm{pe}}^{\star}) - \frac{1}{2n_{\mathrm{tot}}} - \gamma b(s,a, M\widehat{Q}_{\mathrm{pe}}^{\star}) \\
&\leq r(s,a) + \gamma P_{sa} M\widehat{Q}_{\mathrm{pe}}^{\star} - \frac{1}{2n_{\mathrm{tot}}} \\
&\leq r(s,a) + \gamma P_{sa} M\widehat{Q} \\
&= r(s,a) + \gamma P_{sa} M^{\widehat{\pi}}\widehat{Q} = \mathcal{T}^{\widehat{\pi}}(\widehat{Q})(s,a)
\end{aligned}
$$

where the first inequality is due to $\widehat{\mathcal{T}}_{\mathrm{pe}}(\widehat{Q}) \geq \widehat{Q}$ from Lemma B.4, the second inequality is due to monotonicity of $\widehat{\mathcal{T}}_{\mathrm{pe}}$ (Lemma B.1) and the fact that $\widehat{Q} \leq \widehat{Q}_{\mathrm{pe}}^{\star}$ (Lemma B.4), the third inequality is by triangle inequality, the fourth inequality is from Lemma B.7, the fifth inequality is from the trivial fact that elementwise $T_{\beta(s,a)}(\widehat{P}_{sa}, M\widehat{Q}_{\mathrm{pe}}^{\star}) \leq M\widehat{Q}_{\mathrm{pe}}^{\star}$, the sixth inequality follows from $\widehat{Q}_{\mathrm{pe}}^{\star} \leq \widehat{Q} + \frac{1}{2n_{\mathrm{tot}}}\mathbf{1}$ due to Lemma B.4 (since by monotonicity of $M$, $M\widehat{Q}_{\mathrm{pe}}^{\star} \leq M(\widehat{Q} + \frac{1}{2n_{\mathrm{tot}}}\mathbf{1}) = M\widehat{Q} + \frac{1}{2n_{\mathrm{tot}}}\mathbf{1}$), and the final equality is from the definition of $\widehat{\pi}$ (from Algorithm 1) since it is greedy with respect to $\widehat{Q}$. Combining the two cases we have shown that $\mathcal{T}^{\widehat{\pi}}(\widehat{Q}) \geq \widehat{Q}$ as desired. Combining the two cases we have shown that $\mathcal{T}^{\widehat{\pi}}(\widehat{Q}) \geq \widehat{Q}$ as desired. $\qquad \square$

### B.5 Policy hitting radius lemmas

In this subsection we establish some key properties regarding the relationship between $T_{\mathrm{hit}}$ and certain discounted policy occupancy measures which will appear in later analysis steps. We also establish some facts about $T_{\mathrm{hit}}$ of general interest and compare it to the mixing time.

Recall that $\eta_s := \inf\{t \geq 0 : S_t = s\}$ is the first hitting time of state $s$. We define an additional useful quantity: for any $s^{\star} \in \mathcal{S}$, let

$$
T_{\mathrm{hit}}(P, \pi, s^{\star}) := \sup_{s_0} \mathbb{E}_{s_0}^{\pi} \eta_{s^{\star}}.
$$

This is the maximum expected hitting time of state $s^{\star}$ in the Markov chain $P_{\pi}$ (which can be infinite). Then we have

$$
T_{\mathrm{hit}}(P, \pi) := \inf_{s^{\star}} T_{\mathrm{hit}}(P, \pi, s^{\star}) = \inf_{s^{\star}} \sup_{s_0} \mathbb{E}_{s_0}^{\pi} \eta_{s^{\star}}.
$$

$T_{\mathrm{hit}}(P, \pi)$ is finite if and only if $P_{\pi}$ is unichain:

**Lemma B.10.** *Fix a policy $\pi$ and an MDP transition kernel $P$. Then the Markov chain $P_{\pi}$ is unichain if and only if $T_{\mathrm{hit}}(P, \pi)$ is finite.*

*Proof.* First, suppose that $T_{\mathrm{hit}}(P, \pi)$ is finite. Then there exists some $s^{\star}$ such that for all $s_0 \in \mathcal{S}$, $\mathbb{E}_{s_0}^{\pi} \eta_{s^{\star}} < \infty$. Therefore $s^{\star}$ is reachable from any state, so all recurrent classes must contain $s^{\star}$, but since the irreducible closed recurrent classes (along with the transient states) form a partition of $\mathcal{S}$, this implies that there can only be one closed irreducible recurrent class, that is that $P_{\pi}$ is unichain.

Next, suppose that $P_{\pi}$ is unichain. Let $\tilde{s}^{\star}$ be some state in the single closed irreducible recurrent class of $P_{\pi}$. Now we argue that $\mathbb{E}_{s_0}^{\pi}[\eta_{\tilde{s}^{\star}}] < \infty$ for any $s_0 \in \mathcal{S}$. First, it is a standard fact (in finite Markov chains) that letting $C$ be the recurrent class, we have $M := \max_{s_0 \in C} \mathbb{E}_{s_0}^{\pi}[\eta_{\tilde{s}^{\star}}] < \infty$ (e.g. Kemeny and Snell [1976], where $\mathbb{E}_{s_0}^{\pi}[\eta_{\tilde{s}^{\star}}]$ is referred to as the mean first passage time). Now letting $s_0$ be any fixed transient state, since there exists a unique irreducible recurrent class $C$, letting $\eta_C = \inf\{t \geq 0 : S_t \in C\}$ be its first hitting time, it is also a standard fact (for finite Markov chains) that $\mathbb{E}_{s_0}^{\pi} \eta_C < \infty$ (replacing $C$ with a single absorbing state, the new chain becomes an absorbing chain, and the absorption time formulas in Kemeny and Snell [1976] imply $\mathbb{E}_{s_0}^{\pi} \eta_C < \infty$). Then a

calculation using the strong Markov property (where $\mathcal{F}_{\eta_C}$ is the stopped sigma-algebra associated with the stopping time $\eta_C$) implies that

$$\mathbb{E}_{s_0}^\pi[\eta_{\tilde{s}^\star}] = \mathbb{E}_{s_0}^\pi \mathbb{E}_{s_0}^\pi [\eta_{\tilde{s}^\star} \mid \mathcal{F}_{\eta_C}] = \mathbb{E}_{s_0}^\pi \left[ \mathbb{E}_{S_{\eta_C}}^\pi [\eta_{\tilde{s}^\star}] + \eta_C \right] \le \mathbb{E}_{s_0}^\pi [M + \eta_C] < \infty.$$

Since there are only a finite number of such transient states $s_0$, the maximum of $\mathbb{E}_{s_0}^\pi[\eta_{\tilde{s}^\star}]$ over all such states is finite. Hence $T_{\mathrm{hit}}(P, \pi) \le \max_{s_0 \in \mathcal{S}} \mathbb{E}_{s_0}^\pi[\eta_{\tilde{s}^\star}] < \infty$. $\qquad\square$

Define $d_{\gamma, s_0}^\pi \in \mathbb{R}^{\mathcal{S}}$ as

$$d_{\gamma, s_0}^\pi(s) = \sum_{t=0}^\infty \gamma^t e_{s_0}^\top P_\pi^t e_s.$$

We often drop the dependence on $\gamma, \pi$ and simply write $d_{s_0}$. We also define $d^\star(s) = \frac{1}{1-\gamma}\mu^\star(s)$.

**Lemma B.11.** *Let $s^\star \in \mathcal{S}$ satisfy $T_{\mathrm{hit}}(P, \pi) = T_{\mathrm{hit}}(P, \pi, s^\star)$. Then*

$$\sup_{s_0} \sum_{s \in \mathcal{S}} |d_{s_0}(s) - d_{s^\star}(s)| \le 2T_{\mathrm{hit}}(P, \pi)$$

*and*

$$\sup_{s_0, s_1} \sum_{s \in \mathcal{S}} |d_{s_0}(s) - d_{s_1}(s)| \le 4T_{\mathrm{hit}}(P, \pi).$$

*Proof.* We use a coupling argument, and these calculations are somewhat inspired by those in [Cheikhi and Russo, 2023, Lemma B.13]. Starting with the first statement, fix some $s_0 \in \mathcal{S}$. Let $S_0^\star, S_1^\star, \ldots$, be the stochastic process with distribution given by the Markov chain $P_\pi$ with starting state $s^\star$, and let $S_0, S_1, \ldots$, be the stochastic process with distribution given by the Markov chain $P_\pi$ but with starting state $s_0$. Let $\eta_{s^\star} = \inf\{t : S_t = s^\star\}$ be the first hitting time of the state $s^\star$ by the process $(S_t)_{t=0}^\infty$. Now define the process $S_0', S_1', \ldots$ identically to $(S_t)_{t=0}^\infty$ but to follow $(S_t^\star)_{t=0}^\infty$ once it reaches $s^\star$, that is $S_{\eta_{s^\star}}' = S_0^\star, S_{\eta_{s^\star}+1}' = S_1^\star$, and so on. It is a standard fact due to the Markov property that $(S_t')_{t=0}^\infty$ has the same distribution as $(S_t)_{t=0}^\infty$. Now add an absorbing terminal state $q$ (which we do not consider as an element of $\mathcal{S}$) and for all $t \ge 1$ let $Z_t \sim \mathrm{Bernoulli}(\gamma)$ (independently), and define the processes $(\tilde{S}_t')_{t=0}^\infty$ and $(\tilde{S}_t^\star)_{t=0}^\infty$ by $\tilde{S}_0' = S_0'$, $\tilde{S}_0^\star = S_0^\star$, and for all $t \ge 0$,

$$\tilde{S}_{t+1}^\star = \begin{cases} q & \exists k \in \{1, \ldots, t+1\} \text{ such that } Z_k = 1 \\ S_{t+1}^\star & \text{otherwise} \end{cases},$$

$$\tilde{S}_{t+1}' = \begin{cases} q & \exists k \in \{1, \ldots, t+1\} \text{ such that } Z_k = 1 \\ S_{t+1}' & \text{otherwise} \end{cases}.$$

Intuitively speaking, $(\tilde{S}_t')_{t=0}^\infty$ and $(\tilde{S}_t^\star)_{t=0}^\infty$ will reach the absorbing state $q$ at the same time, and the probability of reaching it on any given timestep is $\gamma$ if it has not yet been reached. It is a standard fact that $d_{\gamma, s_0}^\pi(s) = \mathbb{E}\sum_{t=0}^\infty \mathbb{I}(\tilde{S}_t' = s)$ and that $d_{\gamma, s^\star}^\pi(s) = \mathbb{E}\sum_{t=0}^\infty \mathbb{I}(\tilde{S}_t^\star = s)$. Hence using the above coupling we can bound $d_{\gamma, s_0}^\pi(s) - d_{\gamma, s^\star}^\pi(s)$. Specifically we have

$$\sum_{s \in \mathcal{S}} |d_{\gamma, s_0}^\pi(s) - d_{\gamma, s^\star}^\pi(s)| = \sum_{s \in \mathcal{S}} \left| \mathbb{E}\sum_{t=0}^\infty \left( \mathbb{I}(\tilde{S}_t' = s) - \mathbb{I}(\tilde{S}_t^\star = s) \right) \right|$$

$$\le \sum_{s \in \mathcal{S}} \mathbb{E} \left| \sum_{t=0}^\infty \left( \mathbb{I}(\tilde{S}_t' = s) - \mathbb{I}(\tilde{S}_t^\star = s) \right) \right|$$

$$= \mathbb{E} \sum_{s \in \mathcal{S}} \left| \sum_{t=0}^{\eta_q - 1} \left( \mathbb{I}(\tilde{S}_t' = s) - \mathbb{I}(\tilde{S}_t^\star = s) \right) \right| \qquad (28)$$

where in the final equality we let $\eta_q = \inf\{t \geq 1 : Z_t = 1\}$ be the first hitting time of the terminal state. Now we consider two cases. On the event that $\eta_q \leq \eta_{s^\star}$, we have

$$\sum_{s\in\mathcal{S}}\left|\sum_{t=0}^{\eta_q-1}\left(\mathbb{I}(\tilde{S}'_t = s) - \mathbb{I}(\tilde{S}^\star_t = s)\right)\right| \leq \sum_{s\in\mathcal{S}}\sum_{t=0}^{\eta_q-1}\left|\mathbb{I}(\tilde{S}'_t = s) - \mathbb{I}(\tilde{S}^\star_t = s)\right|$$

$$= \sum_{t=0}^{\eta_q-1} 2\mathbb{I}(\tilde{S}'_t \neq \tilde{S}^\star_t)$$

$$= 2\eta_q \leq 2\eta_{s^\star}.$$

On the event that $\eta_{s^\star} < \eta_q$, we have

$$\sum_{s\in\mathcal{S}}\left|\sum_{t=0}^{\eta_q-1}\left(\mathbb{I}(\tilde{S}'_t = s) - \mathbb{I}(\tilde{S}^\star_t = s)\right)\right|$$

$$= \sum_{s\in\mathcal{S}}\left|\sum_{t=0}^{\eta_q-1}\left(\mathbb{I}(S'_t = s) - \mathbb{I}(S^\star_t = s)\right)\right|$$

$$= \sum_{s\in\mathcal{S}}\left|\sum_{t=0}^{\eta_{s^\star}-1}\mathbb{I}(S'_t = s) + \sum_{t=\eta_{s^\star}}^{\eta_q-1}\mathbb{I}(S'_t = s) - \sum_{t=0}^{\eta_q-\eta_{s^\star}-1}\mathbb{I}(S^\star_t = s) - \sum_{t=\eta_q-\eta_{s^\star}}^{\eta_q-1}\mathbb{I}(S^\star_t = s)\right|$$

$$= \sum_{s\in\mathcal{S}}\left|\sum_{t=0}^{\eta_{s^\star}-1}\mathbb{I}(S'_t = s) - \sum_{t=\eta_q-\eta_{s^\star}}^{\eta_q-1}\mathbb{I}(S^\star_t = s)\right|$$

$$= \sum_{s\in\mathcal{S}}\left|\sum_{t=0}^{\eta_{s^\star}-1}\left(\mathbb{I}(S'_t = s) - \mathbb{I}(S^\star_{t+\eta_q-\eta_{s^\star}} = s)\right)\right|$$

$$\leq \sum_{s\in\mathcal{S}}\sum_{t=0}^{\eta_{s^\star}-1}\left|\mathbb{I}(S'_t = s) - \mathbb{I}(S^\star_{t+\eta_q-\eta_{s^\star}} = s)\right|$$

$$= 2\sum_{t=0}^{\eta_{s^\star}-1}\mathbb{I}(S'_t \neq S^\star_{t+\eta_q-\eta_{s^\star}}) \leq 2\eta_{s^\star}$$

using the fact that $S'_{\eta_{s^\star}} = S^\star_0, S'_{\eta_{s^\star}+1} = S^\star_1, \ldots$ to cancel terms. Combining the bounds for the two cases with (28), we have that

$$\sum_{s\in\mathcal{S}}\left|d^\pi_{\gamma,s_0}(s) - d^\pi_{\gamma,s^\star}(s)\right| \leq \mathbb{E}2\eta_{s^\star} \leq 2T_{\mathrm{hit}}(P,\pi)$$

as desired.

The second statement of the lemma follows immediately from the first, since by triangle inequality

$$\sup_{s_0,s_1}\sum_{s\in\mathcal{S}}|d_{s_0}(s) - d_{s_1}(s)| = \sup_{s_0,s_1}\|d_{s_0} - d_{s_1}\|_1 \leq \sup_{s_0,s_1}\|d_{s_0} - d_{s^\star}\|_1 + \|d_{s^\star} - d_{s_1}\|_1 \leq 4T_{\mathrm{hit}}(P,\pi).$$

$\square$

**Lemma B.12.** *Let $\pi$ be a policy such that $P_\pi$ is unichain, and let $\mu^\pi \in \mathbb{R}^{\mathcal{S}}$ denote its stationary distribution. Then*

$$\sum_{s\in\mathcal{S}}\left|d^\pi_{\gamma,s_0}(s) - \frac{1}{1-\gamma}\mu^\pi(s)\right| \leq 4T_{\mathrm{hit}}(P,\pi).$$

*Proof.* Since $\mu^\pi$ is a stationary distribution, we have for any $s \in \mathcal{S}$ that

$$\sum_{s'\in\mathcal{S}}\mu^\pi(s')d^\pi_{\gamma,s'}(s) = (\mu^\pi)^\top(I - \gamma P_\pi)^{-1}e_s = (\mu^\pi)^\top\sum_{t=0}^{\infty}\gamma^t P_\pi^t e_s = \sum_{t=0}^{\infty}\gamma^t(\mu^\pi)^\top e_s = \frac{1}{1-\gamma}\mu^\pi(s)$$

(since $(\mu^\pi)^\top P_\pi = (\mu^\pi)^\top$). Then we can calculate by Jensen's inequality that for any fixed $s \in \mathcal{S}$,

$$
\left| d^\pi_{\gamma,s_0}(s) - \frac{1}{1-\gamma}\mu^\pi(s) \right| = \left| d^\pi_{\gamma,s_0}(s) - \sum_{s'\in\mathcal{S}} \mu^\pi(s') d^\pi_{\gamma,s'}(s) \right|
$$

$$
= \left| \sum_{s'\in\mathcal{S}} \mu^\pi(s') \left( d^\pi_{\gamma,s_0}(s) - d^\pi_{\gamma,s'}(s) \right) \right|
$$

$$
\leq \sum_{s'\in\mathcal{S}} \mu^\pi(s') \left| d^\pi_{\gamma,s_0}(s) - d^\pi_{\gamma,s'}(s) \right|.
$$

Therefore

$$
\sum_{s\in\mathcal{S}} \left| d^\pi_{\gamma,s_0}(s) - \frac{1}{1-\gamma}\mu^\pi(s) \right| \leq \sum_{s\in\mathcal{S}}\sum_{s'\in\mathcal{S}} \mu^\pi(s') \left| d^\pi_{\gamma,s_0}(s) - d^\pi_{\gamma,s'}(s) \right| = \sum_{s'\in\mathcal{S}} \mu^\pi(s') \sum_{s\in\mathcal{S}} \left| d^\pi_{\gamma,s_0}(s) - d^\pi_{\gamma,s'}(s) \right|
$$

$$
\leq \sum_{s'\in\mathcal{S}} \mu^\pi(s') 4T_{\text{hit}}(P,\pi) = 4T_{\text{hit}}(P,\pi)
$$

where in the second inequality step we used Lemma B.11. $\qquad\square$

**Lemma B.13.** *For any policy $\pi$, $\|h^\pi\|_{\text{span}} \leq 4T_{\text{hit}}(P,\pi)$.*

*Proof.* Note that by Lemma B.10, if $P_\pi$ is not unichain then $T_{\text{hit}}(P,\pi) = \infty$ and so the desired bound holds trivially (note $\|h^\pi\|_{\text{span}}$ is always finite). So we can now focus on the case that $P_\pi$ is unichain. This implies $\rho^\pi$ is a state-independent constant. In this case it is a standard fact (e.g. [Puterman, 1994, Corollary 8.2.4]) that for any $s, s' \in \mathcal{S}$,

$$
h^\pi(s) - h^\pi(s') = \lim_{\gamma\to 1^-} V^\pi_\gamma(s) - V^\pi_\gamma(s').
$$

Therefore

$$
\|h^\pi\|_{\text{span}} = \max_{s,s'\in\mathcal{S}} h^\pi(s) - h^\pi(s')
$$

$$
= \max_{s,s'\in\mathcal{S}} \lim_{\gamma\to 1^-} V^\pi_\gamma(s) - V^\pi_\gamma(s')
$$

$$
= \max_{s,s'\in\mathcal{S}} \lim_{\gamma\to 1^-} e_s^\top (I - \gamma P_\pi)^{-1} r_\pi - e_{s'}^\top (I - \gamma P_\pi)^{-1} r_\pi
$$

$$
= \max_{s,s'\in\mathcal{S}} \lim_{\gamma\to 1^-} (d^\pi_{\gamma,s} - d^\pi_{\gamma,s'}) r_\pi
$$

$$
\leq \max_{s,s'\in\mathcal{S}} \lim_{\gamma\to 1^-} \left\| d^\pi_{\gamma,s} - d^\pi_{\gamma,s'} \right\|_1 \|r_\pi\|_\infty
$$

$$
\leq \max_{s,s'\in\mathcal{S}} \lim_{\gamma\to 1^-} 4T_{\text{hit}}(P,\pi)
$$

$$
= 4T_{\text{hit}}(P,\pi)
$$

where the inequality steps are by Holder's inequality and Lemma B.11. $\qquad\square$

### B.5.1 Relationship between policy hitting radius and uniform mixing time

Here we argue that there is generally no relationship between the policy hitting radius and the mixing time. First, if $P_\pi$ is a unichain and periodic Markov chain, then the mixing time will be infinite/undefined whereas $T_{\text{hit}}(P,\pi) < \infty$ by Lemma B.10.

Now we show an example where the mixing time can be arbitrarily smaller than the policy hitting radius. Suppose that $P, \pi$ are defined so that $P_\pi$ is the random walk on the complete graph on $L$ nodes, where $L$ is any positive integer. Then $\mu^\pi(s) = 1/L$ for all $s \in \mathcal{S}$, and after just one step from any starting state we have that $S_1$ has distribution $\mu^\pi$ so $\tau(\pi) = 1$. However, for any fixed starting state $s_0$ and any state $s \neq s_0$, we have that $\eta_s \sim \text{Geom}(1/L)$, so $\mathbb{E}^\pi_{s_0}\eta_s = L$, and hence $T_{\text{hit}}(P,\pi) = L$.

## B.6 Error analysis

Now we can continue with analyzing the relationship between $\widehat{Q}^\star_{\text{pe}}$ and $\rho^{\pi^\star}$, for a comparator policy $\pi^\star$. Having established pessimism (Lemma B.9), which implies an upper bound on $\widehat{Q}^\star_{\text{pe}}$, we now seek to lower-bound this quantity. Since (by Lemma B.1) $\widehat{Q}^\star_{\text{pe}} \geq \widehat{Q}^{\pi^\star}_{\text{pe}}$, it suffices to lower-bound $\widehat{Q}^{\pi^\star}_{\text{pe}}$ in terms of $V^{\pi^\star}$, which is then related to $\rho^{\pi^\star}$.

**Lemma B.14.** *For any probability distribution $\mu \in \Delta^{\mathcal{S}}$, any $V \in \mathbb{R}^{\mathcal{S}}$, and any $\beta \in [0, 1]$, we have that*

$$\mathbb{V}_\mu \left[ T_\beta(\mu, V) \right] \leq \mathbb{V}_\mu \left[ V \right].$$

*Proof.* We prove this by showing the more general statement that for any random variable $X$ and any scalar $a$,

$$\mathbb{V} \left[ \min(X, a) \right] \leq \mathbb{V} \left[ X \right].$$

Let $T = \min(X, a)$ and $\Delta = X - T$. Then

$$\mathbb{V} \left[ X \right] = \mathbb{V} \left[ T \right] + \mathbb{V} \left[ \Delta \right] + 2\text{Cov}(T, \Delta).$$

Thus to show $\mathbb{V} \left[ X \right] \geq \mathbb{V} \left[ T \right]$ it suffices to show that $\text{Cov}(T, \Delta) \geq 0$. Now we compute

$$\begin{aligned}
\text{Cov}(T, \Delta) &= \mathbb{E} \left[ \Delta(T - \mathbb{E}T) \right] \\
&= \mathbb{E} \left[ \Delta(T - \mathbb{E}T)\mathbb{I}\{X \geq a\} \right] + \mathbb{E} \left[ \Delta(T - \mathbb{E}T)\mathbb{I}\{X < a\} \right].
\end{aligned}$$

On the event $\{X < a\}$ we have $\Delta = 0$, so $\mathbb{E} \left[ \Delta(T - \mathbb{E}T)\mathbb{I}\{X < a\} \right] = 0$. On the event $\{X \geq a\}$, $(T - \mathbb{E}T) \geq 0$ since $T = a$ and $\mathbb{E}T \leq a$, and $\Delta \geq 0$, so $\mathbb{E} \left[ \Delta(T - \mathbb{E}T)\mathbb{I}\{X \geq a\} \right] \geq 0$. Therefore $\text{Cov}(T, \Delta) \geq 0$ as desired. $\qquad\square$

**Lemma B.15.** *Fix any deterministic policy $\pi^\star$. Under the event in Lemma B.8,*

$$V^{\pi^\star} - \widehat{V}^{\pi^\star}_{\text{pe}} \leq (I - \gamma P_{\pi^\star})^{-1} \gamma \tilde{b}_{\pi^\star}$$

*where*

$$\tilde{b}_{\pi^\star}(s) = 2\sqrt{\beta(s, \pi^\star(s)) \mathbb{V}_{P_{s\pi^\star(s)}} \left[ \widehat{V}^{\pi^\star}_{\text{pe}} \right]} + 4\beta(s, \pi^\star(s)) \left\| \widehat{V}^{\pi^\star}_{\text{pe}} \right\|_{\text{span}} + \frac{12}{n_{\text{tot}}}.$$

*We also have that*

$$\widehat{V}^{\pi^\star}_{\text{pe}} - \gamma P_{\pi^\star} \widehat{V}^{\pi^\star}_{\text{pe}} + \gamma \tilde{b}_{\pi^\star} \geq r_{\pi^\star}. \tag{29}$$

*Proof.* Fix $s \in \mathcal{S}, a \in \mathcal{A}$. First we handle the case that $\beta(s, a) \leq 1$. This implies that $n(s, a) \geq 1 + \alpha = 1 + 8\log\left(\frac{6S^2 An_{\text{tot}}}{(1-\gamma)\delta}\right)$. By the definition (8) of $\widehat{\mathcal{T}}^{\pi^\star}_{\text{pe}}$ we have that

$$\widehat{Q}^{\pi^\star}_{\text{pe}}(s, a) \geq r(s, a) + \gamma \widehat{P}_{sa} T_{\beta(s,a)}(\widehat{P}_{sa}, \widehat{V}^{\pi^\star}_{\text{pe}}) - \gamma b(s, a, \widehat{V}^{\pi^\star}_{\text{pe}}). \tag{30}$$

By the definition of $T_{\beta(s,a)}(\widehat{P}_{sa}, \widehat{V}_{\mathrm{pe}}^{\pi^\star})$ we have that (elementwise)

$$
\begin{aligned}
\widehat{P}_{sa}T_{\beta(s,a)}(\widehat{P}_{sa}, \widehat{V}_{\mathrm{pe}}^{\pi^\star}) &= \sum_{s'} \widehat{P}_{sa}(s')T_{\beta(s,a)}(\widehat{P}_{sa}, \widehat{V}_{\mathrm{pe}}^{\pi^\star})(s') \\
&= \sum_{s':\widehat{V}_{\mathrm{pe}}^{\pi^\star}(s') \leq Q_{\beta(s,a)}(\widehat{P}_{sa}, \widehat{V}_{\mathrm{pe}}^{\pi^\star})} \widehat{P}_{sa}(s')T_{\beta(s,a)}(\widehat{P}_{sa}, \widehat{V}_{\mathrm{pe}}^{\pi^\star})(s') \\
&\qquad + \sum_{s':\widehat{V}_{\mathrm{pe}}^{\pi^\star}(s') > Q_{\beta(s,a)}(\widehat{P}_{sa}, \widehat{V}_{\mathrm{pe}}^{\pi^\star})} \widehat{P}_{sa}(s')T_{\beta(s,a)}(\widehat{P}_{sa}, \widehat{V}_{\mathrm{pe}}^{\pi^\star})(s') \\
&= \sum_{s':\widehat{V}_{\mathrm{pe}}^{\pi^\star}(s') \leq Q_{\beta(s,a)}(\widehat{P}_{sa}, \widehat{V}_{\mathrm{pe}}^{\pi^\star})} \widehat{P}_{sa}(s')\widehat{V}_{\mathrm{pe}}^{\pi^\star}(s') \\
&\qquad + \sum_{s':\widehat{V}_{\mathrm{pe}}^{\pi^\star}(s') > Q_{\beta(s,a)}(\widehat{P}_{sa}, \widehat{V}_{\mathrm{pe}}^{\pi^\star})} \widehat{P}_{sa}(s')Q_{\beta(s,a)}(\widehat{P}_{sa}, \widehat{V}_{\mathrm{pe}}^{\pi^\star}) \\
&\geq \sum_{s':\widehat{V}_{\mathrm{pe}}^{\pi^\star}(s') \leq Q_{\beta(s,a)}(\widehat{P}_{sa}, \widehat{V}_{\mathrm{pe}}^{\pi^\star})} \widehat{P}_{sa}(s')\widehat{V}_{\mathrm{pe}}^{\pi^\star}(s') \\
&\qquad + \sum_{s':\widehat{V}_{\mathrm{pe}}^{\pi^\star}(s') > Q_{\beta(s,a)}(\widehat{P}_{sa}, \widehat{V}_{\mathrm{pe}}^{\pi^\star})} \widehat{P}_{sa}(s')\left(\widehat{V}_{\mathrm{pe}}^{\pi^\star}(s') - \left\|\widehat{V}_{\mathrm{pe}}^{\pi^\star}\right\|_{\mathrm{span}}\right) \\
&> \widehat{P}_{sa}\widehat{V}_{\mathrm{pe}}^{\pi^\star} - \beta(s,a)\left\|\widehat{V}_{\mathrm{pe}}^{\pi^\star}\right\|_{\mathrm{span}} \tag{31}
\end{aligned}
$$

where in the final inequality we used that $\sum_{s':\widehat{V}_{\mathrm{pe}}^{\pi^\star}(s') > Q_{\beta(s,a)}(\widehat{P}_{sa}, \widehat{V}_{\mathrm{pe}}^{\pi^\star})} \widehat{P}_{sa}(s') < \beta(s,a)$. Using (24) from Lemma B.8 to relate $\widehat{P}_{sa}\widehat{V}_{\mathrm{pe}}^{\pi^\star}$ to $P_{sa}\widehat{V}_{\mathrm{pe}}^{\pi^\star}$, we can further bound

$$
\begin{aligned}
\widehat{P}_{sa}T_{\beta(s,a)}(\widehat{P}_{sa}, \widehat{V}_{\mathrm{pe}}^{\pi^\star}) &\geq P_{sa}\widehat{V}_{\mathrm{pe}}^{\pi^\star} - \left|(\widehat{P}_{sa} - P_{sa})\widehat{V}_{\mathrm{pe}}^{\pi^\star}\right| - \beta(s,a)\left\|\widehat{V}_{\mathrm{pe}}^{\pi^\star}\right\|_{\mathrm{span}} \\
&\geq P_{sa}\widehat{V}_{\mathrm{pe}}^{\pi^\star} - \sqrt{\frac{2\mathbb{V}_{P_{sa}}\left[\widehat{V}_{\mathrm{pe}}^{\pi^\star}\right]\log\left(\frac{6S^2An_{\mathrm{tot}}}{(1-\gamma)\delta}\right)}{n(s,a)}} - \left\|\widehat{V}_{\mathrm{pe}}^{\pi^\star}\right\|_{\mathrm{span}}\frac{\log\left(\frac{6S^2An_{\mathrm{tot}}}{(1-\gamma)\delta}\right)}{3n(s,a)} \\
&\quad - \frac{3}{n_{\mathrm{tot}}} - \beta(s,a)\left\|\widehat{V}_{\mathrm{pe}}^{\pi^\star}\right\|_{\mathrm{span}} \\
&\geq P_{sa}\widehat{V}_{\mathrm{pe}}^{\pi^\star} - \sqrt{\beta(s,a)\mathbb{V}_{P_{sa}}\left[\widehat{V}_{\mathrm{pe}}^{\pi^\star}\right]} - 2\beta(s,a)\left\|\widehat{V}_{\mathrm{pe}}^{\pi^\star}\right\|_{\mathrm{span}} - \frac{3}{n_{\mathrm{tot}}}. \tag{32}
\end{aligned}
$$

To finish lower-bounding (30) we must also lower-bound $b(s,a,\widehat{V}_{\mathrm{pe}}^{\pi^\star})$. It is immediate to see that $\left\|T_{\beta(s,a)}(\widehat{P}_{sa}, \widehat{V}_{\mathrm{pe}}^{\pi^\star})\right\|_{\mathrm{span}} \leq \left\|\widehat{V}_{\mathrm{pe}}^{\pi^\star}\right\|_{\mathrm{span}}$, and also by Lemma B.14 (since we are in the $\beta(s,a) \leq 1$ case) we have that $\mathbb{V}_{\widehat{P}_{sa}}\left[T_{\beta(s,a)}(\widehat{P}_{sa}, \widehat{V}_{\mathrm{pe}}^{\pi^\star})\right] \leq \mathbb{V}_{\widehat{P}_{sa}}\left[\widehat{V}_{\mathrm{pe}}^{\pi^\star}\right]$. These two facts yield that

$$
\begin{aligned}
b(s,a,\widehat{V}_{\mathrm{pe}}^{\pi^\star}) &= \max\left\{\sqrt{\beta(s,a)\mathbb{V}_{\widehat{P}_{sa}}\left[T_{\beta(s,a)}(\widehat{P}_{sa}, \widehat{V}_{\mathrm{pe}}^{\pi^\star})\right]}, \beta(s,a)\left\|T_{\beta(s,a)}(\widehat{P}_{sa}, \widehat{V}_{\mathrm{pe}}^{\pi^\star})\right\|_{\mathrm{span}}\right\} + \frac{5}{n_{\mathrm{tot}}} \\
&\leq \max\left\{\sqrt{\beta(s,a)\mathbb{V}_{\widehat{P}_{sa}}\left[\widehat{V}_{\mathrm{pe}}^{\pi^\star}\right]}, \beta(s,a)\left\|\widehat{V}_{\mathrm{pe}}^{\pi^\star}\right\|_{\mathrm{span}}\right\} + \frac{5}{n_{\mathrm{tot}}} \\
&\leq \sqrt{\beta(s,a)\mathbb{V}_{\widehat{P}_{sa}}\left[\widehat{V}_{\mathrm{pe}}^{\pi^\star}\right]} + \beta(s,a)\left\|\widehat{V}_{\mathrm{pe}}^{\pi^\star}\right\|_{\mathrm{span}} + \frac{5}{n_{\mathrm{tot}}}. \tag{33}
\end{aligned}
$$

Furthermore, using the bound (23) from Lemma B.8, we can further bound (33) as

$$b(s, a, \widehat{V}_{\mathrm{pe}}^{\pi^\star})$$

$$\leq \sqrt{\beta(s,a)\mathbb{V}_{\widehat{P}_{sa}}\left[\widehat{V}_{\mathrm{pe}}^{\pi^\star}\right]} + \beta(s,a)\left\|\widehat{V}_{\mathrm{pe}}^{\pi^\star}\right\|_{\mathrm{span}} + \frac{5}{n_{\mathrm{tot}}}$$

$$\leq \sqrt{\beta(s,a)}\left(\sqrt{\mathbb{V}_{P_{sa}}\left[\widehat{V}_{\mathrm{pe}}^{\pi^\star}\right]} + \left\|\widehat{V}_{\mathrm{pe}}^{\pi^\star}\right\|_{\mathrm{span}}\sqrt{\frac{2\log\left(\frac{6S^2 An_{\mathrm{tot}}}{(1-\gamma)\delta}\right)}{n(s,a)}} + \frac{4}{n_{\mathrm{tot}}}\right) + \beta(s,a)\left\|\widehat{V}_{\mathrm{pe}}^{\pi^\star}\right\|_{\mathrm{span}} + \frac{5}{n_{\mathrm{tot}}}$$

$$\leq \sqrt{\beta(s,a)}\left(\sqrt{\mathbb{V}_{P_{sa}}\left[\widehat{V}_{\mathrm{pe}}^{\pi^\star}\right]} + \left\|\widehat{V}_{\mathrm{pe}}^{\pi^\star}\right\|_{\mathrm{span}}\sqrt{\beta(s,a)} + \frac{4}{n_{\mathrm{tot}}}\right) + \beta(s,a)\left\|\widehat{V}_{\mathrm{pe}}^{\pi^\star}\right\|_{\mathrm{span}} + \frac{5}{n_{\mathrm{tot}}}$$

$$\leq \sqrt{\beta(s,a)\mathbb{V}_{P_{sa}}\left[\widehat{V}_{\mathrm{pe}}^{\pi^\star}\right]} + 2\beta(s,a)\left\|\widehat{V}_{\mathrm{pe}}^{\pi^\star}\right\|_{\mathrm{span}} + \frac{9}{n_{\mathrm{tot}}} \tag{34}$$

(using the definition of $\beta(s,a)$ and the fact that we are in the $\beta(s,a) \leq 1$ case).

Combining (34) and (32) with (30) we obtain that

$$\widehat{Q}_{\mathrm{pe}}^{\pi^\star}(s,a) \geq r(s,a) + \gamma P_{sa}\widehat{V}_{\mathrm{pe}}^{\pi^\star} - 2\gamma\sqrt{\beta(s,a)\mathbb{V}_{P_{sa}}\left[\widehat{V}_{\mathrm{pe}}^{\pi^\star}\right]} - 4\gamma\beta(s,a)\left\|\widehat{V}_{\mathrm{pe}}^{\pi^\star}\right\|_{\mathrm{span}} - \frac{12\gamma}{n_{\mathrm{tot}}}$$

$$= r(s,a) + \gamma P_{sa}\widehat{V}_{\mathrm{pe}}^{\pi^\star} - \gamma\tilde{b}(s,a)$$

where we define $\tilde{b}(s,a) = \sqrt{\beta(s,a)\mathbb{V}_{P_{sa}}\left[\widehat{V}_{\mathrm{pe}}^{\pi^\star}\right]} + 4\beta(s,a)\left\|\widehat{V}_{\mathrm{pe}}^{\pi^\star}\right\|_{\mathrm{span}} + \frac{12}{n_{\mathrm{tot}}}$.

Now for the simpler case that $\beta(s,a) > 1$, we have that

$$\widehat{Q}_{\mathrm{pe}}^{\pi^\star}(s,a) = r(s,a) + \gamma\min_{s'}\widehat{V}_{\mathrm{pe}}^{\pi^\star}(s')$$

$$\geq r(s,a) + \gamma P_{sa}\widehat{V}_{\mathrm{pe}}^{\pi^\star} - \gamma\left\|\widehat{V}_{\mathrm{pe}}^{\pi^\star}\right\|_{\mathrm{span}}$$

$$\geq r(s,a) + \gamma P_{sa}\widehat{V}_{\mathrm{pe}}^{\pi^\star} - \gamma\beta(s,a)\left\|\widehat{V}_{\mathrm{pe}}^{\pi^\star}\right\|_{\mathrm{span}}$$

$$\geq r(s,a) + \gamma P_{sa}\widehat{V}_{\mathrm{pe}}^{\pi^\star} - \gamma\tilde{b}(s,a).$$

Combining the two cases of $\beta(s,a)$, we have for all $s,a$ that $\widehat{Q}_{\mathrm{pe}}^{\pi^\star}(s,a) \geq r(s,a) + \gamma P_{sa}\widehat{V}_{\mathrm{pe}}^{\pi^\star} - \gamma\tilde{b}(s,a)$. Therefore by monotonicity of $M^{\pi^\star}$,

$$\widehat{V}_{\mathrm{pe}}^{\pi^\star} = M^{\pi^\star}\widehat{Q}_{\mathrm{pe}}^{\pi^\star} \geq M^{\pi^\star}\left(r + \gamma P\widehat{V}_{\mathrm{pe}}^{\pi^\star} - \gamma\tilde{b}\right) = r_{\pi^\star} + \gamma P_{\pi^\star}\widehat{V}_{\mathrm{pe}}^{\pi^\star} - \gamma\tilde{b}_{\pi^\star}.$$

We also have $\widehat{V}_{\mathrm{pe}}^{\pi^\star} - \gamma P_{\pi^\star}\widehat{V}_{\mathrm{pe}}^{\pi^\star} + \gamma\tilde{b}_{\pi^\star} \geq r_{\pi^\star}$, which will be needed later. By the Bellman equation for $\pi^\star$ we also have that $V^{\pi^\star} = r_{\pi^\star} + \gamma P_{\pi^\star}V^{\pi^\star}$. Combining these, rearranging, and using the monotonicity of multiplication by $(I - \gamma P_{\pi^\star})^{-1}$ (since all its entries are nonnegative), we obtain

$$V^{\pi^\star} - \widehat{V}_{\mathrm{pe}}^{\pi^\star} \leq r_{\pi^\star} + \gamma P_{\pi^\star}V^{\pi^\star} - r_{\pi^\star} + \gamma\tilde{b}_{\pi^\star} - \gamma P_{\pi^\star}\widehat{V}_{\mathrm{pe}}^{\pi^\star} = \gamma\tilde{b}_{\pi^\star} + \gamma P_{\pi^\star}(V^{\pi^\star} - \widehat{V}_{\mathrm{pe}}^{\pi^\star})$$

$$\implies (I - \gamma P_{\pi^\star})(V^{\pi^\star} - \widehat{V}_{\mathrm{pe}}^{\pi^\star}) \leq \gamma\tilde{b}_{\pi^\star}$$

$$\implies V^{\pi^\star} - \widehat{V}_{\mathrm{pe}}^{\pi^\star} \leq (I - \gamma P_{\pi^\star})^{-1}\gamma\tilde{b}_{\pi^\star}$$

as desired. □

**Lemma B.16.** *Fix a deterministic unichain policy $\pi^\star$. Suppose that for all $s \in \mathcal{S}$, $n(s, \pi^\star(s)) \geq m\mu^{\pi^\star}(s) + 4 + 4T_{\mathrm{hit}}(P, \pi^\star)$, $\frac{1}{1-\gamma} \geq m$, and $\frac{1}{1-\gamma} \geq 2$. Then under the event in Lemma B.8, we have that*

$$\max_{s_0 \in \mathcal{S}}\left(V^{\pi^\star}(s_0) - \widehat{V}_{\mathrm{pe}}^{\pi^\star}(s_0)\right)$$

$$\leq \frac{1}{1-\gamma}\sqrt{\frac{2048S\left\|\widehat{V}_{\mathrm{pe}}^{\pi^\star}\right\|_{\mathrm{span}}\log\left(\frac{6S^2 An_{\mathrm{tot}}}{(1-\gamma)\delta}\right)}{m}} + \frac{640S\left\|\widehat{V}_{\mathrm{pe}}^{\pi^\star}\right\|_{\mathrm{span}}\log\left(\frac{6S^2 An_{\mathrm{tot}}}{(1-\gamma)\delta}\right)}{(1-\gamma)m} + \frac{12}{(1-\gamma)n_{\mathrm{tot}}}.$$

*Proof.* First we note that, using Lemma B.15, we have

$$\max_{s_0 \in \mathcal{S}} \left( V^{\pi^\star}(s_0) - \widehat{V}_{\mathrm{pe}}^{\pi^\star}(s_0) \right) \leq \max_{s_0 \in \mathcal{S}} e_{s_0}^\top (I - \gamma P_{\pi^\star})^{-1} \gamma \tilde{b}_{\pi^\star} = \max_{s_0 \in \mathcal{S}} \left\langle d_{\gamma,s_0}^{\pi^\star}, \tilde{b}_{\pi^\star} \right\rangle.$$

We will now fix some arbitrary $s_0 \in \mathcal{S}$ and try to bound $\left\langle d_{\gamma,s_0}^{\pi^\star}, \tilde{b}_{\pi^\star} \right\rangle$. By the assumptions in the lemma statement we have that for all $s \in \mathcal{S}$,

$$n(s, \pi^\star(s)) \geq m\mu^{\pi^\star}(s) + 4T_{\mathrm{hit}}(P, \pi^\star) = (1 - \gamma)m \frac{1}{1 - \gamma} \mu^{\pi^\star}(s) + 4T_{\mathrm{hit}}(P, \pi^\star)$$

$$\geq (1 - \gamma)m d_{\gamma,s_0}^{\pi^\star}(s) - (1 - \gamma)m \left| d_{\gamma,s_0}^{\pi^\star}(s) - \frac{1}{1 - \gamma} \mu^{\pi^\star}(s) \right| + 4T_{\mathrm{hit}}(P, \pi^\star)$$

$$\geq (1 - \gamma)m d_{\gamma,s_0}^{\pi^\star}(s) - (1 - \gamma)m 4T_{\mathrm{hit}}(P, \pi^\star) + 4T_{\mathrm{hit}}(P, \pi^\star)$$

$$\geq (1 - \gamma)m d_{\gamma,s_0}^{\pi^\star}(s)$$

where the third inequality is a consequence of Lemma B.12. For convenience we will let $C := (1 - \gamma)m$, and so we have shown that $n(s, \pi^\star(s)) \geq C d_{\gamma,s_0}^{\pi^\star}(s)$ for all $s \in \mathcal{S}$. Also for convenience abbreviate $\ell = \log\left( \frac{6S^2 A n_{\mathrm{tot}}}{(1-\gamma)\delta} \right)$. Using the fact that $n(s, \pi^\star(s)) \geq 4$ which implies $\frac{1}{\max\{n(s,\pi^\star(s))-1,1\}} = \frac{1}{n(s,\pi^\star(s))-1} \leq \frac{4/3}{n(s,\pi^\star(s))} \leq \frac{2}{n(s,\pi^\star(s))}$, we can simplify $\tilde{b}_{\pi^\star}$ as

$$\tilde{b}_{\pi^\star}(s) = 2\sqrt{\beta(s, \pi^\star(s)) \mathbb{V}_{P_{s\pi^\star(s)}}\left[\widehat{V}_{\mathrm{pe}}^{\pi^\star}\right]} + 4\beta(s, \pi^\star(s)) \left\|\widehat{V}_{\mathrm{pe}}^{\pi^\star}\right\|_{\mathrm{span}} + \frac{12}{n}$$

$$= 2\sqrt{\frac{8\ell}{n(s, \pi^\star(s)) - 1} \mathbb{V}_{P_{s\pi^\star(s)}}\left[\widehat{V}_{\mathrm{pe}}^{\pi^\star}\right]} + 4\frac{8\ell}{n(s, \pi^\star(s)) - 1} \left\|\widehat{V}_{\mathrm{pe}}^{\pi^\star}\right\|_{\mathrm{span}} + \frac{12}{n_{\mathrm{tot}}}$$

$$\leq 2\sqrt{\frac{16\ell}{n(s, \pi^\star(s))} \mathbb{V}_{P_{s\pi^\star(s)}}\left[\widehat{V}_{\mathrm{pe}}^{\pi^\star}\right]} + 4\frac{16\ell}{n(s, \pi^\star(s))} \left\|\widehat{V}_{\mathrm{pe}}^{\pi^\star}\right\|_{\mathrm{span}} + \frac{12}{n_{\mathrm{tot}}}.$$

Using this and the fact that $n(s, \pi^\star(s)) \geq C d_{\gamma,s_0}^{\pi^\star}(s)$ for all $s \in \mathcal{S}$, we have

$$\left\langle d_{\gamma,s_0}^{\pi^\star}, \tilde{b}_{\pi^\star} \right\rangle \leq \sum_{s \in \mathcal{S}} d_{\gamma,s_0}^{\pi^\star}(s) \left( 2\sqrt{\frac{16\ell}{n(s, \pi^\star(s))} \mathbb{V}_{P_{s\pi^\star(s)}}\left[\widehat{V}_{\mathrm{pe}}^{\pi^\star}\right]} + 4\frac{16\ell}{n(s, \pi^\star(s))} \left\|\widehat{V}_{\mathrm{pe}}^{\pi^\star}\right\|_{\mathrm{span}} + \frac{12}{n_{\mathrm{tot}}} \right)$$

$$\leq \sum_{s \in \mathcal{S}} d_{\gamma,s_0}^{\pi^\star}(s) \left( 2\sqrt{\frac{16\ell}{C d_{\gamma,s_0}^{\pi^\star}(s)} \mathbb{V}_{P_{s\pi^\star(s)}}\left[\widehat{V}_{\mathrm{pe}}^{\pi^\star}\right]} + 4\frac{16\ell}{C d_{\gamma,s_0}^{\pi^\star}(s)} \left\|\widehat{V}_{\mathrm{pe}}^{\pi^\star}\right\|_{\mathrm{span}} + \frac{12}{n_{\mathrm{tot}}} \right)$$

$$= \sum_{s \in \mathcal{S}} 2\sqrt{d_{\gamma,s_0}^{\pi^\star}(s) \frac{16\ell}{C} \mathbb{V}_{P_{s\pi^\star(s)}}\left[\widehat{V}_{\mathrm{pe}}^{\pi^\star}\right]} + S4\frac{16\ell}{C} \left\|\widehat{V}_{\mathrm{pe}}^{\pi^\star}\right\|_{\mathrm{span}} + \sum_{s \in \mathcal{S}} d_{\gamma,s_0}^{\pi^\star}(s) \frac{12}{n_{\mathrm{tot}}}$$

$$\leq \sqrt{\frac{64S\ell}{C}} \sqrt{\sum_{s \in \mathcal{S}} d_{\gamma,s_0}^{\pi^\star}(s) \mathbb{V}_{P_{s\pi^\star(s)}}\left[\widehat{V}_{\mathrm{pe}}^{\pi^\star}\right]} + \frac{64S\ell}{C} \left\|\widehat{V}_{\mathrm{pe}}^{\pi^\star}\right\|_{\mathrm{span}} + \frac{12}{(1 - \gamma)n_{\mathrm{tot}}} \tag{35}$$

where in the final inequality we used Cauchy-Schwarz to bound the first term.

Now we focus on bounding the quantity $\sum_{s \in \mathcal{S}} d_{\gamma,s_0}^{\pi^\star}(s) \mathbb{V}_{P_{s\pi^\star(s)}}\left[\widehat{V}_{\mathrm{pe}}^{\pi^\star}\right]$. Let $c = \min_{s \in \mathcal{S}} \widehat{V}_{\mathrm{pe}}^{\pi^\star}(s)$ and $\overline{V} = \widehat{V}_{\mathrm{pe}}^{\pi^\star} - c\mathbf{1}$. Then

$$\overline{V} \circ \overline{V} - \gamma^2 P_{\pi^\star}\overline{V} \circ P_{\pi^\star}\overline{V} = (\overline{V} - \gamma P_{\pi^\star}\overline{V}) \circ (\overline{V} + \gamma P_{\pi^\star}\overline{V})$$

$$\leq (\overline{V} - \gamma P_{\pi^\star}\overline{V} + \gamma\tilde{b}_{\pi^\star} + (1 - \gamma)c\mathbf{1}) \circ (\overline{V} + \gamma P_{\pi^\star}\overline{V})$$

$$\leq 2\left\|\overline{V}\right\|_\infty (\overline{V} - \gamma P_{\pi^\star}\overline{V} + \gamma\tilde{b}_{\pi^\star} + (1 - \gamma)c\mathbf{1}) \tag{36}$$

where for the first inequality we used that $\overline{V} + \gamma P_{\pi^\star}\overline{V} \geq \mathbf{0}$ and that $\tilde{b}_{\pi^\star} + (1 - \gamma)c\mathbf{1} \geq \mathbf{0}$, and for the second inequality we used that $\overline{V} + \gamma P_{\pi^\star}\overline{V} \leq 2\left\|\overline{V}\right\|_\infty \mathbf{1}$ and that $\overline{V} - \gamma P_{\pi^\star}\overline{V} + \gamma\tilde{b}_{\pi^\star} + (1 - \gamma)c\mathbf{1} \geq \mathbf{0}$, which follows from the fact that

$$\overline{V} - \gamma P_{\pi^\star}\overline{V} + \gamma\tilde{b}_{\pi^\star} + (1 - \gamma)c\mathbf{1} = \widehat{V}_{\mathrm{pe}}^{\pi^\star} - \gamma P_{\pi^\star}\widehat{V}_{\mathrm{pe}}^{\pi^\star} + \gamma\tilde{b}_{\pi^\star} \geq r_{\pi^\star} \geq \mathbf{0}$$

using (29) in the inequality step. Thus

$$
\left\langle d_{\gamma,s_0}^{\pi^\star}, \mathbb{V}_{P_{\pi^\star}}\left[\widehat{V}_{\mathrm{pe}}^{\pi^\star}\right]\right\rangle = \left\langle d_{\gamma,s_0}^{\pi^\star}, \mathbb{V}_{P_{\pi^\star}}\left[\overline{V}\right]\right\rangle
$$

$$
= \left\langle d_{\gamma,s_0}^{\pi^\star}, P_{\pi^\star}(\overline{V})^{\circ 2} - (P_{\pi^\star}\overline{V})^{\circ 2}\right\rangle
$$

$$
= \left\langle d_{\gamma,s_0}^{\pi^\star}, P_{\pi^\star}(\overline{V})^{\circ 2} - \frac{1}{\gamma^2}(\overline{V})^{\circ 2} + \frac{1}{\gamma^2}\left((\overline{V})^{\circ 2} - \gamma^2(P_{\pi^\star}\overline{V})^{\circ 2}\right)\right\rangle
$$

$$
\overset{(i)}{\leq} \left\langle d_{\gamma,s_0}^{\pi^\star}, P_{\pi^\star}(\overline{V})^{\circ 2} - \frac{1}{\gamma^2}(\overline{V})^{\circ 2} + \frac{1}{\gamma^2}2\left\|\overline{V}\right\|_\infty \left(\overline{V} - \gamma P_{\pi^\star}\overline{V} + \gamma \tilde{b}_{\pi^\star} + (1-\gamma)c\mathbf{1}\right)\right\rangle
$$

$$
\overset{(ii)}{\leq} \left\langle d_{\gamma,s_0}^{\pi^\star}, \frac{1}{\gamma^2}2\left\|\overline{V}\right\|_\infty \left(\overline{V} - \gamma P_{\pi^\star}\overline{V} + \gamma \tilde{b}_{\pi^\star} + (1-\gamma)c\mathbf{1}\right)\right\rangle
$$

$$
= \frac{2\left\|\overline{V}\right\|_\infty}{\gamma^2} e_{s_0}^\top (I - \gamma P_{\pi^\star})^{-1}\left((I - \gamma P_{\pi^\star})\overline{V} + \gamma \tilde{b}_{\pi^\star} + (1-\gamma)c\mathbf{1}\right)
$$

$$
= \frac{2\left\|\overline{V}\right\|_\infty}{\gamma^2} e_{s_0}^\top (I - \gamma P_{\pi^\star})^{-1}\left((I - \gamma P_{\pi^\star})\widehat{V}_{\mathrm{pe}}^\star + \gamma \tilde{b}_{\pi^\star}\right)
$$

$$
= \frac{2\left\|\overline{V}\right\|_\infty}{\gamma^2} e_{s_0}^\top \widehat{V}_{\mathrm{pe}}^\star + \frac{4\left\|\overline{V}\right\|_\infty}{\gamma^2}\langle d_{\gamma,s_0}^{\pi^\star}, \gamma \tilde{b}_{\pi^\star}\rangle
$$

$$
\leq \frac{2\left\|\overline{V}\right\|_\infty}{\gamma^2}\frac{1}{1-\gamma} + \frac{4\left\|\overline{V}\right\|_\infty}{\gamma}\langle d_{\gamma,s_0}^{\pi^\star}, \tilde{b}_{\pi^\star}\rangle. \tag{37}
$$

In $(i)$ we use (36) and in $(ii)$ we use that

$$
\left\langle d_{\gamma,s_0}^{\pi^\star}, P_{\pi^\star}(\overline{V})^{\circ 2} - \frac{1}{\gamma^2}(\overline{V})^{\circ 2}\right\rangle \leq \left\langle d_{\gamma,s_0}^{\pi^\star}, P_{\pi^\star}(\overline{V})^{\circ 2} - \frac{1}{\gamma}(\overline{V})^{\circ 2}\right\rangle
$$

$$
= \frac{1}{\gamma}e_{s_0}^\top (I - \gamma P_{\pi^\star})^{-1}(\gamma P_{\pi^\star} - I)(\overline{V})^{\circ 2} \leq 0.
$$

Combining the bound (37) with (35) (and noting that $\left\|\overline{V}\right\|_\infty = \left\|\widehat{V}_{\mathrm{pe}}^{\pi^\star}\right\|_{\mathrm{span}}$), we obtain that

$$
\left\langle d_{\gamma,s_0}^{\pi^\star}, \tilde{b}_{\pi^\star}\right\rangle \leq \sqrt{\frac{64S\ell}{C}}\sqrt{\frac{2\left\|\widehat{V}_{\mathrm{pe}}^{\pi^\star}\right\|_{\mathrm{span}}}{\gamma^2}\frac{1}{1-\gamma} + \frac{4\left\|\widehat{V}_{\mathrm{pe}}^{\pi^\star}\right\|_{\mathrm{span}}}{\gamma}\left\langle d_{\gamma,s_0}^{\pi^\star}, \tilde{b}_{\pi^\star}\right\rangle}
$$

$$
+ \frac{64S\ell}{C}\left\|\widehat{V}_{\mathrm{pe}}^{\pi^\star}\right\|_{\mathrm{span}} + \frac{12}{(1-\gamma)n_{\mathrm{tot}}}
$$

$$
\leq \sqrt{\frac{512\left\|\widehat{V}_{\mathrm{pe}}^{\pi^\star}\right\|_{\mathrm{span}}S\ell}{C}}\left(\sqrt{\frac{1}{1-\gamma}} + \sqrt{\left\langle d_{\gamma,s_0}^{\pi^\star}, \tilde{b}_{\pi^\star}\right\rangle}\right)
$$

$$
+ \frac{64S\ell}{C}\left\|\widehat{V}_{\mathrm{pe}}^{\pi^\star}\right\|_{\mathrm{span}} + \frac{12}{(1-\gamma)n_{\mathrm{tot}}}
$$

where we simplified by using that $\sqrt{a+b} \leq \sqrt{a} + \sqrt{b}$ and that $\frac{1}{\gamma} \leq 2$ (since $\frac{1}{1-\gamma} \geq 2$ implies that $\gamma \geq \frac{1}{2}$). The above is a quadratic inequality in $x := \sqrt{\left\langle d_{\gamma,s_0}^{\pi^\star}, \tilde{b}_{\pi^\star}\right\rangle}$ of the form

$$
x^2 \leq x\sqrt{8y} + \sqrt{\frac{8y}{1-\gamma}} + y + \frac{12}{(1-\gamma)n_{\mathrm{tot}}}
$$

where $y = \frac{64S\left\|\widehat{V}_{\mathrm{pe}}^{\pi^\star}\right\|_{\mathrm{span}}\ell}{C}$. From the quadratic formula we obtain that

$$
x \leq \frac{\sqrt{8y} + \sqrt{8y + 4\left(\sqrt{\frac{8y}{1-\gamma}} + y + \frac{12}{(1-\gamma)n_{\mathrm{tot}}}\right)}}{2}
$$

and then squaring both sides we obtain that

$$
\begin{aligned}
\left\langle d_{\gamma,s_0}^{\pi^\star}, \tilde{b}_{\pi^\star} \right\rangle = x^2 &\leq \frac{\left( \sqrt{8y} + \sqrt{8y + 4\left( \sqrt{\frac{8y}{1-\gamma}} + y + \frac{12}{(1-\gamma)n_{\text{tot}}} \right)} \right)^2}{4} \\
&\leq \frac{1}{2}\left( 8y + 8y + 4\left( \sqrt{\frac{8y}{1-\gamma}} + y + \frac{12}{(1-\gamma)n_{\text{tot}}} \right) \right) \\
&= 10y + \sqrt{\frac{32y}{1-\gamma}} + \frac{12}{(1-\gamma)n_{\text{tot}}} \\
&= 10\frac{64S\left\| \widehat{V}_{\text{pe}}^{\pi^\star} \right\|_{\text{span}} \ell}{C} + \sqrt{32\frac{64S\left\| \widehat{V}_{\text{pe}}^{\pi^\star} \right\|_{\text{span}} \ell}{C(1-\gamma)}} + \frac{12}{(1-\gamma)n_{\text{tot}}}
\end{aligned}
$$

using that $(a+b)^2 \leq 2a^2 + 2b^2$. Recalling the definitions of $C = (1-\gamma)m$ and $\ell = \log\left( \frac{6S^2An_{\text{tot}}}{(1-\gamma)\delta} \right)$, and also since the above bound held for arbitrary $s_0$, we have thus shown that

$$
\begin{aligned}
&\max_{s_0 \in \mathcal{S}} \left( V^{\pi^\star}(s_0) - \widehat{V}_{\text{pe}}^{\pi^\star}(s_0) \right) \\
&\leq \frac{1}{1-\gamma}\sqrt{\frac{2048S\left\| \widehat{V}_{\text{pe}}^{\pi^\star} \right\|_{\text{span}} \log\left( \frac{6S^2An_{\text{tot}}}{(1-\gamma)\delta} \right)}{m}} + \frac{640S\left\| \widehat{V}_{\text{pe}}^{\pi^\star} \right\|_{\text{span}} \log\left( \frac{6S^2An_{\text{tot}}}{(1-\gamma)\delta} \right)}{(1-\gamma)m} + \frac{12}{(1-\gamma)n_{\text{tot}}}.
\end{aligned}
$$

$\square$

### B.7 Controlling the empirical span

While Lemma B.16 is approaching the desired result, it involves the empirical span term $\left\| \widehat{V}_{\text{pe}}^{\pi^\star} \right\|_{\text{span}}$ which we would like to bound in terms of $\left\| V^{\pi^\star} \right\|_{\text{span}}$. Such a bound is the objective of this subsection, and makes crucial use of our assumption of data even for states which are transient under $P_{\pi^\star}$.

**Lemma B.17.** *Fix a deterministic unichain policy $\pi^\star$. Suppose that $n(s, \pi^\star(s)) \geq 72(T_{\text{hit}}(P, \pi^\star))^2 \log\left( \frac{2S}{\delta} \right)$ for all $s \in \mathcal{S}$. Then with probability at least $1 - \delta$,*

$$
T_{\text{hit}}(\widehat{P}, \pi^\star) \leq 24T_{\text{hit}}(P, \pi^\star).
$$

*Proof.* The proof of this lemma is inspired by that of Zurek and Chen [2024, Lemma 4]. For any MDP $\mathcal{M}$ and $s \in \mathcal{S}$ we let $E_{s_0,\mathcal{M}}^\pi$ denote the expectation with respect to the Markov chain induced by $\pi$ in the MDP $\mathcal{M}$ from starting state $s_0$, and similarly we let $\mathbb{P}_{s_0,\mathcal{M}}^\pi(E) = \mathbb{E}_{s_0,\mathcal{M}}^\pi[\mathbb{I}(E)]$ denote the associated probability measure. Let $s^\star \in \mathcal{S}$ satisfy $T_{\text{hit}}(P, \pi^\star) = T_{\text{hit}}(P, \pi, s^\star)$. Let $\widehat{\mathcal{M}}$ be the MDP $(\widehat{P}, r)$. Then

$$
T_{\text{hit}}(\widehat{P}, \pi^\star) \leq T_{\text{hit}}(\widehat{P}, \pi^\star, s^\star) = \max_{s_0 \in \mathcal{S}} \mathbb{E}_{s_0,\widehat{\mathcal{M}}}^\pi[\eta_{s^\star}]. \tag{38}
$$

Supposing that $k \in \mathbb{N}$ satisfies $\max_{s_0 \in \mathcal{S}} \mathbb{P}_{s_0, \widehat{\mathcal{M}}}(\eta_{s^\star} \geq k) \leq \frac{1}{2}$, then we have for any $s_0'$ that

$$
\begin{aligned}
\mathbb{E}^\pi_{s_0', \widehat{\mathcal{M}}}[\eta_{s^\star}] &= \sum_{t=0}^{\infty} \mathbb{P}_{s_0', \widehat{\mathcal{M}}}(\eta_{s^\star} > t) \\
&= \sum_{i=0}^{\infty} \sum_{t=0}^{k-1} \mathbb{P}_{s_0', \widehat{\mathcal{M}}}(\eta_{s^\star} > ik + t) \\
&\leq \sum_{i=0}^{\infty} \sum_{t=0}^{k-1} \mathbb{P}_{s_0', \widehat{\mathcal{M}}}(\eta_{s^\star} > ik) \\
&= k \sum_{i=0}^{\infty} \mathbb{P}_{s_0', \widehat{\mathcal{M}}}(\eta_{s^\star} > ik) \\
&\leq k \sum_{i=0}^{\infty} 2^{-i} = 2k
\end{aligned}
\tag{39}
$$

where the final inequality step used that

$$
\mathbb{P}_{s_0', \widehat{\mathcal{M}}}(\eta_{s^\star} > ik) \leq \left( \max_{s_0 \in \mathcal{S}} \mathbb{P}_{s_0, \widehat{\mathcal{M}}}(\eta_{s^\star} > k) \right)^i \leq 2^{-i}
$$

which follows from the following standard arguments: for any integer $i \geq 1$ (since this formula obviously holds for $i = 0$), we have

$$
\begin{aligned}
&\mathbb{P}_{s_0', \widehat{\mathcal{M}}}(\eta_{s^\star} > ik) \\
&\leq \mathbb{P}_{s_0', \widehat{\mathcal{M}}}(\eta_{s^\star} \geq ik) \\
&= \mathbb{P}_{s_0', \widehat{\mathcal{M}}} \left( \eta_{s^\star} \notin \{0, \ldots, (i-1)k - 1\} \text{ and } \eta_{s^\star} \notin \{(i-1)k, \ldots, ik - 1\} \right) \\
&\overset{(i)}{=} \mathbb{E}_{s_0', \widehat{\mathcal{M}}} \mathbb{P}_{s_0', \widehat{\mathcal{M}}} \left( \eta_{s^\star} \notin \{0, \ldots, (i-1)k - 1\} \text{ and } \eta_{s^\star} \notin \{(i-1)k, \ldots, ik - 1\} \mid \mathcal{F}_{(i-1)k} \right) \\
&\overset{(ii)}{=} \mathbb{E}_{s_0', \widehat{\mathcal{M}}} \left[ \mathbb{I}\left( \eta_{s^\star} \notin \{0, \ldots, (i-1)k - 1\} \right) \mathbb{P}_{s_0', \widehat{\mathcal{M}}} \left( \eta_{s^\star} \notin \{(i-1)k, \ldots, ik - 1\} \mid \mathcal{F}_{(i-1)k} \right) \right] \\
&\overset{(iii)}{=} \mathbb{E}_{s_0', \widehat{\mathcal{M}}} \left[ \mathbb{I}\left( \eta_{s^\star} \notin \{0, \ldots, (i-1)k - 1\} \right) \mathbb{P}_{S_k, \widehat{\mathcal{M}}} \left( \eta_{s^\star} \notin \{0, \ldots, k - 1\} \right) \right] \\
&\overset{(iv)}{\leq} \frac{1}{2} \mathbb{E}_{s_0', \widehat{\mathcal{M}}} \left[ \mathbb{I}\left( \eta_{s^\star} \notin \{0, \ldots, (i-1)k - 1\} \right) \right] \\
&= \frac{1}{2} \mathbb{P}_{s_0', \widehat{\mathcal{M}}}(\eta_{s^\star} \geq (i-1)k)
\end{aligned}
$$

where $\mathcal{F}_{(i-1)k}$ is the sigma-algebra generated by $S_0, \ldots, S_{(i-1)k}$, step $(i)$ is the tower property, step $(ii)$ is because the event $\eta_{s^\star} \notin \{0, \ldots, (i-1)k - 1\}$ is $\mathcal{F}_{(i-1)k}$-measurable, step $(iii)$ is the Markov property (e.g., [Durrett, 2019, Theorem 5.2.3]), and step $(iv)$ is because $\mathbb{P}_{S_k, \widehat{\mathcal{M}}}(\eta_{s^\star} \notin \{0, \ldots, k - 1\}) = \mathbb{P}_{S_k, \widehat{\mathcal{M}}}(\eta_{s^\star} \geq k) \leq \frac{1}{2}$ (this last inequality holding almost surely, due to the assumption that $\max_{s_0 \in \mathcal{S}} \mathbb{P}_{s_0, \widehat{\mathcal{M}}}(\eta_{s^\star} \geq k) \leq \frac{1}{2}$). Since these arguments held for arbitrary $i$, we can repeat them to obtain the desired bound.

Now we try to find such a $k$. Define the reward function $\bar{r}$ by $\bar{r}(s, a) = \mathbb{I}(s \neq s^\star)$ and also let $P'$ be the same transition matrix as $P$ except with state $s^\star$ made to be absorbing for all actions. Then, for some $\bar{\gamma}$ to be chosen later, letting $V^{\pi^\star}_{\bar{\gamma}, \mathcal{M}'}$ be the discounted value function for policy $\pi^\star$ in MDP $\mathcal{M}' = (P', \bar{r})$, and letting $\mathbb{E}^{\pi^\star}_{s_0, \mathcal{M}'}, \mathbb{E}^{\pi^\star}_{s_0, \mathcal{M}}$ denote expectations with respect to the MDPs $\mathcal{M}'$ and $\mathcal{M}$

respectively, we have that

$$V^{\pi^\star}_{\overline{\gamma},\mathcal{M}'}(s_0) = \mathbb{E}^{\pi^\star}_{s_0,\mathcal{M}'} \sum_{t=0}^{\infty} \overline{\gamma}^t \mathbb{I}(S_t \neq s^\star)$$

$$= \mathbb{E}^{\pi^\star}_{s_0,\mathcal{M}} \sum_{t=0}^{\infty} \overline{\gamma}^t \mathbb{I}(\eta_{s^\star} > t)$$

$$\leq \mathbb{E}^{\pi^\star}_{s_0,\mathcal{M}} \sum_{t=0}^{\infty} \mathbb{I}(\eta_{s^\star} > t)$$

$$= \mathbb{E}^{\pi^\star}_{s_0,\mathcal{M}}[\eta_{s^\star}] \leq T_{\mathrm{hit}}(P,\pi^\star,s^\star).$$

This implies $\left\| V^{\pi^\star}_{\overline{\gamma},\mathcal{M}'} \right\|_{\mathrm{span}} \leq T_{\mathrm{hit}}(P,\pi^\star,s^\star)$, which will be needed shortly.

Let $\widehat{P}'$ similarly be the same transition matrix as $\widehat{P}$ except $s^\star$ is absorbing for all actions. Let $\widehat{\mathcal{M}}'$ be the MDP $(\widehat{P}',\overline{r})$. Then for any $k \in \mathbb{N}$ we have

$$V^{\pi^\star}_{\overline{\gamma},\widehat{\mathcal{M}}'}(s_0) = \mathbb{E}^{\pi^\star}_{s_0,\widehat{\mathcal{M}}'} \sum_{t=0}^{\infty} \overline{\gamma}^t \mathbb{I}(S_t \neq s^\star)$$

$$= \mathbb{E}^{\pi^\star}_{s_0,\widehat{\mathcal{M}}} \sum_{t=0}^{\infty} \overline{\gamma}^t \mathbb{I}(\eta_{s^\star} > t)$$

$$\geq \mathbb{E}^{\pi^\star}_{s_0,\widehat{\mathcal{M}}} \sum_{t=0}^{k-1} \overline{\gamma}^t \mathbb{I}(\eta_{s^\star} > t)$$

$$\geq \mathbb{E}^{\pi^\star}_{s_0,\widehat{\mathcal{M}}} \sum_{t=0}^{k-1} \overline{\gamma}^{k-1} \mathbb{I}(\eta_{s^\star} > k-1)$$

$$= k\overline{\gamma}^{k-1} \mathbb{P}_{s_0,\widehat{\mathcal{M}}}(\eta_{s^\star} > k-1).$$

Rearranging this implies that

$$\mathbb{P}_{s_0,\widehat{\mathcal{M}}}(\eta_{s^\star} > k-1) \leq \frac{V^{\pi^\star}_{\overline{\gamma},\widehat{\mathcal{M}}'}(s_0)}{k\overline{\gamma}^{k-1}} \leq \frac{3V^{\pi^\star}_{\overline{\gamma},\widehat{\mathcal{M}}'}(s_0)}{k} \tag{40}$$

where for the second inequality we set $\overline{\gamma} = 1 - \frac{1}{k}$ and used the fact that $(1 - \frac{1}{k})^{k-1} \geq 1/e \geq 1/3$ for all integers $k > 1$.

Now we bound $V^{\pi^\star}_{\overline{\gamma},\widehat{\mathcal{M}}'}(s_0)$ using concentration inequalities. For concreteness in the following application of Hoeffding we set $k = 12T_{\mathrm{hit}}(P,\pi^\star)$ so $\gamma = 1 - 1/(12T_{\mathrm{hit}}(P,\pi^\star))$. By Hoeffding's inequality, we have for any $s \neq s^\star$ that with probability at least $1 - \delta'$

$$\left| e_s^\top (\widehat{P}'_{\pi^\star} - P'_{\pi^\star})V^{\pi^\star}_{\overline{\gamma},\mathcal{M}'} \right| \leq \sqrt{\frac{\left\| V^{\pi^\star}_{\overline{\gamma},\mathcal{M}'} \right\|^2_{\mathrm{span}} \log\left(\frac{2}{\delta'}\right)}{2n(s,\pi^\star(s))}} \leq \sqrt{\frac{(T_{\mathrm{hit}}(P,\pi^\star,s^\star))^2 \log\left(\frac{2}{\delta'}\right)}{2n(s,\pi^\star(s))}}$$

and trivially we have $\left| e_{s^\star}^\top (\widehat{P}'_{\pi^\star} - P'_{\pi^\star})V^{\pi^\star}_{\overline{\gamma},\mathcal{M}'} \right| = 0$. Therefore by a union bound over all $s \in \mathcal{S}$ and setting $\delta' = \frac{\delta}{S}$, we have with probability at least $1 - \delta$ that

$$\left\| (\widehat{P}'_{\pi^\star} - P'_{\pi^\star})V^{\pi^\star}_{\overline{\gamma},\mathcal{M}'} \right\|_{\infty} \leq \min_{s \in \mathcal{S}} \sqrt{\frac{(T_{\mathrm{hit}}(P,\pi^\star,s^\star))^2 \log\left(\frac{2S}{\delta}\right)}{2n(s,\pi^\star(s))}} \leq \frac{1}{12}$$

where the second inequality uses the condition that $n(s,\pi^\star(s)) \geq \frac{12^2}{2}(T_{\mathrm{hit}}(P,\pi^\star,s^\star))^2 \log\left(\frac{2S}{\delta}\right) = 72(T_{\mathrm{hit}}(P,\pi^\star))^2 \log\left(\frac{2S}{\delta}\right)$ for all $s \in \mathcal{S}$.

Following standard arguments for the difference between two value functions with different transition matrices we have

$$
\begin{aligned}
V^{\pi^\star}_{\overline{\gamma},\widehat{\mathcal{M}}'} - V^{\pi^\star}_{\overline{\gamma},\mathcal{M}'} &= (I - \overline{\gamma}\widehat{P}'_{\pi^\star})^{-1}\overline{r}_{\pi^\star} - (I - \overline{\gamma}P'_{\pi^\star})^{-1}\overline{r}_{\pi^\star} \\
&= (I - \overline{\gamma}\widehat{P}'_{\pi^\star})^{-1}(I - \overline{\gamma}P'_{\pi^\star})(I - \overline{\gamma}P'_{\pi^\star})^{-1}\overline{r}_{\pi^\star} - (I - \overline{\gamma}\widehat{P}'_{\pi^\star})^{-1}(I - \overline{\gamma}\widehat{P}'_{\pi^\star})(I - \overline{\gamma}P'_{\pi^\star})^{-1}\overline{r}_{\pi^\star} \\
&= \overline{\gamma}(I - \overline{\gamma}\widehat{P}'_{\pi^\star})^{-1}(\widehat{P}'_{\pi^\star} - P'_{\pi^\star})(I - \overline{\gamma}P'_{\pi^\star})^{-1}\overline{r}_{\pi^\star} \\
&= \overline{\gamma}(I - \overline{\gamma}\widehat{P}'_{\pi^\star})^{-1}(\widehat{P}'_{\pi^\star} - P'_{\pi^\star})V^{\pi^\star}_{\overline{\gamma},\mathcal{M}'}.
\end{aligned}
$$

Hence

$$
\begin{aligned}
\left\| V^{\pi^\star}_{\overline{\gamma},\widehat{\mathcal{M}}'} - V^{\pi^\star}_{\overline{\gamma},\mathcal{M}'} \right\|_\infty &= \left\| \overline{\gamma}(I - \overline{\gamma}\widehat{P}'_{\pi^\star})^{-1}(\widehat{P}'_{\pi^\star} - P'_{\pi^\star})V^{\pi^\star}_{\overline{\gamma},\mathcal{M}'} \right\|_\infty \\
&\leq \left\| \overline{\gamma}(I - \overline{\gamma}\widehat{P}'_{\pi^\star})^{-1} \right\|_{\infty\to\infty} \left\| (\widehat{P}'_{\pi^\star} - P'_{\pi^\star})V^{\pi^\star}_{\overline{\gamma},\mathcal{M}'} \right\|_\infty \\
&\leq \frac{\overline{\gamma}}{1-\overline{\gamma}}\frac{1}{12} \\
&\leq \frac{k}{12} = T_{\text{hit}}(P,\pi^\star,s^\star).
\end{aligned}
$$

Combining this with (40), we have that

$$
\begin{aligned}
\max_{s_0\in\mathcal{S}} \mathbb{P}_{s_0,\widehat{\mathcal{M}}}(\eta_{s^\star} \geq k) &= \max_{s_0\in\mathcal{S}} \mathbb{P}_{s_0,\widehat{\mathcal{M}}}(\eta_{s^\star} > k-1) \\
&\leq \frac{3\left\| V^{\pi^\star}_{\overline{\gamma},\widehat{\mathcal{M}}'} \right\|_\infty}{k} \leq \frac{3\left\| V^{\pi^\star}_{\overline{\gamma},\mathcal{M}'} \right\|_\infty}{k} + \frac{3\left\| V^{\pi^\star}_{\overline{\gamma},\widehat{\mathcal{M}}'} - V^{\pi^\star}_{\overline{\gamma},\mathcal{M}'} \right\|_\infty}{k} \\
&\leq \frac{3T_{\text{hit}}(P,\pi^\star) + 3T_{\text{hit}}(P,\pi^\star)}{12T_{\text{hit}}(P,\pi^\star)} = \frac{1}{2}.
\end{aligned}
$$

Using $k = 12T_{\text{hit}}(P,\pi^\star)$ in (39) and combining with (38), we conclude that

$$
T_{\text{hit}}(\widehat{P},\pi^\star) \leq \max_{s_0\in\mathcal{S}} \mathbb{E}^{\pi}_{s_0,\widehat{\mathcal{M}}}[\eta_{s^\star}] \leq 2k = 24T_{\text{hit}}(P,\pi^\star)
$$

as desired. $\qquad\square$

**Lemma B.18.** *Fix a deterministic unichain policy $\pi^\star$. Suppose that $n(s,\pi^\star(s)) \geq 1 + \alpha\left(576T_{\text{hit}}(P,\pi^\star)\right)^2$ for all $s\in\mathcal{S}$, where $\alpha = 8\log\left(\frac{6S^2An_{\text{tot}}}{(1-\gamma)\delta}\right)$. Then with probability at least $1-2\delta$,*

$$
\left\| \widehat{V}^{\pi^\star}_{\text{pe}} \right\|_{\text{span}} \leq 3\left\| V^{\pi^\star} \right\|_{\text{span}} + 2.
$$

*Proof.* By the definition (8) of $\widehat{\mathcal{T}}^{\pi^\star}_{\text{pe}}$, we have for any $s\in\mathcal{S}$ that

$$
\begin{aligned}
\widehat{V}^{\pi^\star}_{\text{pe}}(s) &= e_s^\top M^{\pi^\star}\widehat{Q}^{\pi^\star}_{\text{pe}} \\
&= e_s^\top M^{\pi^\star}\widehat{\mathcal{T}}^{\pi^\star}_{\text{pe}}\left(\widehat{Q}^{\pi^\star}_{\text{pe}}\right) \\
&= r(s,\pi^\star(s)) + \gamma\max\Big\{ \widehat{P}_{s\pi^\star(s)}T_{\beta(s,\pi^\star(s))}(\widehat{P}_{s\pi^\star(s)}, M^{\pi^\star}\widehat{Q}^{\pi^\star}_{\text{pe}}) - b(s,\pi^\star(s), M^{\pi^\star}\widehat{Q}^{\pi^\star}_{\text{pe}}), \\
&\qquad\qquad\qquad \min_{s'}(M^{\pi^\star}\widehat{Q}^{\pi^\star}_{\text{pe}})(s') \Big\} \\
&= r_{\pi^\star}(s) + \gamma\max\Big\{ \widehat{P}_{s\pi^\star(s)}T_{\beta(s,\pi^\star(s))}(\widehat{P}_{s\pi^\star(s)}, \widehat{V}^{\pi^\star}_{\text{pe}}) - b(s,\pi^\star(s), \widehat{V}^{\pi^\star}_{\text{pe}}), \min_{s'}(\widehat{V}^{\pi^\star}_{\text{pe}})(s') \Big\} \\
&= r_{\pi^\star}(s) + \gamma\widehat{P}_{s\pi^\star(s)}\widehat{V}^{\pi^\star}_{\text{pe}} + \gamma\max\Big\{ \widehat{P}_{s\pi^\star(s)}\left( T_{\beta(s,\pi^\star(s))}(\widehat{P}_{s\pi^\star(s)}, \widehat{V}^{\pi^\star}_{\text{pe}}) - \widehat{V}^{\pi^\star}_{\text{pe}} \right) - b(s,\pi^\star(s), \widehat{V}^{\pi^\star}_{\text{pe}}), \\
&\qquad\qquad\qquad \min_{s'}(\widehat{V}^{\pi^\star}_{\text{pe}})(s') - \widehat{P}_{s\pi^\star(s)}\widehat{V}^{\pi^\star}_{\text{pe}} \Big\} \\
&= r_{\pi^\star}(s) + \gamma\widehat{P}_{s\pi^\star(s)}\widehat{V}^{\pi^\star}_{\text{pe}} - \gamma\tilde{b}'(s)
\end{aligned}
$$

where we have defined $\tilde{b}' \in \mathbb{R}^{\mathcal{S}}$ as

$$\tilde{b}'(s) = -\max\left\{\widehat{P}_{s\pi^\star(s)}\left(T_{\beta(s,\pi^\star(s))}(\widehat{P}_{s\pi^\star(s)}, \widehat{V}_{\text{pe}}^{\pi^\star}) - \widehat{V}_{\text{pe}}^{\pi^\star}\right) - b(s, \pi^\star(s), \widehat{V}_{\text{pe}}^{\pi^\star}), \min_{s'}(\widehat{V}_{\text{pe}}^{\pi^\star})(s') - \widehat{P}_{s\pi^\star(s)}\widehat{V}_{\text{pe}}^{\pi^\star}\right\}.$$

Note that both terms within the $\max$ in the definition of $\tilde{b}'(s)$ are $\leq 0$, so $\tilde{b}' \geq 0$, and also we can bound

$$\tilde{b}'(s) \leq -\left(\widehat{P}_{s\pi^\star(s)}\left(T_{\beta(s,\pi^\star(s))}(\widehat{P}_{s\pi^\star(s)}, \widehat{V}_{\text{pe}}^{\pi^\star}) - \widehat{V}_{\text{pe}}^{\pi^\star}\right) - b(s, \pi^\star(s), \widehat{V}_{\text{pe}}^{\pi^\star})\right)$$

$$= \widehat{P}_{s\pi^\star(s)}\left(\widehat{V}_{\text{pe}}^{\pi^\star} - T_{\beta(s,\pi^\star(s))}(\widehat{P}_{s\pi^\star(s)}, \widehat{V}_{\text{pe}}^{\pi^\star})\right) + b(s, \pi^\star(s), \widehat{V}_{\text{pe}}^{\pi^\star})$$

$$\overset{(i)}{\leq} \beta(s, \pi^\star(s))\left\|\widehat{V}_{\text{pe}}^{\pi^\star}\right\|_{\text{span}} + b(s, \pi^\star(s), \widehat{V}_{\text{pe}}^{\pi^\star})$$

$$\overset{(ii)}{\leq} \sqrt{\beta(s, \pi^\star(s))\left\|\widehat{V}_{\text{pe}}^{\pi^\star}\right\|_{\text{span}}^2} + 2\beta(s, \pi^\star(s))\left\|\widehat{V}_{\text{pe}}^{\pi^\star}\right\|_{\text{span}} + \frac{5}{n_{\text{tot}}} \tag{41}$$

where $(i)$ is due to the fact that $\widehat{P}_{s\pi^\star(s)}T_{\beta(s,\pi^\star(s))}(\widehat{P}_{s\pi^\star(s)}, \widehat{V}_{\text{pe}}^{\pi^\star}) \geq \widehat{P}_{s\pi^\star(s)}\widehat{V}_{\text{pe}}^{\pi^\star} - \beta(s, \pi^\star(s))\left\|\widehat{V}_{\text{pe}}^{\pi^\star}\right\|_{\text{span}}$, which holds by an argument identical to that of (31), and $(ii)$ holds since

$$b(s, \pi^\star(s), \widehat{V}_{\text{pe}}^{\pi^\star}) = \max\left\{\sqrt{\beta(s, \pi^\star(s))\mathbb{V}_{\widehat{P}_{s\pi^\star(s)}}\left[T_{\beta(s,\pi^\star(s))}(\widehat{P}_{s\pi^\star(s)}, \widehat{V}_{\text{pe}}^{\pi^\star})\right]},\right.$$

$$\left.\beta(s, \pi^\star(s))\left\|T_{\beta(s,\pi^\star(s))}(\widehat{P}_{s\pi^\star(s)}, \widehat{V}_{\text{pe}}^{\pi^\star})\right\|_{\text{span}}\right\} + \frac{5}{n_{\text{tot}}}$$

$$\leq \max\left\{\sqrt{\beta(s, \pi^\star(s))\mathbb{V}_{\widehat{P}_{s\pi^\star(s)}}\left[\widehat{V}_{\text{pe}}^{\pi^\star}\right]}, \beta(s, \pi^\star(s))\left\|\widehat{V}_{\text{pe}}^{\pi^\star}\right\|_{\text{span}}\right\} + \frac{5}{n_{\text{tot}}}$$

$$\leq \max\left\{\sqrt{\beta(s, \pi^\star(s))\left\|\widehat{V}_{\text{pe}}^{\pi^\star}\right\|_{\text{span}}^2}, \beta(s, \pi^\star(s))\left\|\widehat{V}_{\text{pe}}^{\pi^\star}\right\|_{\text{span}}\right\} + \frac{5}{n_{\text{tot}}}$$

$$\leq \sqrt{\beta(s, \pi^\star(s))\left\|\widehat{V}_{\text{pe}}^{\pi^\star}\right\|_{\text{span}}^2} + \beta(s, \pi^\star(s))\left\|\widehat{V}_{\text{pe}}^{\pi^\star}\right\|_{\text{span}} + \frac{5}{n_{\text{tot}}}$$

where we used Lemma B.14 and the fact that $\left\|T_{\beta(s,\pi^\star(s))}(\widehat{P}_{s\pi^\star(s)}, \widehat{V}_{\text{pe}}^{\pi^\star})\right\|_{\text{span}} \leq \left\|\widehat{V}_{\text{pe}}^{\pi^\star}\right\|_{\text{span}}$ in the first inequality, then that $\mathbb{V}_{\widehat{P}_{s\pi^\star(s)}}\left[\widehat{V}_{\text{pe}}^{\pi^\star}\right] \leq \left\|\widehat{V}_{\text{pe}}^{\pi^\star}\right\|_{\text{span}}^2$, and then bounded the $\max$ by the sum. (While Lemma B.14 is stated for $\beta(s, \pi^\star(s)) \leq 1$, if $\beta(s, \pi^\star(s)) > 1$ then $T_{\beta(s,\pi^\star(s))}(\widehat{P}_{s\pi^\star(s)}, \widehat{V}_{\text{pe}}^{\pi^\star})$ is a constant vector so the bound is still true.)

Now since $\tilde{b}'$ satisfies $\widehat{V}_{\text{pe}}^{\pi^\star} = r_{\pi^\star} - \gamma\tilde{b}' + \gamma\widehat{P}_{\pi^\star}\widehat{V}_{\text{pe}}^{\pi^\star}$, we can rearrange to obtain that $\widehat{V}_{\text{pe}}^{\pi^\star} = (I - \gamma\widehat{P}_{\pi^\star})^{-1}(r_{\pi^\star} - \gamma\tilde{b}')$. Likewise by the standard Bellman equation we have that $V^{\pi^\star} = r_{\pi^\star} + \gamma P_{\pi^\star}V^{\pi^\star}$ so $V^{\pi^\star} = (I - \gamma P_{\pi^\star})^{-1}r_{\pi^\star}$. Then we can calculate that

$$V^{\pi^\star} - \widehat{V}_{\text{pe}}^{\pi^\star} = (I - \gamma P_{\pi^\star})^{-1}r_{\pi^\star} - (I - \gamma\widehat{P}_{\pi^\star})^{-1}(r_{\pi^\star} - \gamma\tilde{b}')$$

$$= (I - \gamma\widehat{P}_{\pi^\star})^{-1}(I - \gamma\widehat{P}_{\pi^\star})(I - \gamma P_{\pi^\star})^{-1}r_{\pi^\star}$$

$$\quad - (I - \gamma\widehat{P}_{\pi^\star})^{-1}(I - \gamma P_{\pi^\star})(I - \gamma P_{\pi^\star})^{-1}(r_{\pi^\star} - \gamma\tilde{b}')$$

$$= \gamma(I - \gamma\widehat{P}_{\pi^\star})^{-1}(P_{\pi^\star} - \widehat{P}_{\pi^\star})(I - \gamma P_{\pi^\star})^{-1}r_{\pi^\star} + (I - \gamma\widehat{P}_{\pi^\star})^{-1}\gamma\tilde{b}'$$

$$= \gamma(I - \gamma\widehat{P}_{\pi^\star})^{-1}(P_{\pi^\star} - \widehat{P}_{\pi^\star})V^{\pi^\star} + (I - \gamma\widehat{P}_{\pi^\star})^{-1}\gamma\tilde{b}'. \tag{42}$$

Now we can bound

$$\left\|\widehat{V}_{\text{pe}}^{\pi^\star}\right\|_{\text{span}} = \max_{s,s'}(e_s - e_{s'})^\top\widehat{V}_{\text{pe}}^{\pi^\star}$$

$$= \max_{s,s'}(e_s - e_{s'})^\top\left(V^{\pi^\star} + \widehat{V}_{\text{pe}}^{\pi^\star} - V^{\pi^\star}\right)$$

$$\leq \max_{s,s'}(e_s - e_{s'})^\top\left(V^{\pi^\star}\right) + \max_{s,s'}(e_s - e_{s'})^\top\left(\widehat{V}_{\text{pe}}^{\pi^\star} - V^{\pi^\star}\right)$$

$$= \left\|V^{\pi^\star}\right\|_{\text{span}} + \max_{s,s'}(e_s - e_{s'})^\top\left(\widehat{V}_{\text{pe}}^{\pi^\star} - V^{\pi^\star}\right). \tag{43}$$

Fixing arbitrary $s, s' \in \mathcal{S}$ and letting $\xi = e_s - e_{s'}$, and using (42), we have that

$$
\begin{aligned}
\xi^\top \left( \widehat{V}_{\mathrm{pe}}^{\pi^\star} - V^{\pi^\star} \right) &= \xi^\top \left( \gamma (I - \gamma \widehat{P}_{\pi^\star})^{-1} (\widehat{P}_{\pi^\star} - P_{\pi^\star}) V^{\pi^\star} - (I - \gamma \widehat{P}_{\pi^\star})^{-1} \gamma \tilde{b}' \right) \\
&\leq \gamma \left\| \xi^\top (I - \gamma \widehat{P}_{\pi^\star})^{-1} \right\|_1 \left\| (\widehat{P}_{\pi^\star} - P_{\pi^\star}) V^{\pi^\star} \right\|_\infty + \gamma \left\| \xi^\top (I - \gamma \widehat{P}_{\pi^\star})^{-1} \right\|_1 \left\| \tilde{b}' \right\|_\infty.
\end{aligned}
$$
(44)

Next we bound all the terms in (44). First, $\left\| \xi^\top (I - \gamma \widehat{P}_{\pi^\star})^{-1} \right\|_1 \leq 4 T_{\mathrm{hit}}(\widehat{P}, \pi^\star)$ by Lemma B.11, and furthermore by Lemma B.17, since its conditions are satisfied under the conditions of the present lemma (since $\alpha \geq \log(\frac{2S}{\delta})$), we have with probability at least $1 - \delta$ that $T_{\mathrm{hit}}(\widehat{P}, \pi^\star) \leq 24 T_{\mathrm{hit}}(P, \pi^\star)$. Hence $\left\| \xi^\top (I - \gamma \widehat{P}_{\pi^\star})^{-1} \right\|_1 \leq 96 T_{\mathrm{hit}}(P, \pi^\star)$. Next, for any $s \in \mathcal{S}$, by Hoeffding's inequality, with probability at least $1 - \delta'$ we have

$$
\left| e_s^\top (\widehat{P}_{\pi^\star} - P_{\pi^\star}) V^{\pi^\star} \right| \leq \sqrt{\frac{\|V^{\pi^\star}\|_{\mathrm{span}}^2 \log\left(\frac{2}{\delta'}\right)}{2 n(s, \pi^\star(s))}}
$$

and so by a union bound over all $s \in \mathcal{S}$ and setting $\delta' = \frac{\delta}{S}$, we have that with additional failure probability at most $\delta$ that

$$
\left\| (\widehat{P}_{\pi^\star} - P_{\pi^\star}) V^{\pi^\star} \right\|_\infty \leq \|V^{\pi^\star}\|_{\mathrm{span}} \sqrt{\max_{s \in \mathcal{S}} \frac{\log(\frac{2S}{\delta})}{2 n(s, \pi^\star(s))}}.
$$

Finally, using the bound (41), we have

$$
\begin{aligned}
\left\| \tilde{b}' \right\|_\infty &\leq \max_{s \in \mathcal{S}} \sqrt{\beta(s, \pi^\star(s)) \left\| \widehat{V}_{\mathrm{pe}}^{\pi^\star} \right\|_{\mathrm{span}}^2} + 2 \beta(s, \pi^\star(s)) \left\| \widehat{V}_{\mathrm{pe}}^{\pi^\star} \right\|_{\mathrm{span}} + \frac{5}{n_{\mathrm{tot}}} \\
&\leq \max_{s \in \mathcal{S}} 3 \sqrt{\beta(s, \pi^\star(s))} \left\| \widehat{V}_{\mathrm{pe}}^{\pi^\star} \right\|_{\mathrm{span}} + \frac{5}{n_{\mathrm{tot}}}
\end{aligned}
$$

because our condition on $n(s, \pi^\star(s))$ guarantees that $\beta(s, \pi^\star(s)) \leq 1$ so $\beta(s, \pi^\star(s)) \leq \sqrt{\beta(s, \pi^\star(s))}$.

Combining these three bounds with (44), using that $\gamma \leq 1$, and taking the maximum over all $s, s'$, we have that

$$
\begin{aligned}
&\max_{s, s'} (e_s - e_{s'})^\top \left( \widehat{V}_{\mathrm{pe}}^{\pi^\star} - V^{\pi^\star} \right) \\
&\leq 96 T_{\mathrm{hit}}(P, \pi^\star) \left( \|V^{\pi^\star}\|_{\mathrm{span}} \sqrt{\max_{s \in \mathcal{S}} \frac{\log(\frac{2S}{\delta})}{2 n(s, \pi^\star(s))}} + 3 \sqrt{\max_{s \in \mathcal{S}} \beta(s, \pi^\star(s))} \left\| \widehat{V}_{\mathrm{pe}}^{\pi^\star} \right\|_{\mathrm{span}} + \frac{5}{n_{\mathrm{tot}}} \right)
\end{aligned}
$$

Combining this with (43) and rearranging, we have that

$$
\begin{aligned}
&\left\| \widehat{V}_{\mathrm{pe}}^{\pi^\star} \right\|_{\mathrm{span}} \left( 1 - 3 \cdot 96 T_{\mathrm{hit}}(P, \pi^\star) \sqrt{\max_{s \in \mathcal{S}} \beta(s, \pi^\star(s))} \right) \\
&\leq \|V^{\pi^\star}\|_{\mathrm{span}} \left( 1 + 96 T_{\mathrm{hit}}(P, \pi^\star) \sqrt{\max_{s \in \mathcal{S}} \frac{\log(\frac{2S}{\delta})}{2 n(s, \pi^\star(s))}} \right) + 96 T_{\mathrm{hit}}(P, \pi^\star) \frac{5}{n_{\mathrm{tot}}}.
\end{aligned}
$$
(45)

Noticing that $576 = 3 \cdot 2 \cdot 96$, our condition on $n(s, \pi^\star(s))$ in the lemma statement is chosen exactly so that

$$
\begin{aligned}
\left( 1 - 3 \cdot 96 T_{\mathrm{hit}}(P, \pi^\star) \sqrt{\max_{s \in \mathcal{S}} \beta(s, \pi^\star(s))} \right) &= \left( 1 - 3 \cdot 96 T_{\mathrm{hit}}(P, \pi^\star) \sqrt{\max_{s \in \mathcal{S}} \frac{\alpha}{n(s, \pi^\star(s)) - 1}} \right) \\
&\geq 1 - \frac{1}{2} = \frac{1}{2}.
\end{aligned}
$$

Also since for all $s \in \mathcal{S}$, $\beta(s, \pi^\star(s)) = \frac{\alpha}{\max\{n(s,\pi^\star(s))-1,1\}} = \frac{\alpha}{n(s,\pi^\star(s))-1} \geq \frac{\log(\frac{2S}{\delta})}{2n(s,\pi^\star(s))}$ (since $\alpha \geq 8\log(\frac{2S}{\delta})$ and $n(s,\pi^\star(s)) \geq 4$ so $\max\{n(s,\pi^\star(s))-1,1\} = n(s,\pi^\star(s))-1 \geq \frac{1}{2}n(s,\pi^\star(s))$), we can also simply bound

$$\left( 1 + 96T_{\mathrm{hit}}(P,\pi^\star)\sqrt{\max_{s\in\mathcal{S}} \frac{\log(\frac{2S}{\delta})}{2n(s,\pi^\star(s))}} \right) \leq 1 + \frac{1}{2}.$$

We can also bound $96T_{\mathrm{hit}}(P,\pi^\star)\frac{5}{n_{\mathrm{tot}}} \leq 1$ (by lower-bounding $n_{\mathrm{tot}}$ by $n(s_0,\pi^\star(s_0))$ for one arbitrary $s_0 \in \mathcal{S}$). Combining all these bounds with (45), we obtain

$$\frac{1}{2}\left\| \widehat{V}_{\mathrm{pe}}^{\pi^\star} \right\|_{\mathrm{span}} \leq \frac{3}{2}\left\| V^{\pi^\star} \right\|_{\mathrm{span}} + 1$$

which implies

$$\left\| \widehat{V}_{\mathrm{pe}}^{\pi^\star} \right\|_{\mathrm{span}} \leq 3\left\| V^{\pi^\star} \right\|_{\mathrm{span}} + 2$$

as desired. $\qquad\square$

## B.8 Average-reward-to-discounted reduction

Now we can combine our previous results and relate the discounted MDP quantities to $\rho^{\pi^\star}$ and $h^{\pi^\star}$.

**Lemma B.19.** *There exist some absolute constants $C_1, C_2$ such that the following holds: Fix a deterministic unichain policy $\pi^\star$. Suppose that $n(s,\pi^\star(s)) \geq m\mu^{\pi^\star}(s) + 4 + \alpha\left(576T_{\mathrm{hit}}(P,\pi^\star)\right)^2$ for all $s \in \mathcal{S}$, where $\alpha = 8\log\left(\frac{6S^2 A n_{\mathrm{tot}}}{(1-\gamma)\delta}\right)$, and that $\frac{1}{1-\gamma} \geq m$ and $\frac{1}{1-\gamma} \geq 2$. Then with probability at least $1 - 5\delta$, we have that*

$$\rho^{\widehat{\pi}} \geq \rho^{\pi^\star} - \sqrt{\frac{C_1 S\left(\|h^{\pi^\star}\|_{\mathrm{span}}+1\right)\alpha}{m}}\mathbf{1} - \frac{C_2 S\left(\|h^{\pi^\star}\|_{\mathrm{span}}+1\right)\alpha}{m}\mathbf{1}.$$

*Proof.* By Lemma B.16 (the conditions of which are met here as $\alpha(s,\pi^\star(s))\left(576T_{\mathrm{hit}}(P,\pi^\star)\right)^2 \geq 4T_{\mathrm{hit}}(P,\pi^\star)$), we have under the event of Lemma B.8, which holds with probability at least $1-2\delta$, that

$$\max_{s_0\in\mathcal{S}}\left( V^{\pi^\star}(s_0) - \widehat{V}_{\mathrm{pe}}^{\pi^\star}(s_0) \right)$$

$$\leq \frac{1}{1-\gamma}\sqrt{\frac{2048S\left\|\widehat{V}_{\mathrm{pe}}^{\pi^\star}\right\|_{\mathrm{span}}\log\left(\frac{6S^2 A n_{\mathrm{tot}}}{(1-\gamma)\delta}\right)}{m}} + \frac{640S\left\|\widehat{V}_{\mathrm{pe}}^{\pi^\star}\right\|_{\mathrm{span}}\log\left(\frac{6S^2 A n_{\mathrm{tot}}}{(1-\gamma)\delta}\right)}{(1-\gamma)m} + \frac{12}{(1-\gamma)n_{\mathrm{tot}}}.$$

Combining this with the conclusion of Lemma B.18 which implies $\left\|\widehat{V}_{\mathrm{pe}}^{\pi^\star}\right\|_{\mathrm{span}} \leq 3\left(\|V^{\pi^\star}\|_{\mathrm{span}}+1\right)$ and adds additional failure probability at most $2\delta$ by the union bound, we have that

$$\max_{s_0\in\mathcal{S}}\left( V^{\pi^\star}(s_0) - \widehat{V}_{\mathrm{pe}}^{\pi^\star}(s_0) \right) \leq \frac{1}{1-\gamma}\sqrt{\frac{6144S\left(\|V^{\pi^\star}\|_{\mathrm{span}}+1\right)\log\left(\frac{6S^2 A n_{\mathrm{tot}}}{(1-\gamma)\delta}\right)}{m}}$$

$$+ \frac{1920S\left(\|V^{\pi^\star}\|_{\mathrm{span}}+1\right)\log\left(\frac{6S^2 A n_{\mathrm{tot}}}{(1-\gamma)\delta}\right)}{(1-\gamma)m} + \frac{12}{(1-\gamma)n_{\mathrm{tot}}}.$$
$$(46)$$

For convenience abbreviate the right-hand-side of (46) as $\varepsilon$. Then since $Q^{\widehat{\pi}} \geq \widehat{Q}$ by Lemma B.9 (which holds under the event of Lemma B.7, adding additional failure probability at most $\delta$) and $\widehat{Q} \geq \widehat{Q}_{\mathrm{pe}}^\star - \frac{1}{2n_{\mathrm{tot}}}\mathbf{1}$ by Lemma B.4, we have that

$$V^{\widehat{\pi}} = M^{\widehat{\pi}}Q^{\widehat{\pi}} \geq M^{\widehat{\pi}}\widehat{Q} = M\widehat{Q} \geq M\left(\widehat{Q}_{\mathrm{pe}}^\star - \frac{1}{2n_{\mathrm{tot}}}\mathbf{1}\right) = M\widehat{Q}_{\mathrm{pe}}^\star - \frac{1}{2n_{\mathrm{tot}}}\mathbf{1} = \widehat{V}_{\mathrm{pe}}^\star - \frac{1}{2n_{\mathrm{tot}}}\mathbf{1}.$$
$$(47)$$

Furthermore we have

$$\widehat{V}^{\star}_{\mathrm{pe}} \overset{(i)}{\geq} \widehat{V}^{\pi^{\star}}_{\mathrm{pe}} \overset{(ii)}{\geq} V^{\pi^{\star}} - \varepsilon\mathbf{1} \overset{(iii)}{\geq} \frac{1}{1-\gamma}\rho^{\pi^{\star}} - \left\|V^{\pi^{\star}}\right\|_{\mathrm{span}}\mathbf{1} - \varepsilon\mathbf{1} \tag{48}$$

where $(i)$ is due to Lemma B.1 which gives $\widehat{Q}^{\star}_{\mathrm{pe}} \geq \widehat{Q}^{\pi^{\star}}_{\mathrm{pe}}$, which implies $\widehat{V}^{\star}_{\mathrm{pe}} = M\widehat{Q}^{\star}_{\mathrm{pe}} \geq M^{\pi^{\star}}\widehat{Q}^{\star}_{\mathrm{pe}} \geq M^{\pi^{\star}}\widehat{Q}^{\pi^{\star}}_{\mathrm{pe}}$ using monotonicity of $M^{\pi^{\star}}$. $(ii)$ is due to (46), and $(iii)$ uses $\left\|V^{\pi^{\star}} - \frac{1}{1-\gamma}\rho^{\pi^{\star}}\right\|_{\infty} \leq \left\|V^{\pi^{\star}}\right\|_{\mathrm{span}}$ due to Zurek and Chen [2025a, Lemma 6]. Also by Zurek and Chen [2025a, Lemma 6], we have the elementwise inequality $\rho^{\widehat{\pi}} \geq (1-\gamma)\left(\min_{s\in\mathcal{S}} V^{\widehat{\pi}}(s)\right)\mathbf{1}$. Thus

$$\rho^{\widehat{\pi}} \geq (1-\gamma)\min_{s\in\mathcal{S}} V^{\widehat{\pi}}(s)\mathbf{1}$$

$$\overset{(i)}{\geq} (1-\gamma)\min_{s\in\mathcal{S}} \widehat{V}^{\star}_{\mathrm{pe}}(s)\mathbf{1} - \frac{1-\gamma}{2n_{\mathrm{tot}}}\mathbf{1}$$

$$\overset{(ii)}{\geq} \min_{s\in\mathcal{S}} \rho^{\pi^{\star}}(s)\mathbf{1} - (1-\gamma)\left\|V^{\pi^{\star}}\right\|_{\mathrm{span}}\mathbf{1} - (1-\gamma)\varepsilon\mathbf{1} - \frac{1-\gamma}{2n_{\mathrm{tot}}}\mathbf{1}$$

$$\overset{(iii)}{\geq} \rho^{\pi^{\star}} - (1-\gamma)\left\|V^{\pi^{\star}}\right\|_{\mathrm{span}}\mathbf{1} - \frac{1-\gamma}{2n_{\mathrm{tot}}}\mathbf{1} - \sqrt{\frac{6144S\left(\left\|V^{\pi^{\star}}\right\|_{\mathrm{span}} + 1\right)\log\left(\frac{6S^2An_{\mathrm{tot}}}{(1-\gamma)\delta}\right)}{m}}\mathbf{1}$$

$$\quad - \frac{1920S\left(\left\|V^{\pi^{\star}}\right\|_{\mathrm{span}} + 1\right)\log\left(\frac{6S^2An_{\mathrm{tot}}}{(1-\gamma)\delta}\right)}{m}\mathbf{1} - \frac{12}{n_{\mathrm{tot}}}\mathbf{1}$$

$$\overset{(iv)}{\geq} \rho^{\pi^{\star}} - \sqrt{\frac{6144S\left(\left\|V^{\pi^{\star}}\right\|_{\mathrm{span}} + 1\right)\log\left(\frac{6S^2An_{\mathrm{tot}}}{(1-\gamma)\delta}\right)}{m}}\mathbf{1} - \frac{1933S\left(\left\|V^{\pi^{\star}}\right\|_{\mathrm{span}} + 1\right)\log\left(\frac{6S^2An_{\mathrm{tot}}}{(1-\gamma)\delta}\right)}{m}\mathbf{1}$$

where $(i)$ uses (47), $(ii)$ uses (48), $(iii)$ uses the fact that $\rho^{\pi^{\star}}$ is assumed to be state-independent and the definition of $\varepsilon$ (and canceling/simplifying), and $(iv)$ uses that $\frac{1}{1-\gamma} \geq m$ (so $(1-\gamma) \leq \frac{1}{m}$), that $1-\gamma \leq 1$, and $n_{\mathrm{tot}} \geq m$.

Furthermore, using Zurek and Chen [2025a, Lemma 26] we have (since $\rho^{\pi^{\star}}$ is constant) that $\left\|V^{\pi^{\star}}\right\|_{\mathrm{span}} \leq 2\left\|h^{\pi^{\star}}\right\|_{\mathrm{span}}$. Combining this with the above bound and letting $C_1 = 2 \cdot 6144/8$, $C_2 = 2 \cdot 1933/8$, we obtain the desired bound. $\qquad\square$

## B.9 Completing the proof

Here we complete the proof of the main Theorem 3.2 by checking conditions and simplifying previous results. The following result is actually more general than Theorem 3.2 because it allows an arbitrary unichain deterministic comparator policy $\pi^{\star}$, rather than requiring $\pi^{\star}$ to be gain-optimal. Theorem 3.2 follows immediately from the below theorem by adding this additional requirement that $\rho^{\pi^{\star}} = \rho^{\star}$.

**Theorem B.20.** *There exist absolute constants $C_1', C_2'$ such that the following holds: Fix $\delta > 0$. Let $\gamma = 1 - \frac{1}{n_{\mathrm{tot}}}$ and $\alpha = 8\log\left(\frac{6S^2An_{\mathrm{tot}}}{(1-\gamma)\delta}\right)$. Let $\pi^{\star}$ be a deterministic policy which is unichain with stationary distribution $\mu^{\pi^{\star}}$. Suppose there exists some $m \in \mathbb{N}$ such that*

$$n(s, \pi^{\star}(s)) \geq m\mu^{\pi^{\star}}(s) + \alpha\left(C_2'T_{\mathrm{hit}}(P, \pi^{\star})\right)^2 + 4.$$

*Then letting $\widehat{\pi}$ be the policy returned by Algorithm 1 with inputs $\mathcal{D}$, $r$, $\gamma = 1 - \frac{1}{n_{\mathrm{tot}}}$, and $\delta$, we have with probability at least $1 - 5\delta$ that*

$$\rho^{\widehat{\pi}} \geq \rho^{\pi^{\star}} - \sqrt{\frac{C_1'S\left(\left\|h^{\pi^{\star}}\right\|_{\mathrm{span}} + 1\right)\alpha}{m}}.$$

*Proof.* Note that the condition on $n$ implies that $n_{\mathrm{tot}} \geq 4$, so setting $\frac{1}{1-\gamma} = n_{\mathrm{tot}}$ has $\frac{1}{1-\gamma} \geq 2$. Also we have

$$n_{\mathrm{tot}} \geq \sum_{s\in\mathcal{S}} n(s, \pi^{\star}(s)) \geq \sum_{s\in\mathcal{S}} m\mu^{\pi^{\star}}(s) = m$$

using the assumption on $n(s, \pi^\star(s))$ for all $s$, so setting $\frac{1}{1-\gamma} = n_{\text{tot}}$ also ensures $\frac{1}{1-\gamma} \geq m$. Therefore we can apply Lemma B.19 to obtain that if $n(s, \pi^\star(s)) \geq m\mu^{\pi^\star}(s) + 4 + \alpha \left(576 T_{\text{hit}}(P, \pi^\star)\right)^2$ for all $s \in \mathcal{S}$, then with probability at least $1 - 5\delta$, we have

$$\rho^{\widehat{\pi}} \geq \rho^{\pi^\star} - \sqrt{\frac{C_1 S \left(\left\|h^{\pi^\star}\right\|_{\text{span}} + 1\right)\alpha}{m}} \mathbf{1} - \frac{C_2 S \left(\left\|h^{\pi^\star}\right\|_{\text{span}} + 1\right)\alpha}{m} \mathbf{1}$$

where $\alpha = 8\log\left(\frac{6S^2 A n_{\text{tot}}}{(1-\gamma)\delta}\right) = 8\log\left(\frac{6S^2 A n_{\text{tot}}^2}{\delta}\right)$. Thus we can set $C_2' = 576$. To choose $C_1'$, note that since trivially $\rho^{\pi^\star} \leq \mathbf{1}$ and $\rho^{\widehat{\pi}} \geq \mathbf{0}$, if the term $\frac{C_2 S \left(\left\|h^{\pi^\star}\right\|_{\text{span}} + 1\right)\alpha}{m} \geq 1$ then the bound

$$\rho^{\widehat{\pi}} \geq \rho^{\pi^\star} - \sqrt{\frac{C_2 S \left(\left\|h^{\pi^\star}\right\|_{\text{span}} + 1\right)\alpha}{m}} \mathbf{1}$$

holds vacuously, and otherwise if it is $\leq 1$ then we have

$$\rho^{\widehat{\pi}} \geq \rho^{\pi^\star} - \sqrt{\frac{C_1 S \left(\left\|h^{\pi^\star}\right\|_{\text{span}} + 1\right)\alpha}{m}} \mathbf{1} - \sqrt{\frac{C_2 S \left(\left\|h^{\pi^\star}\right\|_{\text{span}} + 1\right)\alpha}{m}} \mathbf{1}$$

since $\sqrt{x} \geq x$ for $x \in [0,1]$. Since $\sqrt{a} + \sqrt{b} \leq \sqrt{2(a+b)}$, we can take $C_1' = 2(C_1 + C_2)$. $\qquad\square$

## C Proof of Theorem 3.3

Let $T \geq 4$ and $m \in \mathbb{N}$ be arbitrary.

**Step 1: MDP construction** Define $p = \frac{1}{3(m+T)}$, $A = \left\lceil \frac{16}{pT} \right\rceil$, and $q = \frac{1}{AT}$. The set of states is $\mathcal{S} = \{0, 1\}$, and the set of actions is $\mathcal{A} = \{0, 1, \ldots, A-1\}$. The reward function $r : \mathcal{S} \times \mathcal{A} \to [0, 1]$ is defined by $r(0, a) = 1$ and $r(1, a) = 0$ for all $a \in \mathcal{A}$. We define an index set $\Theta = \left\{(i, b) \,\middle|\, i \in \{0, 1\}, b \in \{0, 1, \ldots, A-1\}\right\}$. For each $\theta = (i, b) \in \Theta$, we define the transition matrix $P_\theta$ as follows:

| $s$ | $a$ | $P_\theta(s'\|s, a)$ |
|---|---|---|
| 0 | $i$ | $\mathbb{I}(s' = 0)$ |
| 0 | $1 - i$ | $(1-p)\,\mathbb{I}(s' = 0) + p\mathbb{I}(s' = 1)$ |
| 0 | $\geq 2$ | $\mathbb{I}(s' = 1)$ |
| 1 | $b$ | $\frac{1}{T}\mathbb{I}(s' = 0) + \left(1 - \frac{1}{T}\right)\mathbb{I}(s' = 1)$ |
| 1 | $\neq b$ | $q\mathbb{I}(s' = 0) + (1 - q)\,\mathbb{I}(s' = 1)$ |

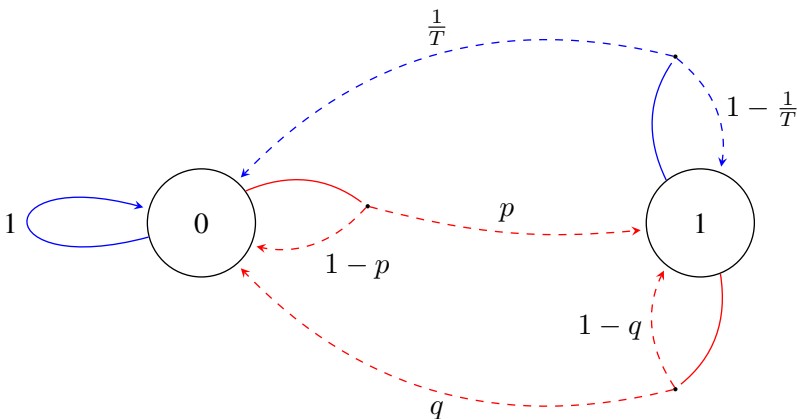

Figure 2: Diagram of the MDP $(P_{(0,0)}, r)$. Arrows splitting into multiple dashed arrows indicate stochastic transitions, and each dashed arrow is annotated with the associated probability. Blue arrows represent action 0 and red arrows represent action 1. In state 1, the red arrow also represents actions $2, \ldots, A-1$ (which are all identical). The reward function does not depend on the action, and is $+1$ in state 0 and $+0$ in state 1. In general, the MDP $(P_{(i,b)}, r)$ is similar, except that the blue arrow in state 0 represents action $i$ and the blue arrow in state 1 represents action $b$.

See Figure 2 for a diagram of the MDP $(P_\theta, r)$ for $\theta = (0, 0)$. We now state some easily verifiable facts about the MDP $(P_\theta, r)$:

- The unique deterministic gain-optimal stationary policy $\pi_\theta^\star$ is the one that takes action $i$ in state 0 and action $b$ in state 1.

- The optimal gain is $\rho_\theta^* = 1$.

- $\mu_\theta^{\pi_\theta^\star}(0) = 1$ and $\mu_\theta^{\pi_\theta^\star}(1) = 0$.

- The policy hitting radius $T_{\text{hit}}(P_\theta, \pi_\theta^\star)$, the optimal bias span $\left\| h_{P_\theta}^{\pi_\theta^\star} \right\|_{\text{span}}$, and the diameter are all at most $T$.

- Suppose a stationary policy $\pi$ usually makes the wrong decisions – specifically $\pi(i|0) < \frac{1}{2}$ and $\pi(b|1) < \frac{4}{A}$. Then $\rho_\theta^\pi < \frac{\frac{4}{A} \cdot \frac{1}{T} + (1-\frac{4}{A})q}{\frac{4}{A} \cdot \frac{1}{T} + (1-\frac{4}{A})q + \frac{p}{2}} \leq \frac{5q}{5q + \frac{p}{2}} \leq \frac{\frac{5p}{16}}{\frac{5p}{16} + \frac{p}{2}} < \frac{1}{2}$. In words, our choice of $A$ is one that is sufficiently large so that randomly guessing the optimal action $b$ in state 1 will not yield a good policy.

Note that action 2 in state 0 is added to keep the diameter bounded by $T$, and actions $3, \ldots, A-1$ in state 0 simply keep the action space independent of the state, consistent with our upper bounds. Since actions $2, \ldots, A-1$ in state 0 are always suboptimal, whenever we consider some policy $\pi$, we will assume that $\pi(a|0) = 0$ for $a \geq 2$.

**Step 2: dataset construction**  For any $\delta \in \left(0, \frac{1}{e^9}\right]$, denote $t_\delta = \left\lceil \frac{T}{6} \log \left(\frac{1}{\delta}\right) \right\rceil$. We define $n : \mathcal{S} \times \mathcal{A} \to \mathbb{N}$ by $n(0,0) = n(0,1) = m + t_\delta$ and $n(1,a) = t_\delta$ for all $a \in \mathcal{A}$. Observe that this choice of $n$ satisfies the desired requirements. Indeed, since $\mu_\theta^{\pi_\theta^\star}(0) = 1$ and $\mu_\theta^{\pi_\theta^\star}(1) = 0$, we have

$$n(0, \pi_\theta^\star(0)) = n(0, i) \geq m + \frac{T}{6} \log \left(\frac{1}{\delta}\right) = m \mu_\theta^{\pi_\theta^\star}(0) + \frac{T}{6} \log \left(\frac{1}{\delta}\right)$$

and

$$n(1, \pi_\theta^\star(1)) = n(1, b) \geq \frac{T}{6} \log \left(\frac{1}{\delta}\right) = m \mu_\theta^{\pi_\theta^\star}(1) + \frac{T}{6} \log \left(\frac{1}{\delta}\right).$$

**Step 3: impossible to do well in all MDPs**  Suppose towards a contradiction that there exists an algorithm $\mathscr{A}$ that maps the dataset $\mathcal{D}$ to a stationary policy $\hat{\pi} = \mathscr{A}(\mathcal{D})$ such that for all $\theta \in \Theta$, $\mathbb{P}_{\theta,n}\left(\rho_\theta^{\hat{\pi}} > \frac{1}{2}\right)$.

Before proceeding, we define some events. Let $\mathcal{B}$ be the bad event that $\mathcal{D}$ contains no transitions from state 0 to state 1 and no transitions from state 1 to state 0. Let $\mathcal{E}_0$ be the event that $\hat{\pi}(0|0) \geq \frac{1}{2}$ ($\hat{\pi}$ prefers action 0 in state 0). Similarly, let $\mathcal{E}_1$ be the event that $\hat{\pi}(1|0) \geq \frac{1}{2}$ ($\hat{\pi}$ prefers action 1 in state 0). For each $a \in \mathcal{A}$, let $\mathcal{F}_a$ be the event that $\hat{\pi}(a|1) \geq \frac{4}{A}$ ($\hat{\pi}$ gives significant weight to action $a$ in state 1).

A key idea is that under event $\mathcal{B}$, the dataset is the same no matter the underlying MDP. That is, under event $\mathcal{B}$, we always have

$$\mathcal{D} = (\ \underbrace{0, \ldots, 0}_{2n(0,0) \text{ times}}\ ,\ \underbrace{1, \ldots, 1}_{An(1,0) \text{ times}}\ ).$$

It follows that for all $\theta, \theta' \in \Theta$,

$$\mathbb{P}_{\theta,n}(\mathcal{E}_i \,|\, \mathcal{B}) = \mathbb{P}_{\theta',n}(\mathcal{E}_i \,|\, \mathcal{B}) \qquad \forall i \in \{0, 1\}$$

and

$$\mathbb{P}_{\theta,n}(\mathcal{F}_a \,|\, \mathcal{B}) = \mathbb{P}_{\theta',n}(\mathcal{F}_a \,|\, \mathcal{B}) \qquad \forall a \in \mathcal{A}.$$

For ease of notation, going forward we will drop the subscript $\theta, n$ when it does not matter what the underlying MDP is.

Since $\mathbb{P}(\mathcal{E}_0 \cup \mathcal{E}_1 \,|\, \mathcal{B}) = 1$, we must have $\mathbb{P}(\mathcal{E}_{i'} \,|\, \mathcal{B}) \geq \frac{1}{2}$ for some $i' \in \{0, 1\}$. Furthermore, for some $a' \in \mathcal{A}$ we have $\mathbb{P}(\mathcal{F}_{a'} \,|\, \mathcal{B}) \leq \frac{1}{4}$, or equivalently, $\mathbb{P}(\mathcal{F}_{a'}^c \,|\, \mathcal{B}) > \frac{3}{4}$. Indeed, if this were not the case, we would have

$$\mathbb{E}\left[\sum_{a \in \mathcal{A}} \hat{\pi}(a|1) \,\middle|\, \mathcal{B}\right] = \sum_{a \in \mathcal{A}} \mathbb{E}\left[\hat{\pi}(a|1) \,|\, \mathcal{B}\right] \geq \sum_{a \in \mathcal{A}} \mathbb{E}\left[\hat{\pi}(a|1) \,|\, \mathcal{F}_a \cap \mathcal{B}\right] \mathbb{P}(\mathcal{F}_a \,|\, \mathcal{B}) > \sum_{a \in \mathcal{A}} \frac{4}{A} \cdot \frac{1}{4} = 1,$$

which is a contradiction because we always have $\sum_{a \in \mathcal{A}} \hat{\pi}(a|1) = 1$.

We have shown that when the dataset does not contain any useful transitions, there must be at least one MDP where the algorithm is likely to make a poor guess. Our last step will be to combine this fact with Lemma C.1 which tells us that the dataset will be useless with large enough probability. We noted above that when the underlying MDP is $(P_{(i',a')}, r)$ and a policy $\pi$ satisfies $\pi(i'|0) < \frac{1}{2}$ and $\pi(a'|1) < \frac{4}{A}$ we have $\rho_{(i',a')}^\pi < \frac{1}{2}$. In particular, under the the event $\mathcal{E}_{i'}^c \cap \mathcal{F}_{a'}^c$ we have $\rho_{(i',a')}^{\hat{\pi}} < \frac{1}{2}$. Subsequently, for $\theta' = (i', a')$, we have

$$\mathbb{P}_{\theta',n}\left(\rho_{\theta'}^{\hat{\pi}} < \frac{1}{2}\right) \geq \mathbb{P}_{\theta',n}(\mathcal{E}_{i'}^c \cap \mathcal{F}_{a'}^c) \geq \mathbb{P}_{\theta',n}(\mathcal{E}_{i'}^c \cap \mathcal{F}_{a'}^c \cap \mathcal{B}) = \mathbb{P}(\mathcal{E}_{i'}^c \cap \mathcal{F}_{a'}^c | \mathcal{B})\mathbb{P}_{\theta'}(\mathcal{B}) \geq \frac{1}{4} \cdot 4\delta = \delta,$$

where the final inequality follows from Lemma C.1.

In summary, we have shown that

$$\max_{\theta \in \Theta} \mathbb{P}_{\theta,n}\left(\rho_\theta^* - \rho_\theta^{\mathscr{A}(\mathcal{D})} \geq \frac{1}{2}\right) \geq \delta,$$

as desired. $\qquad\qquad\square$

## C.1 Auxiliary lemmas

**Lemma C.1.** *For all $\theta \in \Theta$, we have $\mathbb{P}_{\theta,n}(\mathcal{B}) \geq 4\delta$.*

*Proof.* By symmetry $\mathbb{P}_\theta(\mathcal{B})$ are equal for all $\theta$, so for ease of notation we drop the subscript $\theta$. Let $\mathcal{B}_0$ be the event that $\mathcal{D}$ contains no transitions from state 0 to state 1, and let $\mathcal{B}_1$ be the event that $\mathcal{D}$ contains no transitions from state 1 to state 0. Then

$$\mathbb{P}(\mathcal{B}) = \mathbb{P}(\mathcal{B}_0 \cap \mathcal{B}_1) = \mathbb{P}(\mathcal{B}_0)\mathbb{P}(\mathcal{B}_1),$$

with the last equality following by independence. Now,

$$\mathbb{P}(\mathcal{B}_0) = (1 - p)^{m+t_\delta}.$$

Recall that $p = \frac{1}{3(m+T)}$. In the case that $m \geq t_\delta$, we have

$$(1 - p)^{m+t_\delta} \geq \left(1 - \frac{1}{6m}\right)^{2m} \geq \frac{1}{e}, \tag{49}$$

with the last inequality following from Lemma C.2 with $x = 2m$ and $c = 3$. Otherwise, when $m < t_\delta$, we have

$$(1-p)^{m+t_\delta} \geq \left(1 - \frac{1}{6T}\right)^{2t_\delta} \geq 4\delta^{1/3}, \tag{50}$$

with the last inequality following from claim 3 of Lemma C.3 with $x = 2T$. Combining Equations (49) and (50) and the fact that $4\delta^{1/3} \leq \frac{1}{e}$, we have

$$\mathbb{P}(\mathcal{B}_0) \geq 4\delta^{1/3}.$$

Next,

$$\mathbb{P}(\mathcal{B}_1) = \left(1 - \frac{1}{T}\right)^{t_\delta} (1-q)^{(A-1)t_\delta}.$$

Claim 2 of Lemma C.3 with $x = T$ gives us that $\left(1 - \frac{1}{T}\right)^{t_\delta} \geq \delta^{1/3}$. Moreover, recalling that $q = \frac{1}{AT}$, we have

$$(1-q)^{(A-1)t_\delta} \geq (1-q)^{At_\delta} = \left(1 - \frac{1}{AT}\right)^{At_\delta} \geq \delta^{1/3},$$

with the last inequality following from claim 2 of Lemma C.3 with $x = AT$. Hence, $\mathbb{P}(\mathcal{B}_1) \geq \delta^{2/3}$, and consequently, $\mathbb{P}(\mathcal{B}) \geq 4\delta$.

$\square$

**Lemma C.2.** *For all $x \geq 2$ and $c \geq 2$, we have*

$$\left(1 - \frac{1}{cx}\right)^x \geq \frac{1}{e}.$$

*Proof.* We have

$$\begin{aligned}
\log\left(\left(1 - \frac{1}{cx}\right)^x\right) &= x\log\left(1 - \frac{1}{cx}\right) \\
&\geq x\left(-\frac{1}{cx} - \frac{1}{c^2x^2}\right) \\
&= -\frac{1}{c}\left(1 + \frac{1}{cx}\right) \\
&\geq -\frac{2}{c} \\
&\geq -1 \\
&= \log\left(\frac{1}{e}\right),
\end{aligned}$$

where the first inequality follows from $\log(1-y) \geq -y - y^2$ for $y \in [0, 0.68]$. Since $\log x$ is monotonically increasing, we are done. $\square$

**Lemma C.3.** *For any $x \geq 4$, the following holds:*

1. *For any $\delta \in \left(0, \frac{1}{e}\right]$, we have $\left(1 - \frac{1}{x}\right)^{\left\lceil \frac{x}{2}\log\left(\frac{1}{\delta}\right)\right\rceil} \geq \delta$.*

2. *For any $\delta \in \left(0, \frac{1}{e^3}\right]$, we have $\left(1 - \frac{1}{x}\right)^{\left\lceil \frac{x}{6}\log\left(\frac{1}{\delta}\right)\right\rceil} \geq \delta^{1/3}$.*

3. *For any $\delta \in \left(0, \frac{1}{e^9}\right]$, we have $\left(1 - \frac{1}{3x}\right)^{\left\lceil \frac{x}{6}\log\left(\frac{1}{\delta}\right)\right\rceil} \geq 4\delta^{1/3}$.*

*Proof.* We will prove claim 1 by showing that $\left(1 - \frac{1}{x}\right)^{\frac{x}{2}\log\left(\frac{1}{\delta}\right)+1} \geq \delta$. For any $x \geq 4$ and $\delta \in \left(0, \frac{1}{e}\right]$, we have

$$
\begin{aligned}
\log\left(\left(1 - \frac{1}{x}\right)^{\frac{x}{2}\log\left(\frac{1}{\delta}\right)+1}\right) &= \left(\frac{x}{2}\log\left(\frac{1}{\delta}\right)+1\right)\log\left(1 - \frac{1}{x}\right) \\
&\geq \left(\frac{x}{2}\log\left(\frac{1}{\delta}\right)+1\right)\left(-\frac{1}{x} - \frac{1}{x^2}\right) \\
&= \left(\frac{1}{2} + \frac{1}{2x}\right)\log\delta - \frac{1}{x} - \frac{1}{x^2} \\
&\geq \frac{5}{8}\log\delta - \frac{5}{16} \\
&= \frac{5}{8}\log\delta + \frac{5}{16}\log\left(\frac{1}{e}\right) \\
&\geq \left(\frac{5}{8} + \frac{5}{16}\right)\log\delta \\
&\geq \log\delta,
\end{aligned}
$$

where the first inequality follows from $\log(1 - y) \geq -y - y^2$ for $y \in [0, 0.68]$. Since $\log x$ is monotonically increasing, claim 1 follows.

For claim 2, take $x \geq 4$ and $\delta \in \left(0, \frac{1}{e^3}\right]$. Then $\delta' = \delta^{1/3} \in \left(0, \frac{1}{e}\right]$, so by claim 1 we have

$$
\left(1 - \frac{1}{x}\right)^{\left\lceil\frac{x}{6}\log\left(\frac{1}{\delta}\right)\right\rceil} = \left(1 - \frac{1}{x}\right)^{\left\lceil\frac{x}{2}\log\left(\frac{1}{\delta'}\right)\right\rceil} \geq \delta' = \delta^{1/3}.
$$

Finally, for claim 3, take $x \geq 4$ and $\delta \in \left(0, \frac{1}{e^9}\right]$, and let $y = 3x$. Since $\delta' = \delta^{1/3} \in \left(0, \frac{1}{e^3}\right]$, claim 2 gives us that

$$
\left(1 - \frac{1}{3x}\right)^{\left\lceil\frac{x}{6}\log\left(\frac{1}{\delta}\right)\right\rceil} = \left(1 - \frac{1}{y}\right)^{\left\lceil\frac{y}{6}\log\left(\frac{1}{\delta'}\right)\right\rceil} \geq (\delta')^{1/3} \geq 4\delta^{1/3},
$$

where the last inequality holds because $\delta^{1/3} < \frac{1}{8}$. $\qquad\square$

## D   Proof of Theorem 3.4

We define the absolute constants $c_1 = 4$ and $c_2 = 33$. Let $T \geq c_1$, $S \geq c_2$, $k \geq 0$, and $m \geq \max\{TS, kS\}$ be arbitrary.

**Step 1: MDP construction**   Define $S' = S - 1$, $D = T - 2$, $\varepsilon = \frac{1}{256}\sqrt{\frac{TS}{m}}$. Note that $\varepsilon \leq \frac{1}{256}$. Let $p = \frac{1-\varepsilon}{D}$ and $q = \frac{1}{D}$. The set of states is $\mathcal{S} = \{0, 1, \ldots, S'\}$ and the set of actions is $\mathcal{A} = \{0, 1, \ldots, S'\}$. The reward function $r : \mathcal{S} \times \mathcal{A} \to [0, 1]$ is defined to be 1 when $s \neq 0$ and $a \leq 1$, and 0 otherwise. We define an index set $\Theta = \{0, 1\}^{S'}$. For each $\theta \in \Theta$, we define the transition matrix $P_\theta$ as follows:

| $s$ | $a$ | $P_\theta(s'\|s,a)$ |
|---|---|---|
| $0$ | $0$ | $(1-q)\mathbb{I}(s'=s) + \frac{q}{S'}\sum_{s''\geq 1}\mathbb{I}(s'=s'')$ |
| $0$ | $a \geq 1$ | $(1-\frac{q}{2})\mathbb{I}(s'=s) + \frac{q}{2S'}\sum_{s''\geq 1}\mathbb{I}(s'=s'')$ |
| $s \geq 1$ | $\theta_s$ | $(1-p)\mathbb{I}(s'=s) + p\mathbb{I}(s'=0)$ |
| $s \geq 1$ | $1-\theta_s$ | $(1-q)\mathbb{I}(s'=s) + q\mathbb{I}(s'=0)$ |
| $s \geq 2$ | $s$ | $\frac{1}{2}\mathbb{I}(s'=1) + \frac{1}{2S'}\sum_{s''\geq 1}\mathbb{I}(s'=s'')$ |
| $s \geq 1$ | $a \neq s, a \geq 2$ | $\frac{1}{2}\mathbb{I}(s'=a) + \frac{1}{2S'}\sum_{s''\geq 1}\mathbb{I}(s'=s'')$ |

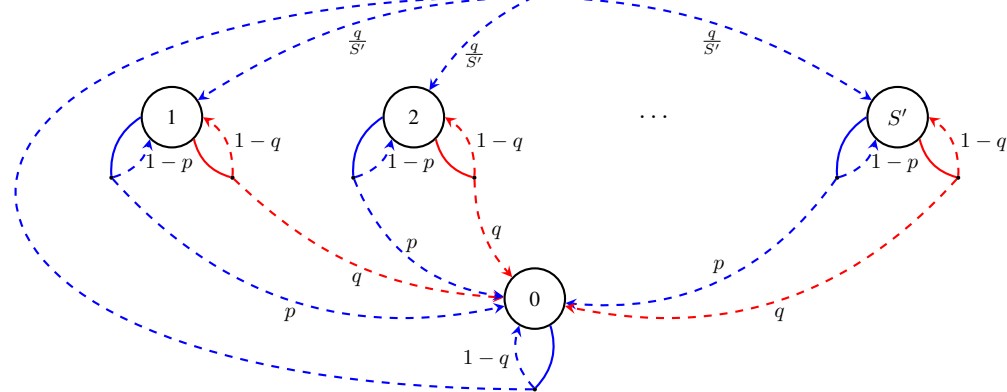

Figure 3: Diagram of the MDP $(P_{(0,\ldots,0)}, r)$ only including actions 0 and 1. Arrows splitting into multiple dashed arrows indicate stochastic transitions, and each dashed arrow is annotated with the associated probability. Blue arrows represent action 0 and red arrows represent action 1. The reward is 0 at state 0 and the reward is 1 at all other states. In general, the MDP $(P_\theta, r)$ is similar, except in each state $s \geq 1$, the blue arrow represents the optimal action $\theta_s$.

Observe that the decision-maker needs to decide between two actions in states $1, \ldots, S'$. Both actions give an immediate reward of 1, but one action has a slightly higher probability of transiting to the bad state 0. At state 0, which has a reward of 0, the agent will likely be trapped for a long time before returning to one of states $1, \ldots, S'$. See Figure 3 for a diagram of the MDP $(P_\theta, r)$ for $\theta = (0, \ldots, 0)$. We now state some easily verifiable facts about the MDP $(P_\theta, r)$:

- The MDP has $S$ states, is unichain, and has diameter $\frac{1}{q} + \frac{1}{1/2} = D + 2 = T$.

- There is a unique gain-optimal policy $\pi_\theta^\star$. It takes action 0 in state 0 and action $\theta_s$ in state $s$ for $s \geq 1$.

- $\mu_\theta^{\pi_\theta^\star}(0) = \frac{p}{p+q} = \frac{1-\varepsilon}{2-\varepsilon}$. By symmetry, it follows that $\mu_\theta^{\pi_\theta^\star}(s) = \frac{1}{S'}\left(1 - \mu_\theta^{\pi_\theta^\star}\right) = \frac{1/S'}{2-\varepsilon}$ for $s \geq 1$.

- The optimal gain is $\rho_\theta^* = 1 - \mu_\theta^{\pi_\theta^\star}(0) = \frac{1}{2-\varepsilon}$.

Note that actions $2, \ldots, S'$ for states $s \geq 1$ are always suboptimal, and only exist to keep the diameter bounded by $T$. Furthermore, actions $1, \ldots, S'$ in state 0 simply keep the action space independent of the state, consistent with our upper bounds. As such, whenever we consider some policy $\pi$, we will assume that it may only take actions 0 and 1 in states $s \geq 1$ and action 0 in state 0.

**Step 2: dataset construction**  We define $n : \mathcal{S} \times \mathcal{A} \to \mathbb{N}$ by $n(0,0) = m$ and

$$n(s, a) = \frac{2m}{S'}$$

for all $s \geq 1$ and $a \in \{0, 1\}$. For all other $(s, a)$ we set $n(s, a) = 0$. Observe that this choice of $n$ satisfies $n(s, \pi_\theta^\star(s)) = \frac{m}{S'} + \frac{m}{S'} \geq m\mu_\theta^{\pi_\theta^\star}(s) + k$ for all $s \in \mathcal{S}$.

**Step 3: reduction to estimation**  Given a stationary policy $\pi$ and some $\theta \in \Theta$, let $L_\theta^\pi(s)$ be the proportion of incorrect actions $\pi$ takes in state $s$. To be precise, we define $L_\theta^\pi(s) = \pi(1 - \theta_s|s)$. We also set $L_\theta^\pi = \sum_{s=1}^{S'} L_\theta^\pi(s)$. By Lemma D.1, we can upper bound the gain of a policy $\pi$ in terms of $L_\theta^\pi$:

$$\rho_\theta^\pi \leq \frac{1 + \varepsilon^2}{2 - \varepsilon(1 - L_\theta^\pi/S')}.$$

Subsequently, for any stationary policy $\pi$,

$$\rho_\theta^* - \rho_\theta^\pi \geq \frac{1}{2 - \varepsilon} - \frac{1 + \varepsilon^2}{2 - \varepsilon(1 - L_\theta^\pi/S')} \geq \frac{\varepsilon L_\theta^\pi/S' - 2\varepsilon^2}{4}. \tag{51}$$

Now, suppose the underlying MDP is $(P_\theta, r)$. Let $\mathscr{A}$ be an algorithm that maps the dataset to a stationary policy $\hat{\pi} = \mathscr{A}(\mathcal{D})$, and consider the estimator $\hat{\theta}^{\mathscr{A}}$ whose $s$th coordinate is $\hat{\pi}(1|s)$. By the definition of $L_\theta^{\hat{\pi}}$, we have $L_\theta^{\hat{\pi}} = \left\|\hat{\theta}^{\mathscr{A}} - \theta\right\|_1$. Our next step is to show that no estimator can achieve low $\ell_1$ error uniformly over $\Theta$ with high probability, a result which will lower bound $L_\theta^{\hat{\pi}}$ and consequently also the sub-optimality of $\hat{\pi}$ for some $\theta$.

**Step 4: Fano's method** We will achieve such a lower bound with Fano's method. First, by the Gilbert-Varshamov Lemma (Lemma D.2), there exists some subset $\Theta' \subset \Theta$ such that $|\Theta'| \geq 2^{S'/8}$ and $\|\theta - \theta'\|_1 \geq S'/8$ for any $\theta \neq \theta' \in \Theta'$. Since $\max_{\theta, \theta' \in \Theta'} \mathrm{KL}(\mathbb{P}_{\theta,n} \| \mathbb{P}_{\theta',n}) \leq (S'/16 - 1) \log 2$ by Lemma D.3, Local Fano's (Lemma D.4) gives us that for any estimator $\hat{\theta}$,

$$\max_\theta \mathbb{E}_{\theta,n}\left[\left\|\hat{\theta} - \theta\right\|_1\right] \geq \frac{S'}{16}\left(1 - \frac{(S'/16 - 1)\log 2 + \log 2}{\log\left(2^{S'/8}\right)}\right) \geq \frac{S'}{32},$$

which implies that

$$\max_{\theta \in \Theta} \mathbb{P}_{\theta,n}\left(\left\|\hat{\theta} - \theta\right\|_1 > \frac{S'}{64}\right) \geq \frac{1}{64}.$$

Since the above holds for estimator of the dataset, it of course holds for $\hat{\theta}^{\mathscr{A}}$, where $\mathscr{A}$ is any algorithm that maps the dataset to a stationary policy. Therefore,

$$\max_\theta \mathbb{P}_{\theta,n}\left(L_\theta^{\mathscr{A}(\mathcal{D})} > \frac{S'}{64}\right) \geq \frac{1}{64}. \tag{52}$$

Now, by Equation 51, in the event that $L_\theta^{\mathscr{A}(\mathcal{D})} > \frac{S'}{64}$,

$$\rho_\theta^* - \rho_\theta^{\mathscr{A}(\mathcal{D})} > \frac{\varepsilon/64 - 2\varepsilon^2}{4} \geq \frac{\varepsilon}{512} = 2^{-17}\sqrt{\frac{TS}{m}},$$

with the second inequality holding by $\varepsilon \leq \frac{1}{256}$. Thus, plugging back into Equation 52 yields

$$\max_\theta \mathbb{P}_{\theta,n}\left(\rho_\theta^* - \rho_\theta^{\mathscr{A}(\mathcal{D})} > c_3\sqrt{\frac{TS}{m}}\right) \geq \frac{1}{64},$$

with $c_3 = 2^{-17}$. $\qquad\square$

## D.1 Auxiliary lemmas

**Lemma D.1.** *Let $\pi$ be a stationary policy on MDP $M_\theta$. Then*

$$\rho_\theta^\pi \leq \frac{1 + \varepsilon^2}{2 - \varepsilon(1 - L_\theta^\pi/S')}.$$

*Proof.* A routine computation (see Lemma D.7) yields

$$\rho_\theta^\pi = \frac{\frac{q}{S'}\sum_{s=1}^{S'}\frac{1}{\kappa_s}}{1 + \frac{q}{S'}\sum_{s=1}^{S'}\frac{1}{\kappa_s}},$$

where $\kappa_s = L_\theta^\pi(s)q + (1 - L_\theta^\pi(s))p = \frac{1 - \varepsilon(1 - L_\theta^\pi(s))}{D}$ is the probability of transiting from state $s$ to state 0 under $\pi$. Since $\frac{x}{1+x}$ is monotonically increasing for $x > -1$, to achieve the desired upper bound for $\rho_\theta^\pi$ it suffices to find an acceptable upper bound for $\lambda := \frac{q}{S'}\sum_{s=1}^{S'}\frac{1}{\kappa_s} = \frac{1}{S'}\sum_{s=1}^{S'}\frac{1}{1 - \varepsilon(1 - L_\theta^\pi(s))}$.

Defining $f(x) = \frac{1}{1-x}$ and $\lambda_s = \varepsilon(1 - L_\theta^\pi(s))$, we have that

$$\lambda = \sum_{s=1}^{S'}\frac{1}{S'}f(\lambda_s).$$

We would like to get a bound that looks like $\lambda \le f\left(\frac{1}{S'}\sum_{s=1}^{S'}\lambda_s\right)$. This goal suggests applying Jensen's inequality, but since $f$ is convex for $x < 1$ it gives us an inequality in the wrong direction. It turns out, however, that because $f$ is nearly linear in the sufficiently small interval of interest, we can obtain an inequality in the right direction with some error term of lower order.

Since $\lambda_s \in [0, \varepsilon]$ for all $s \in \{1, \dots, S'\}$, Lemma D.6 give us

$$\lambda \le f\left(\sum_{s=1}^{S'}\frac{\lambda_s}{S'}\right) + f(0) + f(\varepsilon) - 2f\left(\frac{\varepsilon}{2}\right)$$

$$= \frac{1}{1 - \varepsilon(1 - L_\theta^\pi/S')} + 1 + \frac{1}{1 - \varepsilon} - \frac{2}{1 - \varepsilon/2}$$

$$\le \frac{1}{1 - \varepsilon(1 - L_\theta^\pi/S')} + \varepsilon^2,$$

where the last inequality holds for $\varepsilon < \frac{1}{3}$. Consequently,

$$\rho_\theta^\pi \le \frac{\lambda}{1 + \lambda} \le \frac{\frac{1}{1-\varepsilon(1-L_\theta^\pi/S')} + \varepsilon^2}{1 + \frac{1}{1-\varepsilon(1-L_\theta^\pi/S')} + \varepsilon^2} \le \frac{1 + \varepsilon^2}{2 - \varepsilon(1 - L_\theta^\pi/S')}.$$

$\square$

**Lemma D.2** (Gilbert-Varshamov Lemma [Massart, 2007, Lemma 4.7]). *Let $d \ge 8$. There exists $\Omega_d \subset \{0,1\}^d$ such that $|\Omega_d| \ge 2^{d/8}$ and $\|\omega - \omega'\|_1 \ge d/8$ for all $\omega \ne \omega' \in \Omega_d$.*

**Lemma D.3.** *For any $\theta, \theta' \in \Theta$, we have*

$$\mathrm{KL}(\mathbb{P}_{\theta,n} \,\|\, \mathbb{P}_{\theta',n}) \le \left(\frac{S'}{16} - 1\right)\log 2.$$

*Proof.* Let $\theta, \theta' \in \Theta$. By the construction of $\mathbb{P}_{\theta,n}$ and $\mathbb{P}_{\theta',n}$, we can decompose

$$\mathrm{KL}(\mathbb{P}_{\theta,n} \,\|\, \mathbb{P}_{\theta',n}) = \sum_{s=0}^{S'}\sum_{a \in \{0,1\}} n(s,a)\mathrm{KL}(P_\theta(\cdot\,|\,s,a) \,\|\, P_{\theta'}(\cdot\,|\,s,a)).$$

Recalling our choice of $n$, we can further simplify

$$\mathrm{KL}(\mathbb{P}_{\theta,n} \,\|\, \mathbb{P}_{\theta',n}) = \sum_{s=1}^{S'}\frac{2m}{S'}\left(\mathrm{KL}(P_\theta(\cdot\,|\,s,0) \,\|\, P_{\theta'}(\cdot\,|\,s,0)) + \mathrm{KL}(1_\theta(\cdot\,|\,s,1) \,\|\, P_{\theta'}(\cdot\,|\,s,1))\right),$$

where we remove the $s = 0$ term from the sum because the data coming from state 0 has the same distribution for all possible MDPs. Observing that

$$\frac{2(p-q)^2}{p(1-p)} = \frac{2(\varepsilon/D)^2}{\left(\frac{1-\varepsilon}{D}\right)\left(\frac{D-1+\varepsilon}{D}\right)} \le \frac{2\varepsilon^2}{\left(\frac{1}{2}\right)\left(\frac{D}{2}\right)} = \frac{8\varepsilon^2}{D},$$

we can apply Lemma D.5 to further simplify

$$\mathrm{KL}(\mathbb{P}_{\theta,n} \,\|\, \mathbb{P}_{\theta',n}) = \sum_{s=1}^{S'}\frac{2m}{S'}\left(\mathrm{KL}(P_\theta(\cdot\,|\,s,0) \,\|\, P_{\theta'}(\cdot\,|\,s,0)) + \mathrm{KL}(1_\theta(\cdot\,|\,s,1) \,\|\, P_{\theta'}(\cdot\,|\,s,1))\right)$$

$$\le 2m\left(\mathrm{KL}\left(\mathrm{Ber}(p) \,\|\, \mathrm{Ber}(q)\right) + \mathrm{KL}\left(\mathrm{Ber}(q) \,\|\, \mathrm{Ber}(p)\right)\right)$$

$$\le 2m\frac{8\varepsilon^2}{D}$$

$$= 2m\frac{8 \cdot 2^{-16}\frac{TS}{m}}{T-2}$$

$$\le 2^{-10}S'$$

$$\le \left(\frac{S'}{16} - 1\right)\log 2.$$

The final inequality holds due to the assumption that $S \ge 33 \implies S' \ge 32$. $\square$

**Lemma D.4** (Local Fano's inequality [Wainwright, 2019, Proposition 15.12, Equation 15.34]). *Let $\mathcal{P}$ be a class of distributions with parameter space $\Theta$, and let $\{\mathbb{P}_1, \ldots, \mathbb{P}_N\} \subset \mathcal{P}$. Letting $\theta(\mathbb{P}) \in \Theta$ denote the parameters of $\mathbb{P}$, define $\delta = \min_{j \neq k} \|\theta(\mathbb{P}_j) - \theta(\mathbb{P}_k)\|_1$. For any estimator $\hat{\theta}$, we have*

$$\sup_{\mathbb{P} \in \mathcal{P}} \mathop{\mathbb{E}}_{\mathcal{D} \sim \mathbb{P}} \left[ \left\| \hat{\theta}(\mathcal{D}) - \theta(\mathbb{P}) \right\|_1 \right] \geq \frac{\delta}{2} \left( 1 - \frac{\max_{j,k} \mathrm{KL}(\mathbb{P}_j \,\|\, \mathbb{P}_k) + \log 2}{\log N} \right).$$

**Lemma D.5.** *For any $p, q \in \left(0, \frac{1}{2}\right]$ satisfying $p < q$, we have*

$$\mathrm{KL}\left(\mathrm{Ber}(p) \,\|\, \mathrm{Ber}(q)\right) \leq \mathrm{KL}\left(\mathrm{Ber}(q) \,\|\, \mathrm{Ber}(p)\right) \leq \frac{(p-q)^2}{p(1-p)},$$

*which implies that*

$$\mathrm{KL}\left(\mathrm{Ber}(p) \,\|\, \mathrm{Ber}(q)\right) + \mathrm{KL}\left(\mathrm{Ber}(q) \,\|\, \mathrm{Ber}(p)\right) \leq \frac{2(p-q)^2}{p(1-p)}.$$

*Proof.* By Lemma 10 in Li et al. [2023], we have

$$\mathrm{KL}\left(\mathrm{Ber}(p') \,\|\, \mathrm{Ber}(q')\right) \leq \mathrm{KL}\left(\mathrm{Ber}(q') \,\|\, \mathrm{Ber}(p')\right) \leq \frac{(p'-q')^2}{p'(1-p')}$$

for any $p', q' \in \left[\frac{1}{2}, 1\right)$ satisfying $p' > q'$. The desired result follows immediately by taking $p' = 1 - p$ and $q' = 1 - q$, along with the observation that $\mathrm{KL}\left(\mathrm{Ber}(1-p) \,\|\, \mathrm{Ber}(1-q)\right) = \mathrm{KL}\left(\mathrm{Ber}(p) \,\|\, \mathrm{Ber}(q)\right)$. $\qquad\square$

**Lemma D.6** (Theorem 1 in Simic [2008]). *Let $I = [a, b]$ be a closed interval with $a, b \in \mathbb{R}$, $a < b$. For some $n \in \mathbb{Z}^+$, let $x_1, \ldots, x_n \in I$, and let $p_1, \ldots, p_n > 0$ satisfy $\sum_{i=1}^n p_i = 1$. If $f : [a, b] \to \mathbb{R}$ is convex, then*

$$\sum_{i=1}^n p_i f(x_i) \leq f\left(\sum_{i=1}^n p_i x_i\right) + f(a) + f(b) - 2f\left(\frac{a+b}{2}\right).$$

**Lemma D.7.** *Suppose the underlying MDP is $(P_\theta, r)$. Let $\pi$ be a stationary policy such that for each $s \neq 0$, if the current state is $s$ then the probability of transiting to state 0 after taking action according to $\pi$ is $\kappa_s$. Then*

$$\rho_\theta^\pi = \frac{\frac{q}{S'} \sum_{s=1}^{S'} \frac{1}{\kappa_s}}{1 + \frac{q}{S'} \sum_{s=1}^{S'} \frac{1}{\kappa_s}}.$$

*Proof.* We first solve for $\mu_\theta^\pi(0)$ by considering the balance equations for the MDP $(P_\theta, r)$. For each $s \neq 0$, we have

$$\mu_\theta^\pi(s) = \frac{q}{S'} \mu_\theta^\pi(0) + (1 - \kappa_s)\mu_\theta^\pi(s).$$

Rearranging gives us

$$\mu_\theta^\pi(s) = \frac{q}{S'} \mu_\theta^\pi(0) \frac{1}{\kappa_s}.$$

Since $\sum_{s=0}^{S'} \mu_\theta^\pi(s) = 1$, we have

$$\mu_\theta^\pi(0) = 1 - \sum_{s=1}^{S'} \mu_\theta^\pi(s) = 1 - \mu_\theta^\pi(0) \frac{q}{S'} \sum_{s=1}^{S'} \frac{1}{\kappa_s}.$$

We then solve for $\mu_\theta^\pi(0)$ to obtain

$$\mu_\theta^\pi(0) = \frac{1}{1 + \frac{q}{S'} \sum_{s=1}^{S'} \frac{1}{\kappa_s}}.$$

Since the reward is 0 in state 0 and 1 in all other states, we conclude that

$$\rho_\theta^\pi = 1 - \mu_\theta^\pi(0) = \frac{\frac{q}{S'} \sum_{s=1}^{S'} \frac{1}{\kappa_s}}{1 + \frac{q}{S'} \sum_{s=1}^{S'} \frac{1}{\kappa_s}}.$$

$\qquad\square$

# E  Deferred proofs and auxiliary lemmas

## E.1  Proof of Lemma B.2

*Proof of Lemma B.2.* Letting $V, V' \in \mathbb{R}^{\mathcal{S}}$ satisfy $V \geq V'$ elementwise, we seek to show that

$$\overline{\mathcal{T}}_{\text{pe}}(V) \geq \overline{\mathcal{T}}_{\text{pe}}(V').$$

Since this is an elementwise bound, we can fix arbitrary $s \in \mathcal{S}, a \in \mathcal{A}$ and show that $\overline{\mathcal{T}}_{\text{pe}}(V)(s, a) \geq \overline{\mathcal{T}}_{\text{pe}}(V')(s, a)$. From here on, since $s, a$ are fixed, we abbreviate $\beta(s, a) \in \mathbb{R}$ as $\beta$ for notational convenience.

Consider the simpler function $\widetilde{\mathcal{T}} : \mathbb{R}^{\mathcal{S}} \to \mathbb{R}$ (which depends on our fixed $s, a$) defined as

$$\widetilde{\mathcal{T}}(V'') := \widehat{P}_{sa} T_{\beta}(\widehat{P}_{sa}, V'') - \max\left\{ \sqrt{\beta \mathbb{V}_{\widehat{P}_{sa}}\left[T_{\beta}(\widehat{P}_{sa}, V'')\right]}, \beta \left\|T_{\beta}(\widehat{P}_{sa}, V'')\right\|_{\text{span}} \right\}$$

for any $V'' \in \mathbb{R}^{\mathcal{S}}$. Note that

$$\overline{\mathcal{T}}_{\text{pe}}(V'')(s, a) = r(s, a) + \gamma \max\left\{ \widehat{P}_{sa} T_{\beta}(\widehat{P}_{sa}, V'') - b(s, a, V''), \min_{s'}(V'')(s') \right\}$$

$$= r(s, a) + \gamma \max\left\{ \widetilde{\mathcal{T}}(V'') - \frac{5}{n_{\text{tot}}}, \min_{s'}(V'')(s') \right\}.$$

Therefore, if we could show that

$$\widetilde{\mathcal{T}}(V) \geq \widetilde{\mathcal{T}}(V'), \tag{53}$$

then since clearly $V \geq V'$ implies $\min_{s'}(V)(s') \geq \min_{s'}(V')(s')$, we could immediately conclude that

$$\overline{\mathcal{T}}_{\text{pe}}(V)(s, a) = r(s, a) + \gamma \max\left\{ \widetilde{\mathcal{T}}(V) - \frac{5}{n_{\text{tot}}}, \min_{s'}(V)(s') \right\}$$

$$\geq r(s, a) + \gamma \max\left\{ \widetilde{\mathcal{T}}(V') - \frac{5}{n_{\text{tot}}}, \min_{s'}(V')(s') \right\}$$

$$= \overline{\mathcal{T}}_{\text{pe}}(V')(s, a)$$

as desired.

Thus we now focus on showing (53). First we can quickly handle the case that $\beta > 1$, since in this case for any $V'' \in \mathbb{R}^{\mathcal{S}}$ we have $T_{\beta}(\widehat{P}_{sa}, V'') = (\min_{s'} V''(s')) \mathbf{1}$, and then

$$\widetilde{\mathcal{T}}(V) = \widehat{P}_{sa} T_{\beta}(\widehat{P}_{sa}, V) - \max\left\{ \sqrt{\beta \mathbb{V}_{\widehat{P}_{sa}}\left[T_{\beta}(\widehat{P}_{sa}, V)\right]}, \beta \left\|T_{\beta}(\widehat{P}_{sa}, V)\right\|_{\text{span}} \right\}$$

$$= \left(\min_{s'} V(s')\right) \widehat{P}_{sa} \mathbf{1} - 0 = \min_{s'} V(s')$$

$$\geq \min_{s'} V'(s') = \left(\min_{s'} V'(s')\right) \widehat{P}_{sa} \mathbf{1} - 0$$

$$= \widehat{P}_{sa} T_{\beta}(\widehat{P}_{sa}, V') - \max\left\{ \sqrt{\beta \mathbb{V}_{\widehat{P}_{sa}}\left[T_{\beta}(\widehat{P}_{sa}, V')\right]}, \beta \left\|T_{\beta}(\widehat{P}_{sa}, V')\right\|_{\text{span}} \right\}$$

$$= \widetilde{\mathcal{T}}(V'),$$

confirming (53). Now we can focus on the case that $\beta \leq 1$.

The fact that $\beta \leq 1$ means that the following expression for $T_{\beta}$ holds: for any $s' \in \mathcal{S}$ and $V'' \in \mathbb{R}^{\mathcal{S}}$, we have

$$T_{\beta}(\widehat{P}_{sa}, V'')(s') = \min\left\{ V''(s'), Q_{\beta}(\widehat{P}_{sa}, V'') \right\}$$

where $Q_{\beta}(\widehat{P}_{sa}, V'') = \sup\{V''(x) : x \in \mathcal{S}, \sum_{x' \in \mathcal{S} : V(x') \geq V(x)} \widehat{P}_{sa}(x') \geq \beta\}$ is the $1 - \beta$ quantile of $V''$ with respect to $\widehat{P}_{sa}$ (in words, we choose the largest $V''(x)$ such that $\widehat{P}_{sa}$ places probability at

least $\beta$ on states $x'$ with $V''(x') \geq V''(x)$). We will make use of the function $Q_\beta$ shortly. We also make the useful definitions

$$\widetilde{\mathcal{T}}_1(V) := \widehat{P}_{sa} T_\beta(\widehat{P}_{sa}, V) - \beta \left\| T_\beta(\widehat{P}_{sa}, V) \right\|_{\text{span}}$$

$$\widetilde{\mathcal{T}}_2(V) := \widehat{P}_{sa} T_\beta(\widehat{P}_{sa}, V) - \sqrt{\beta \mathbb{V}_{\widehat{P}_{sa}} \left[ T_\beta(\widehat{P}_{sa}, V) \right]}$$

so that we can decompose $\widetilde{\mathcal{T}}$ as $\widetilde{\mathcal{T}}(V) = \min \left\{ \widetilde{\mathcal{T}}_1(V), \widetilde{\mathcal{T}}_2(V) \right\}$. To show (53), it suffices to show that this holds when $V$ and $V'$ differ in only one coordinate, since then we could decompose $V = V' + \sum_{s' \in \mathcal{S}} e_{s'} e_{s'}^\top (V - V')$ and apply the inequalities $\widetilde{\mathcal{T}} \left( V' + \sum_{s'=1}^{k-1} e_{s'} e_{s'}^\top (V - V') \right) \leq \widetilde{\mathcal{T}} \left( V' + \sum_{s'=1}^{k} e_{s'} e_{s'}^\top (V - V') \right)$ for each $k = 1, \ldots, S$. Therefore we fix one state $x \in \mathcal{S}$ and try to show $\widetilde{\mathcal{T}}(V)$ is montonically non-decreasing as $V(x)$ increases (with the other entries of $V$ held constant). We will show this by using Lemma E.1, which says that if a univariate function is continuous and at all but a finite number of points has a non-negative right derivative, then it must be non-decreasing.

First we justify that $\widetilde{\mathcal{T}}$ is continuous. Since we have decomposed $\widetilde{\mathcal{T}}$ as the composition of many continuous functions, it suffices to check that $Q_\beta(\widehat{P}_{sa}, V)$ is a continuous function of $V(x)$. This follows immediately from Lemma E.3, which shows 1-Lipschitzness. (We remark that the $1 - \beta$ quantile is well-known to be discontinuous in $\beta$, a fact which is irrelevant here since $\beta$ is fixed and we instead vary $V(x)$.)

We will now compute the right derivative at all values of $V(x)$ such that $V(x)$ is not equal to $V(s')$ for some other $s' \in \mathcal{S}$ with $s' \neq x$ (which is a finite set). We define some new notation for this purpose. With respect to this fixed value of $V(x)$, let $\mathcal{S}_> = \{s' \in \mathcal{S} : V(s') > V(x)\}$ and $\mathcal{S}_< = \{s' \in \mathcal{S} : V(s') < V(x)\}$. Define a neighborhood of $V(x)$, the open interval $U := (\max_{s' \in \mathcal{S}_<} V(s'), \min_{s' \in \mathcal{S}_>} V(s'))$. Let $V' \in \mathbb{R}^\mathcal{S}$ have $V'(s') = V(s')$ for all $s' \neq x$, and we vary $V'(x)$ within the neighborhood $U$ of $V(x)$ in order to compute the (full/two-sided) derivatives $\frac{d\widetilde{\mathcal{T}}_1(V')}{dV'(x)}\Big|_{V'(x)=V(x)}$ and $\frac{d\widetilde{\mathcal{T}}_2(V')}{dV'(x)}\Big|_{V'(x)=V(x)}$. Once we have computed these two derivatives, we will be able to compute the right derivative of $\widetilde{\mathcal{T}}(V')$, since if both $\widetilde{\mathcal{T}}_1(V')$ and $\widetilde{\mathcal{T}}_2(V')$ are differentiable at a point $V(x)$, then by Lemma E.2 the right derivative of $\widetilde{\mathcal{T}}(V')$ satisfies

$$
\frac{d\widetilde{\mathcal{T}}(V')}{dV'(x)}\bigg|_{V'(x)=V(x)^+} = \frac{d}{dV'(x)}\bigg|_{V'(x)=V(x)^+} \left( \min \left\{ \widetilde{\mathcal{T}}_1(V'), \widetilde{\mathcal{T}}_2(V') \right\} \right)
$$

$$
= \begin{cases}
\frac{d\widetilde{\mathcal{T}}_1(V')}{dV'(x)}\Big|_{V'(x)=V(x)} & \widetilde{\mathcal{T}}_1(V) < \widetilde{\mathcal{T}}_2(V) \\
\frac{d\widetilde{\mathcal{T}}_2(V')}{dV'(x)}\Big|_{V'(x)=V(x)} & \widetilde{\mathcal{T}}_1(V) > \widetilde{\mathcal{T}}_2(V) \\
\min \left\{ \frac{d\widetilde{\mathcal{T}}_1(V')}{dV'(x)}\Big|_{V'(x)=V(x)}, \frac{d\widetilde{\mathcal{T}}_2(V')}{dV'(x)}\Big|_{V'(x)=V(x)} \right\} & \widetilde{\mathcal{T}}_1(V) = \widetilde{\mathcal{T}}_2(V)
\end{cases}.
$$

$$(54)$$

To compute the derivatives of $\widetilde{\mathcal{T}}_1(V')$ and $\widetilde{\mathcal{T}}_2(V')$, we also analyze the functions $Q_\beta(\widehat{P}_{sa}, V')$ and $T_\beta(\widehat{P}_{sa}, V')$ on the set $U$ (all considered as functions of $V'(x)$). For any set $\mathcal{S}' \subseteq \mathcal{S}$, let $\widehat{P}_{sa}(\mathcal{S}') = \sum_{s' \in \mathcal{S}'} \widehat{P}_{sa}(s')$. We define three possible cases depending on the (fixed) state $x$:

$$\beta \leq \widehat{P}_{sa}(\mathcal{S}_>) \tag{55}$$

$$\widehat{P}_{sa}(\mathcal{S}_>) < \beta \leq \widehat{P}_{sa}(\mathcal{S}_>) + \widehat{P}_{sa}(x) \tag{56}$$

$$\widehat{P}_{sa}(\mathcal{S}_>) + \widehat{P}_{sa}(x) < \beta. \tag{57}$$

1. In case (55), we have $Q_\beta(\widehat{P}_{sa}, V') = Q_\beta(\widehat{P}_{sa}, V)$ on the entire interval $U$ and also that for any $V'(x) \in U$, $Q_\beta(\widehat{P}_{sa}, V) > V'(x)$ (since the $(1 - \beta)$-percentile is achieved at some

state $s' \in \mathcal{S}_>$), so $T_\beta(\widehat{P}_{sa}, V')(x) = V'(x)$ and $T_\beta(\widehat{P}_{sa}, V')(s') = T_\beta(\widehat{P}_{sa}, V)(s')$ for all $s' \neq x$. Therefore

$$\left. \frac{dT_\beta(\widehat{P}_{sa}, V')(s')}{dV'(x)} \right|_{V'(x)=V(x)} = \begin{cases} 1 & s' = x \\ 0 & \text{otherwise} \end{cases}$$

and

$$\begin{aligned}
\left. \frac{d\widetilde{\mathcal{T}}_1(V')}{dV'(x)} \right|_{V'(x)=V(x)} &= \left. \frac{d}{dV'(x)} \right|_{V'(x)=V(x)} \left( \widehat{P}_{sa} T_\beta(\widehat{P}_{sa}, V') - \beta \left\| T_\beta(\widehat{P}_{sa}, V') \right\|_{\text{span}} \right) \\
&= \left. \frac{d}{dV'(x)} \right|_{V'(x)=V(x)} \left( \widehat{P}_{sa} T_\beta(\widehat{P}_{sa}, V') - \beta Q_\beta(\widehat{P}_{sa}, V') + \beta \min_{s'} V'(s') \right) \\
&= \left. \frac{d}{dV'(x)} \right|_{V'(x)=V(x)} \left( \widehat{P}_{sa} T_\beta(\widehat{P}_{sa}, V') - \beta Q_\beta(\widehat{P}_{sa}, V) + \beta \min_{s'} V'(s') \right) \\
&= \widehat{P}_{sa}(x) + \beta \begin{cases} 1 & \mathcal{S}_< = \emptyset \\ 0 & \text{otherwise} \end{cases} \\
&\geq \widehat{P}_{sa}(x) \geq 0.
\end{aligned}$$

2. In case (56), we have $Q_\beta(\widehat{P}_{sa}, V') = V'(x)$ on the entire interval $U$. Thus $T_\beta(\widehat{P}_{sa}, V')(s') = V'(x)$ if $s' \in \mathcal{S}_> \cup \{x\}$, and $T_\beta(\widehat{P}_{sa}, V')(s') = V'(s') = V(s')$ for $s' \in S_<$. Thus

$$\left. \frac{dT_\beta(\widehat{P}_{sa}, V')(s')}{dV'(x)} \right|_{V'(x)=V(x)} = \begin{cases} 1 & s' \in \mathcal{S}_> \cup \{x\} \\ 0 & \text{otherwise} \end{cases}$$

and

$$\begin{aligned}
\left. \frac{d\widetilde{\mathcal{T}}_1(V')}{dV'(x)} \right|_{V'(x)=V(x)} &= \left. \frac{d}{dV'(x)} \right|_{V'(x)=V(x)} \left( \widehat{P}_{sa} T_\beta(\widehat{P}_{sa}, V') - \beta \left\| T_\beta(\widehat{P}_{sa}, V') \right\|_{\text{span}} \right) \\
&= \left. \frac{d}{dV'(x)} \right|_{V'(x)=V(x)} \left( \widehat{P}_{sa} T_\beta(\widehat{P}_{sa}, V') - \beta Q_\beta(\widehat{P}_{sa}, V') + \beta \min_{s'} V'(s') \right) \\
&= \left. \frac{d}{dV'(x)} \right|_{V'(x)=V(x)} \left( \widehat{P}_{sa} T_\beta(\widehat{P}_{sa}, V') - \beta V'(x) + \beta \min_{s'} V'(s') \right) \\
&= \widehat{P}_{sa}(\mathcal{S}_> \cup \{x\}) - \beta + \beta \begin{cases} 1 & \mathcal{S}_< = \emptyset \\ 0 & \text{otherwise} \end{cases} \\
&\geq \widehat{P}_{sa}(\mathcal{S}_> \cup \{x\}) - \beta \geq 0.
\end{aligned}$$

3. In case (57), we have $Q_\beta(\widehat{P}_{sa}, V') = Q_\beta(\widehat{P}_{sa}, V)$ and also that $T_\beta(\widehat{P}_{sa}, V')(x) = Q_\beta(\widehat{P}_{sa}, V) < V'(x)$ (since $V'(x) < Q_\beta(\widehat{P}_{sa}, V)$ in this case), so $T_\beta(\widehat{P}_{sa}, V') = T_\beta(\widehat{P}_{sa}, V)$ on the interval $U$. Also $\min_{s'} V'(s') < V'(x)$ on $U$, so $\min_{s'} V'(s') = \min_{s'} V(s')$ on $U$. Thus

$$\left. \frac{dT_\beta(\widehat{P}_{sa}, V')(s')}{dV'(x)} \right|_{V'(x)=V(x)} = 0$$

for all $s' \in \mathcal{S}$, and

$$\begin{aligned}
\left. \frac{d\widetilde{\mathcal{T}}_1(V')}{dV'(x)} \right|_{V'(x)=V(x)} &= \left. \frac{d}{dV'(x)} \right|_{V'(x)=V(x)} \left( \widehat{P}_{sa} T_\beta(\widehat{P}_{sa}, V') - \beta Q_\beta(\widehat{P}_{sa}, V') + \beta \min_{s'} V'(s') \right) \\
&= \left. \frac{d}{dV'(x)} \right|_{V'(x)=V(x)} \left( \widehat{P}_{sa} T_\beta(\widehat{P}_{sa}, V) - \beta Q_\beta(\widehat{P}_{sa}, V) + \beta \min_{s'} V(s') \right) \\
&= 0.
\end{aligned}$$

Next we calculate $\frac{d\widetilde{\mathcal{T}}_2(V')}{dV'(x)}\Big|_{V'(x)=V(x)}$ . First, letting $T \in \mathbb{R}^{\mathcal{S}}$, if $\mathbb{V}_{\widehat{P}_{sa}}[T] \neq 0$ then (recalling $\widehat{P}_{sa}$ is a row vector so $\widehat{P}_{sa}^{\top}$ is a column vector)

$$
\begin{aligned}
\nabla_T \sqrt{\mathbb{V}_{\widehat{P}_{sa}}[T]} &= \frac{1}{2} \frac{1}{\sqrt{\mathbb{V}_{\widehat{P}_{sa}}[T]}} \nabla_T \left( \widehat{P} T^{\circ 2} - (\widehat{P}T)^{\circ 2} \right) \\
&= \frac{1}{\sqrt{\mathbb{V}_{\widehat{P}_{sa}}[T]}} \left( \widehat{P}_{sa}^{\top} \circ T - (\widehat{P}_{sa}T)\widehat{P}_{sa}^{\top} \right) \\
&= \frac{1}{\sqrt{\mathbb{V}_{\widehat{P}_{sa}}[T]}} \widehat{P}_{sa}^{\top} \circ \left( T - (\widehat{P}_{sa}T)\mathbf{1} \right) \\
&\leq \frac{\|T\|_{\text{span}}}{\sqrt{\mathbb{V}_{\widehat{P}_{sa}}[T]}} \widehat{P}_{sa}^{\top}
\end{aligned}
\tag{58}
$$

where the final inequality is elementwise and uses the fact that for any $s'$, $T(s') - \widehat{P}_{sa}T \leq \max_{s''} T(s'') - \min_{s''} T(s'') = \|T\|_{\text{span}}$. Now we will combine this calculation with the chain rule to lower bound $\frac{d\widetilde{\mathcal{T}}_2(V')}{dV'(x)}\Big|_{V'(x)=V(x)}$. Note that in light of (54), we only need to bound $\frac{d\widetilde{\mathcal{T}}_2(V')}{dV'(x)}\Big|_{V'(x)=V(x)}$ when $\widetilde{\mathcal{T}}_1(V) > \widetilde{\mathcal{T}}_2(V)$ or equivalently when our fixed value of $V(x)$ satisfies

$$
\sqrt{\mathbb{V}_{\widehat{P}_{sa}}\left[T_\beta(\widehat{P}_{sa}, V)\right]} > \sqrt{\beta} \left\| T_\beta(\widehat{P}_{sa}, V) \right\|_{\text{span}}.
\tag{59}
$$

Since we have already excluded the finite set of values of $V(x)$ where $V(x)$ is equal to $V(s')$ for some other state $s' \neq x$, the only way for $\mathbb{V}_{\widehat{P}_{sa}}\left[T_\beta(\widehat{P}_{sa}, V)\right] = 0$ is if $\widehat{P}_{sa}(x) = 1$, but in that case we have $\left\| T_\beta(\widehat{P}_{sa}, V) \right\|_{\text{span}} = 0$ which contradicts (59). Therefore we can calculate that if $V(x)$ satisfies (59), we have

$$
\begin{aligned}
\frac{d\widetilde{\mathcal{T}}_2(V')}{dV'(x)}\Big|_{V'(x)=V(x)} &= \frac{d}{dV'(x)}\Big|_{V'(x)=V(x)} \left( \widehat{P}_{sa} T_\beta(\widehat{P}_{sa}, V) - \sqrt{\beta \mathbb{V}_{\widehat{P}_{sa}}\left[T_\beta(\widehat{P}_{sa}, V)\right]} \right) \\
&= \sum_{s'\in\mathcal{S}} \left( \frac{\partial}{\partial T(s')}\Big|_{T(s')=T_\beta(\widehat{P}_{sa},V)(s')} \left( \widehat{P}_{sa}T - \sqrt{\beta \mathbb{V}_{\widehat{P}_{sa}}[T]} \right) \right) \cdot \frac{dT_\beta(\widehat{P}_{sa},V')(s')}{dV'(x)}\Big|_{V'(x)=V(x)} \\
&= \sum_{s'\in\mathcal{S}} \left( \widehat{P}_{sa}(s') - \sqrt{\beta} \frac{\partial\sqrt{\mathbb{V}_{\widehat{P}_{sa}}[T]}}{\partial T(s')}\Big|_{T(s')=T_\beta(\widehat{P}_{sa},V)(s')} \right) \cdot \frac{dT_\beta(\widehat{P}_{sa},V')(s')}{dV'(x)}\Big|_{V'(x)=V(x)} \\
&\geq \sum_{s'\in\mathcal{S}} \left( \widehat{P}_{sa}(s') - \sqrt{\beta} \frac{\left\| T_\beta(\widehat{P}_{sa},V) \right\|_{\text{span}}}{\sqrt{\mathbb{V}_{\widehat{P}_{sa}}\left[T_\beta(\widehat{P}_{sa},V)\right]}} \widehat{P}_{sa}(s') \right) \cdot \frac{dT_\beta(\widehat{P}_{sa},V')(s')}{dV'(x)}\Big|_{V'(x)=V(x)} \\
&> \sum_{s'\in\mathcal{S}} \left( \widehat{P}_{sa}(s') - \widehat{P}_{sa}(s') \right) \cdot \frac{dT_\beta(\widehat{P}_{sa},V')(s')}{dV'(x)}\Big|_{V'(x)=V(x)} \\
&= 0
\end{aligned}
$$

where the first inequality step is using the fact that $\frac{dT_\beta(\widehat{P}_{sa},V')(s')}{dV'(x)}\Big|_{V'(x)=V(x)} \geq 0$ for all $s'$ (verified above in all three cases) and inequality (58), and the second inequality step uses (59). $\qquad\square$

## E.2 Auxiliary lemmas

**Lemma E.1.** *If $f : \mathbb{R} \to \mathbb{R}$ is a continuous function that has a nonnegative right derivative for all but finitely many points, then $f$ is monotonically non-decreasing.*

*Proof.* We make the following claim: for $a, b \in \mathbb{R}$ with $a < b$, if $f : [a, b] \to \mathbb{R}$ is continuous on $[a, b]$ and has a nonnegative right derivative on $(a, b)$, then $f$ is monotonically non-decreasing on $[a, b]$.

We first prove the lemma assuming that the claim holds. Let $f : \mathbb{R} \to \mathbb{R}$ be a continuous function that has a nonnegative right derivative for all but finitely many points. Let $x, y \in \mathbb{R}$ satisfy $x < y$, and denote by $a_1, \ldots, a_{n-1}$ the points in $(x, y)$ where $f$ either is not right-differentiable or has negative right derivative. Also denote $a_0 = x$ and $a_n = y$. By the claim, $f$ is monotonically increasing on $[a_{i-1}, a_i]$ for each $i = 1, \ldots, n$. Hence $f(x) = f(a_0) \le f(a_1) \le \cdots \le f(a_n) = f(y)$. Since $x$ and $y$ were arbitrary, we conclude that $f$ is monotonically increasing.

It remains to prove the claim. Let $a, b \in \mathbb{R}$ with $a < b$, and let $f : [a, b] \to \mathbb{R}$ be continuous on $[a, b]$ with a nonnegative right derivative on $(a, b)$. Suppose towards a contradiction that there exist $x, y \in [a, b]$ such that $x < y$ and $f(x) > f(y)$. Since $f$ is continuous, we can assume that $x > a$ (if $x = a$ we have $x + \delta < y$ and $f(x + \delta) > f(y)$ for sufficiently small $\delta > 0$).

Now, set $r := \frac{f(y) - f(x)}{y - x} < 0$ and

$$z := \inf \left\{ t \in (x, y] \;\middle|\; \frac{f(t) - f(x)}{t - x} < \frac{r}{2} \right\}.$$

Consider the case where $z = x$. $f$ has a nonnegative right derivative at $x$, so there exists $w \in (x, y]$ such that $\frac{f(t) - f(x)}{t - x} > \frac{r}{2}$ for all $t \in (x, w]$. However, this implies a contradiction:

$$z = \inf \left\{ t \in (x, y] \;\middle|\; \frac{f(t) - f(x)}{t - x} < \frac{r}{2} \right\} \ge w > x = z.$$

We next consider the case where $z > x$. Note that by continuity of $f$, the function $g(t) := \frac{f(t) - f(x)}{t - x}$ is continuous on $(x, y]$. It follows that $g(z) = \frac{f(z) - f(x)}{z - x} = \frac{r}{2}$. Indeed, if we had $g(z) > \frac{r}{2}$, then by continuity of $g$ there would exist $\delta > 0$ such that $g(t) > \frac{r}{2}$ for $t \in [z, z + \delta]$, which would imply that $z \ge z + \delta$. And by a similar argument, $g(z) < \frac{r}{2}$ would imply $z \le z - \delta$.

At $z$ the right-derivative is nonnegative, so there exists $w \in (z, y]$ such that $\frac{f(t) - f(z)}{t - z} > \frac{r}{2}$ for all $t \in (z, w]$. Consequently, for all $t \in (z, w]$, we have

$$\frac{f(t) - f(x)}{t - x} = \frac{1}{t - x}(f(t) - f(z) + f(z) - f(x)) > \frac{1}{t - x}\left(\frac{r}{2}(t - z) + (z - x)\right) = \frac{r}{2},$$

which implies the following contradiction:

$$z = \inf \left\{ t \in (x, y] \;\middle|\; \frac{f(t) - f(x)}{t - x} < \frac{r}{2} \right\} \ge w > z.$$

$\square$

**Lemma E.2.** *Let $f, g : \mathbb{R} \to \mathbb{R}$ be differentiable at some $x \in \mathbb{R}$, and suppose $f(x) = g(x)$. Then $\phi : \mathbb{R} \to \mathbb{R}$ defined by $\phi(t) = \min\{f(t), g(t)\}$ is right-differentiable at $x$, and its right derivative satisfies $\phi'_+(x) = \min\{f'(x), g'(x)\}$.*

*Proof.* We first consider the case where $f'(x) < g'(x)$. Since $\lim_{h \to 0} \frac{f(x+h) - f(x)}{h} < \lim_{h \to 0} \frac{g(x+h) - g(x)}{h}$, there exists some $\delta > 0$ such that $\frac{f(x+h) - f(x)}{h} < \frac{g(x+h) - g(x)}{h}$ for all $h \in (0, \delta)$. Subsequently, since $f(x) = g(x)$, we have $f(x + h) < g(x + h)$ for all $h \in (0, \delta)$. It follows that $\phi(x + h) = f(x + h)$ for all $h \in (0, \delta)$, and thus

$$\lim_{h \to 0^+} \frac{\phi(x + h) - \phi(x)}{h} = \lim_{h \to 0^+} \frac{f(x + h) - f(x)}{h} = f'(x) = \min\{f'(x), g'(x)\}.$$

Next, the case where $f'(x) > g'(x)$ is identical to the previous case except we swap the roles of $f$ and $g$.

Finally, we consider the case where $f'(x) = g'(x)$. Here we can even show that $\phi$ is differentiable at $x$. Let $\{h_n\}_{n\in\mathbb{N}}$ be a sequence such that $h_n \to 0$. To show that $\frac{\phi(x+h_n)-\phi(x)}{h_n} \to f'(x)$, fix $\varepsilon > 0$. Since $\frac{f(x+h_n)-f(x)}{h_n} \to f'(x)$ and $\frac{g(x+h_n)-g(x)}{h_n} \to g'(x)$, there exist $N_1, N_2 \in \mathbb{N}$ such that

$$n \geq N_1 \implies \left| \frac{f(x+h_n) - f(x)}{h_n} - f'(x) \right| \leq \varepsilon$$

and

$$n \geq N_2 \implies \left| \frac{g(x+h_n) - g(x)}{h_n} - g'(x) \right| \leq \varepsilon.$$

Taking $N = \max\{N_1, N_2\}$, we have for all $n \geq \mathbb{N}$,

$$\left| \frac{\phi(x+h_n) - \phi(x)}{h_n} - f'(x) \right|$$
$$\leq \max\left\{ \left| \frac{f(x+h_n)-f(x)}{h_n} - f'(x) \right|, \left| \frac{g(x+h_n)-g(x)}{h_n} - g'(x) \right| \right\}$$
$$\leq \max\{\varepsilon, \varepsilon\} = \varepsilon,$$

where the first inequality holds due to $f(x) = g(x)$, $f'(x) = g'(x)$, and the fact that for each $n$, either $\phi(x+h_n) = f(x+h_n)$ or $\phi(x+h_n) = g(x+h_n)$. Thus, we have that $\frac{\phi(x+h_n)-\phi(x)}{h_n} \to f'(x)$. Since the sequence $\{h_n\}_{n\in\mathbb{N}}$ was arbitrary, we conclude that

$$\phi'(x) = \lim_{h\to 0} \frac{\phi(x+h) - \phi(x)}{h} = f'(x) = \min\{f'(x), g'(x)\}.$$

$\square$

**Lemma E.3.** *For any probability distribution $\mu \in \mathbb{R}^{\mathcal{S}}$ and any $\beta \in [0,1]$, the largest-$(1-\beta)$-quantile function*

$$Q_\beta(\mu, V'') = \sup\{V''(x) : x \in \mathcal{S}, \sum_{x'\in\mathcal{S}:V(x')\geq V(x)} \mu(x') \geq \beta\}$$

*satisfies*

$$|Q_\beta(\mu, V) - Q_\beta(\mu, V')| \leq \|V - V'\|_\infty$$

*for any $V, V' \in \mathbb{R}^{\mathcal{S}}$.*

*Proof.* First, we note that the definition of $Q_\beta$ can be written equivalently as

$$Q_\beta(\mu, V'') = \sup\left\{ \min_{s'\in\mathcal{S}'} V''(s') : \mathcal{S}' \subseteq \mathcal{S} \text{ and } \sum_{s'\in\mathcal{S}'} \mu(s') \geq \beta \right\}.$$

Without loss of generality we can assume that $Q_\beta(\mu, V) \geq Q_\beta(\mu, V')$, so it suffices to lower-bound $Q_\beta(\mu, V')$. By the definition of $Q_\beta(\mu, V)$ (and the fact that $\mathcal{S}$ is finite so the supremum within its definition is attained exactly), there exists some set $\mathcal{S}' \subseteq \mathcal{S}$ such that

$$Q_\beta(\mu, V) = \min_{s'\in\mathcal{S}'} V(s')$$

and $\sum_{s'\in\mathcal{S}'} \mu(s') \geq \beta$. Therefore since

$$V'(s') \geq V(s') - \|V - V'\|_\infty \geq Q_\beta(\mu, V) - \|V - V'\|_\infty$$

for all $s' \in \mathcal{S}'$, we have that

$$Q_\beta(\mu, V'') \geq Q_\beta(\mu, V) - \|V - V'\|_\infty$$

as desired. $\square$

