# OpenReview forum: "Optimal Single-Policy Sample Complexity and Transient Coverage for Average-Reward Offline RL"
_NeurIPS.cc/2025/Conference — NeurIPS 2025 poster_

### Official Review · Reviewer_Agk8 · 2025-06-30

**Clarity:** 3
**Significance:** 2
**Originality:** 3
**Rating:** 4
**Confidence:** 3

**Summary:**

This paper studies span-based results for offline reinforcement learning in average-reward MDPs. Unlike prior offline results that focus on unichain MDPs, the authors leverage the span-based approach (Matthew et al. (2024)) to derive results for the more general class of weakly communicating MDPs. They introduce a new quantity, $T_{\text{hit}}(P,\pi^\*)$, and show that pessimistic value iteration with a span-based bonus function succeeds if at least $O(T_{\text{hit}}(P,\pi^\*)^2)$ samples are collected for each state. Furthermore, they establish an $O(T_{\text{hit}}(P,\pi^\*))$ lower bound on the required number of samples per state. The theoretical results are supported by rigorous proofs.

**Questions:**

1. Is this quantile clipping technique useful in online RL, especially in span-based average-reward online RL?

2. The lower bound provided is only linear because it is based on the argument that certain transitions remain unobserved unless at least $O(T_{hit}(P,\pi^\*))$ samples are collected for each state. At a high level, I think transitions cannot be accurately identified with fewer than $O(T_{hit}(P,\pi^\*)^2)$ samples, which may be a potential avenue to further strengthen and improve the current lower bound result. Is it possible to improve your lower bound result in this way?

**Ethical Concerns:**

["NO or VERY MINOR ethics concerns only"]

**Final Justification:**

I appreciate the author's rebuttal. I will keep my score.

**Limitations:**

yes.

**Paper Formatting Concerns:**

no.

**Quality:**

3

**Strengths And Weaknesses:**

Strengths:

1. The paper is well-written and easy to follow. The intuition of the newly introduced term $T_{hit}(P,\pi^*)$ is clear, and the proof sketch can help the reader to quickly grasp the high-level idea of the proof.

2. The negative result for offline RL is both interesting and important. It highlights that offline RL requires sufficient samples for every state, even for states that are never visited by the optimal policy. This is a valuable insight, as many existing offline RL works rely only on the single-policy coverage assumption.

3. This paper is the first to consider the application of the pessimistic value iteration algorithm in average-reward MDPs. The proposed quantile clipping technique is both novel and interesting. The theoretical analysis is rigorous and may also be of independent interest.

Weaknesses:

1. The $\Omega(T_{hit}(P,\pi^\*))$ lower bound result does not match the $O(T_{hit}(P,\pi^\*)^2)$ sample complexity result.

2. There is no experiment to primarily examine whether $O(T_{hit}(P,\pi^\*)^2)$ samples are needed or $O(T_{hit}(P,\pi^\*))$ is enough. Some preliminary experiments that could provide theoretical insight would be great.

---

> ### Author Rebuttal · Authors · 2025-07-30
>
> Thank you for your positive comments. We would like to respond to some of your listed weaknesses and questions.
> 1. Weaknesses/question 2: Closing the gap between the upper and lower bounds (for the amount of data required for transient states) is the main quantitative/information-theoretical question left unresolved by our work and is an interesting question for future research. Our conjecture is that the lower bound is basically tight, and that potentially the same algorithm with a sharper analysis could be used to close the gap.
> 2. Question 1: We believe that there is a very high possibility that the quantile clipping technique could be applied in other settings to obtain sharper (span-based) penalty/bonus terms. The most direct application in online RL would likely be to the discounted setting (considered in e.g. arXiv:2010.00587), but as you suggest, it seems likely that improvements for discounted online RL could then be useful for the average-reward setting (possibly through a direct reduction between the settings).

---

> > ### Comment · Reviewer_Agk8 · 2025-08-05
> >
> > Thanks for the authors' rebuttal. I will keep my score.

---

### Official Review · Reviewer_6ZrZ · 2025-07-01

**Clarity:** 2
**Significance:** 2
**Originality:** 3
**Rating:** 5
**Confidence:** 2

**Summary:**

This paper addresses the sample complexity of average-reward offline RL in the weakly communicating MDPs. The established bound only depends on complexity measures corresponding to the optimal policy, instead of being uniform over all policies. The lower bounds results demonstrate the unimprovable dependence on two novel measures, i.e., the *transient state dataset coverage* and the *effective size*.

**Questions:**

1. Is the *transient state dataset coverage* assumption only necessary for a small suboptimality $\delta$ (as Theorem 3.3 requires $\delta \le e^{-9}$)?
2. How does Alg. 1 perform in the unichain MDPs? Can it learn a near-optimal policy without the *transient state dataset coverage* assumption?
3. In the hard instance (Section 4.2), the "difficulty" may also be explained by the *uniform mixing time*, i.e., it requires a sufficient amount of time to recover the transition from state 2 to state 1 for any arbitrary policy. It seems that both the uniform mixing time and the *transient state* are somewhat equivalent in this case.

**Ethical Concerns:**

["NO or VERY MINOR ethics concerns only"]

**Final Justification:**

The authors' response addressed most of my concerns. I maintain the acceptance of this paper.

**Limitations:**

yes

**Quality:**

3

**Strengths And Weaknesses:**

* **Strengths:**
    1. This paper presents a comprehensive analysis of the average-reward offline RL in the weakly communicating MDPs. The results improve over the previous work in two ways. First, the derived sample complexity bound only depends on the optimal policy. Second, two novel complexity measures are introduced, and the dependence is demonstrated through the lower bounds.
    2. An interesting result is that sufficient data is required even for transient state-action pairs. This could provide further insights to the community.

* **Weaknesses:**
    1. The derived bound requires data from all states, which could be limiting in practice.
    2. It is unclear whether the *transient state dataset coverage* assumption is fundamentally different from the *uniform mixing time* in the construction of the hard instance (see Question 3 below).

---

> ### Author Rebuttal · Authors · 2025-07-30
>
> Thank you for your positive comments. Below we respond to your questions.
> 1. Question 1: In Theorem 3.3 $\delta$ is actually the failure probability parameter. The suboptimality of the policy is $1/2$ (with probability at least $\delta$) in the theorem, so in short, the transient data coverage assumption is necessary at least for any suboptimality less than $1/2$.
> 2. The hard instances used in Theorem 3.3 are actually unichain, so the direct answer is no: even for unichain MDPs, transient data coverage is necessary in general to get a convergence rate scaling with the bias span of the optimal policy (as opposed to a uniform-style parameter like the uniform mixing time; see the paragraph beginning on line 263 for more discussion on which convergence rates are shown by Theorem 3.3 to be impossible without transient data coverage).
> 3. You are correct that the uniform mixing time is large for the family of hard instances used in Theorem 3.3, so in this sense, uniform-mixing-based bounds would also suggest that these instances are difficult. However, as mentioned in the paragraph after Theorem 3.3, the uniform mixing time and the policy hitting radius parameter (as used in the transient data coverage assumption) actually behave in fundamentally different ways for these instances: as the parameter $m$ is increased the policy hitting radius stays bounded (by the other parameter $T$), while the uniform mixing time grows proportionally to at least $m$. Notice that because the uniform mixing time is $\Omega(m)$, any offline RL theorem with a convergence rate depending on the uniform mixing time like $O\left(\sqrt{\frac{\tau_{unif}}{m}}\right)$ (in the best case/ignoring other factors) would have a vacuous $O(1)$ suboptimality bound for these instances, whether or not transient data is provided. On the other hand, our lower bound shows that non-vacuous suboptimality is impossible when there is insufficient transient data, but once there is enough transient data, then our upper bound can apply and yield a vanishing suboptimality (as $T$ is fixed and $m$ increases). In summary, no algorithm with a uniform-mixing-based convergence guarantee would yield nonvacuous performance bounds for these instances, while our algorithm and lower bound together reveal that vanishing error is possible but only with sufficient transient data.

---

> > ### Comment · Reviewer_6ZrZ · 2025-08-05
> >
> > Thank you for your response. I maintain my score.

---

### Official Review · Reviewer_88bt · 2025-07-02

**Clarity:** 3
**Significance:** 3
**Originality:** 3
**Rating:** 4
**Confidence:** 2

**Summary:**

This paper concerns the problem of offline RL in average-reward MDPs, where, due to the long horizon, there are unique challenges in handling distribution shift and non-uniform coverage. As I understand it, the main contribution is establishing a statistical upper bound utilizing only single-policy complexity terms, namely, the bias span and a novel policy hitting radius. The algorithm utilizes pessimistic value iteration with quantile clipping. Lower bounds establish that single-policy coverage alone is insufficient, and that the dependence on certain quantities (such as $m$) in the upper bound are tight.

**Questions:**

Could you highlight the elements of your analysis or algorithm that enable you to derive single-policy guarantees? (and perhaps what was lacking in previous work?)

**Ethical Concerns:**

["NO or VERY MINOR ethics concerns only"]

**Limitations:**

I think so

**Quality:**

3

**Strengths And Weaknesses:**

I am not familiar with average-reward RL, so I'm afraid my feedback is largely superficial, and I cannot evaluate this paper well in the context of existing work. As far as I can tell the paper seems carefully done and well-written, and I appreciate that the paper investigates lower bounds against their positive results.

Some parts of the bounds, such as $m$, were unfamiliar to me and so somewhat difficult to interpret from the exposition. Overall the paper is fairly dense, and I'm not sure how easy it is to extract or intuit which analytical or algorithmic interventions (policy hitting radius?) enabled the derivation of single-policy complexity results.

---

> ### Author Rebuttal · Authors · 2025-07-30
>
> Thank you for your positive comments. We now discuss a few points related to your question regarding key elements of the algorithm and analysis.
> 1. One key algorithmic feature is the use of quantile clipping and an empirical-span-based penalty function. This point is discussed in greater detail in the paragraph starting on line 195 as well as the proof sketch in section 4, but here we provide a brief summary. Using an empirical-span-based penalty term is essential for the average-reward setting, intuitively because it avoids a dependence on the effective horizon $\frac{1}{1-\gamma}$, which must be taken very large to obtain good average-reward performance. Naive implementations of empirical-span-based penalties fail (by breaking the monotonicity and contractivity properties of the associated pessimistic Bellman operator), but the new quantile clipping technique remedies these issues, getting the sharper penalty terms to work. As mentioned by reviewer Agk8/our response to reviewer Agk8, we feel like these sharper penalties may be of broader interest and utility.
> 2. It is possible that a key missing idea from previous work was the observation that transient data coverage is required. Since this is shown to be necessary by our Theorem 3.3, it suggests that prior attempts to obtain convergence rates involving single-policy complexity measures without this assumption were doomed to fail.
> 3. Many algorithms in the average-reward setting are plagued by the need for prior knowledge of complexity parameters (such as bias span or mixing time) in order to run the algorithm. Our algorithm does not face such difficulties, and we believe the analytical techniques we use to avoid this problem (discussed in the paragraph beginning on line 333) may be of independent interest to researchers working on average-reward RL.

---

> > ### Comment · Reviewer_88bt · 2025-08-09
> > **Thanks**
> >
> > Thanks for your response, I maintain my positive original evaluation.

---

### Official Review · Reviewer_wkNW · 2025-07-03

**Clarity:** 3
**Significance:** 3
**Originality:** 3
**Rating:** 4
**Confidence:** 3

**Summary:**

This paper studies average reward offline RL. In average reward setting, distribution shift and coverage issues are amplified. The paper targets these challenges and focus on overcoming limitations of prior theory that relied on uniform complexity measures. The main contribution is an offline RL algorithm with guarantees that depend only on the target policy’s characteristics, rather than on worst-case MDP-wide constants. They introduce two policy-dependent complexity metrics which measures how long it takes for the policy to reach a representative state in its stationary distribution support. It then derive the single-policy sample complexity bounds for average-reward offline RL.

**Questions:**

I have no questions.

**Ethical Concerns:**

["NO or VERY MINOR ethics concerns only"]

**Final Justification:**

I decide to maintain the current score.

**Limitations:**

yes

**Quality:**

3

**Strengths And Weaknesses:**

Strengths:

-The paper introduces policy-dependent complexity measures to characterize offline RL sample complexity, the bias span and the policy hitting radius. By depending only on the target policy’s dynamics, these metrics avoid the overly conservative uniform bounds.

-The paper provides matching upper and lower bounds, offering a convergence rate $\tilde{O}(\frac{1}{\sqrt{m}})$.

-The algorithm deals with the weakly communicating MDP, which is more general compared to related work.

Weaknesses:

The flip side of the paper’s single-policy focus is a stringent coverage assumption: the offline dataset must have adequate samples for every state-action pair along the optimal policy, even for states that the optimal policy would rarely or never visit on its own. This means the behavior policy that generated the dataset needs to have explored a bit of all parts of the state space (at least enough to “cover” the optimal action in each state). While the paper justifies that such assumption is provably necessary in the worst case, it still represents a potential weakness in applicability: if a dataset lacks any support in a part of the state space that turns out to be relevant, the theory suggests no offline algorithm can reliably avoid suboptimal decisions involving that part. The paper’s requirement is essentially to have at least on the order of $\tilde{O}(T_{\mathit{hit}}^2)$ samples for each state $s$ of the action $\pi^*(s)$. In many cases this might be reasonable as $T_{\mathit{hit}}$ could be moderate and this requirement does not scale with $m$, but in others it may be unrealistic. Thus, the results hinge on an idealized assumption that the offline data, while biased, is somewhat broad in coverage.

---

> ### Author Rebuttal · Authors · 2025-07-30
>
> Thank you for your positive comments. We agree that the transient data coverage assumption, which our lower bound shows is necessary, is nevertheless an assumption which may not always hold in practice. We believe that by identifying this obstacle, our work may help lead to future research which can circumvent this issue, possibly by identifying forms of side information which can be provided to algorithms to prevent them from failing in the difficult scenario where there is no transient data coverage. For example, it may be sufficient to have knowledge that all nonzero transitions have probability at least $p$ for some known $p > 0$ (enabling the algorithm to estimate the support of $P$). Also, as you mention, our work can handle more general classes of MDPs than prior work, and the amount of transient data needed for vanishing suboptimality is $O(1)$ (independent of $m$ and bounded in terms of $T_{hit}$ of only the target policy).

---

> > ### Comment · Reviewer_wkNW · 2025-08-06
> >
> > Thank you for the response. I will take it into consideration.

---

### Decision · Program_Chairs · 2025-09-17

**Decision:**

Accept (poster)

**Comment:**

This paper studies the sample complexity of offline RL in the average-reward setting. A notable contribution of this work is a novel lower bound demonstrating the necessity of a transient state coverage complexity measure that is unnecessary in other settings (e.g., finite horizon). While the lower bound does not match the upper bound exactly, the reviewers found the exiting result strong enough for publication.